

# Comparison of $CO_2$ from NOAA Carbon Tracker reanalysis model and satellites over Africa

Anteneh Getachew Mengistu [1] and Gizaw Mengistu Tsidu[1,2]

[1]Addis Ababa University, Addis Ababa, Ethiopia
[2]Botswana International University of Science and Technology, Palapye, Botswana

**Correspondence:** Anteneh G (antenehgetachew7@gmail.com)

**Abstract.** The scarcity of ground-based observations, poor global coverage and resolution of satellite observations necessitate the use of data generated from models to assess spatio-temporal variations of atmospheric $CO_2$ concentrations in a near continuous manner in a global and regional scale. Africa is one of the most data scarce region as satellite observation at the equator is limited by cloud cover and there are very limited number of ground based measurements. As a result, use of simulations from models are mandatory to fill this data gap. However, the first step in the use of data from models requires assessment of model skill in capturing limited existing observations. Even though, the NOAA Carbon Tracker model is evaluated using TCCON and satellite observations at a global level, its performance should be assessed at a regional scale, specifically in a regions like Africa with a highly varying climatic responses and a growing local source. In this study, NOAA CT2016 $CO_2$ is compared with the ACOS GOSAT observation over Africa using five years datasets covering the period from April 2009 to June 2014. In addition, NOAA CT2016 $CO_2$ is compared with OCO-2 observation over Africa using two years data covering the period from January 2015 to December 2016. The results show that the $XCO_2$ retrieved from GOSAT and OCO-2 are lower than CT2016 model simulation by 0.42 and 0.93 ppm on average respectively, which lie within the range of the errors associated with the GOSAT and OCO-2 $XCO_2$ retrievals. The mean correlations of 0.73 and 0.6, a regional precisions of 3.49 and 3.77 ppm, and the relative accuracies of 1.22 and 1.95 ppm were found between the model and the two data sets implying the performance of the model in Africa's land regions is reasonably good despite shortage of in-situ observations over the region assimilated in the model. These differences, however, exhibit spatial and seasonal scale variations. Moreover, the model shows some weakness in capturing the whole distribution. For example, the probability of detection ranges from 0.6 to 1 and critical success index ranges from 0.4 to 1 over the continent when the analysis includes data above the $95^{th}$ percentile and the whole data respectively. This shows the model misses the higher extreme ends of the $CO_2$ distribution. Spatially, GOSAT and OCO-2 $XCO_2$ are lower than that of CT2016 by upto 4 ppm over North Africa ($10\ ^0 - 35\ ^0N$) whereas it exceeds CT2016 $XCO_2$ by 3 ppm over Equatorial Africa ($10\ ^0S - 10\ ^0N$). Larger spatial mean biases of 2.11 and 1.8 ppm, 1.25 and 0.73 ppm in CT2016 $XCO_2$ with respect to that of GOSAT and OCO-2 are observed during winter (DJF) and spring (MAM) while small biases of -0.15 and 0.21 ppm, and 0.2 and -1.14 ppm are observed during summer (JJA) and autumn (SON) respectively. The model simulation has the ability to capture seasonal cycles with a small discrepancy over the North Africa and during winter seasons over all regions. In these cases, the model overestimates the local emissions and underestimate $CO_2$ loss.





## 1  Introduction

An understanding of the regional contributions and trends of carbon dioxide ($CO_2$) is critical to design mitigation strategies aimed at stabilizing atmospheric greenhouse gases. The present-day concentration of atmospheric $CO_2$ continues to rise. Ongoing emissions of $CO_2$ and other greenhouse gases will influence the global climate system during the next decades and

centuries. Approximately one-half of the $CO_2$ emissions from human activities is accumulating in the Earth's atmosphere, whereas the remaining portion of emitted $CO_2$ is absorbed by sink on land and in the ocean (Raupach et al., 2007).

Several studies (e.g., (Stocker, 2014; Deng et al., 2016)) have shown that the column averaged dry air mole fractions of $CO_2$ ($XCO_2$) has undergone rapid changes from pre-industrial period value of 280 ppm to 396 ppm as recently as 2013. This change has been attributed to anthropogenic factors such as fossil fuel combustion, land use change, biomass burning, emission

from industries such as cement. Notably, this value exceeds the highest $XCO_2$ level retrieved from ice cores representing the past 800,000 years. This increasing trend has been persistent. For example according to the World Meteorological Organization (WMO) 2016 report, the average concentrations of $CO_2$ hit 403.3 ppm, up from 400 ppm in 2015. As a result, there is evidence that shows an annual positive increasing rate of $1.99 \pm 0.43\ ppmyr^{-1}$ (http://www.esrl.noaa. gov/gmd/ccgg/trends/global.html Houghton, 2007). These positive trends have led to imbalance of $0.58 \pm 0.15\ Wm^2$ in energy budget between 2005 and

2010 at the top of the atmosphere (Hansen et al., 2011). To this end, changes in atmospheric temperature, hydrology, sea ice, and sea levels are attributed to climate forcing agents dominated by $CO_2$ (Santer et al., 2013; Stocker et al., 2013). However, understanding the climate response to anthropogenic forcing in a more traceable manner is still difficult due to a major uncertainty in carbon-climate feedbacks (Friedlingstein et al., 2006). Part of this uncertainty is due to lack of sufficient data on regional and global carbon cycle. This is compounded with inappropriate modeling practices to capture spatio-temporal

variability of carbon cycle. These problems can be solved through strengthening carbon monitoring networks and setting up proper modeling. A model, with appropriate physical and mathematical formulations and sufficiently tuned by observations, can be used to understand the spatio-temporal nature of atmospheric $CO_2$ source and sink as well its associated drivers.

Towards this, a number of national and international efforts have been initiated in the recent past by different government and non-government agencies across the globe. Among these efforts Ground-based observations of greenhouse gas using

Total Carbon Column Observing Network (TCCON) provide accurate and high–frequent $CO_2$ measurements. For example, it has been established that TCCON has a precision of 0.25% for measurements taken under clear sky conditions (Wunch et al., 2011). Therefore, TCCON is an important ground-based data source for validation and bias correction of retrieved $CO_2$ from satellite (Wunch et al., 2011). The number of TCCON sites is limited and can not establish accurate $CO_2$ amount and flux on subcontinental or regional scale. Moreover, some studies shows that the large uncertainty is amplified due to uneven

global distribution of TCCON sites (Gurney et al., 2002; Hungershoefer et al., 2010). In addition, none of these ground based observation networks were found in Africa.

On the other hand, the $CO_2$ concentration retrieved from the satellite based $CO_2$ absorption spectra have the advantages of unified, long-term, and the global coverage observation as compared to ground-based measurements. It has been established from theoretical studies that accurate and precise satellite derived atmospheric $CO_2$ can appreciably minimize the uncertainties



in estimated $CO_2$ surface flux (Rayner and O'Brien, 2001; Chevallier, 2007). Other studies have revealed that significant improvement in estimation of weekly and monthly $CO_2$ fluxes can be achieved subject to $CO_2$ retrieval error of less than 4 ppm from satellite and modeling scheme whereby $CO_2$ concentration is an independent parameter of carbon cycle model (Houweling et al., 2004; Hungershoefer et al., 2010). $XCO_2$ shows temporal variability on different time scales: diurnal,

synoptic, seasonal, inter-annual, and long term (Olsen and Randerson, 2004; Keppel-Aleks et al., 2011). More recent missions such as the Greenhouse gases Observing SATellite (GOSAT) (Hamazaki et al., 2005), the Orbiting Carbon Observatory-2 (OCO-2) (Boesch et al., 2011) and planned missions such as the Active Sensing of $CO_2$ Emissions over Nights, Days, and Seasons (ASCENDS) (Dobbs et al., 2008) have been and are being developed specifically to resolve surface sources and sinks of $CO_2$. GOSAT observations started in 2009 and provide $XCO_2$ based on spectra in the Short-Wavelength InfraRed (SWIR)

region. The $XCO_2$ derived from GOSAT exhibits standard deviation of about 2 ppm with respect to ground-based and in-situ air-borne observations (Yokota et al., 2009; NIES GOSAT Project, 2012).

On the other hand, atmospheric transport models, such as the NOAA Carbon Tracker (CT) is an integrated modeling system that assimilate $CO_2$ from other observations. The atmospheric transport in the NOAA CT integrated modeling system is simulated using the global two-way nested Transport Model 5 (TM5) forced by the time-varying meteorology from the European

Center for Medium-Range Weather Forecasts (ECMWF), and $CO_2$ fluxes due to terrestrial biosphere exchange.

Both satellite and model data should be validated against other independent satellite observations and/or in-situ observations before using them to answer scientific questions. As a result, a number of validation and intercomparison have been conducted in previous studies. For example, Kulawik et al. (2016) found root mean square error of 1.7, and 0.9 ppm in GOSAT and CT2013b $XCO_2$ relative to TCCON respectively. Other authors have undertaken validation exercises and found bias

of $-8.85 \pm 4.75\ ppm$ in NIES $XCO_2$ with respect to TCCON (Morino et al., 2010); root mean square error of $-1.48$ and $2.09\ ppm$ in NIES Level 2 V02.XX $XCO_2$ (Yoshida et al., 2013); and bias of $-0.68 \pm 2.56\ ppm$ in NIES level 2 V02.XX XCO2 with respect to air craft observations (Inoue et al., 2013). Moreover, strong consistency between the ACOS and NIES $XCO_2$ monthly averages time series over different regions was reported. For example, Deng et al. (2016) found the greatest mean difference $(1.43 \pm 0.60\ ppm)$ over China and the least over Brazil $(-0.03 \pm 0.64\ ppm)$ in the two time series of monthly

means. Globally, ACOS $XCO_2$ is higher than NIES by about 1 ppm and has smaller bias than NIES data. Moreover, comparison of NIES Level 2 V02.XX $XCO_2$ with GEOS-Chem model simulations revealed that the $XCO_2$ was lower than the model by 2 ppm on average globally (Lei et al., 2014).

OCO-2 the second world's full-time dedicated $CO_2$ measurement satellite was successfully launched by National Aeronautics and Space Administration (NASA) on 2 July 2014. OCO-2 measures atmospheric carbon dioxide with the accuracy,

resolution, and coverage required to detect $CO_2$ source and sink on global and regional scale. Liang et al. (2017) compared OCO-2 and GOSAT with TCCON. They found a mean measurement accuracy of -0.27 and -0.41 ppm with an RMSE of 1.56 and 2.216 ppm for $XCO_2$ measurements by OCO-2 and GOSAT with respect to TCCON respectively. Moreover, they found the measurement accuracy of GOSAT decreased to -0.62 ppm with an RMSE of 2.3 ppm during 2014 to 2016. Liang et al. (2017) also indicates GOSAT shows a larger seasonal variability in describing amplitudes than OCO-2, with greater amplitude

in the northern hemisphere than the southern hemisphere. Lei et al. (2014) also showed regional difference of $XCO_2$ between



the ACOS and NIES datasets. For example, a larger regional difference from 0.6 to 5.6 ppm was obtained over China land region, while it is from 1.6 to 3.7 ppm over global land region and from 1.4 to 2.7 ppm over US land region. These findings suggest that it is important to assess the accuracy and uncertainty of $XCO_2$ from models with respect to observations (e.g., GOSAT $XCO_2$) over other regions as well.

5    Therefore, this paper aims to assess the performance of Carbon Tracker model in capturing observed $XCO_2$ from GOSAT and OCO-2 satellites over Africa using various statistical metrics. Moreover, the skill of the model in capturing the amplitudes and phases of observed seasonal cycles over different parts of the continent is evaluated and the consistence of the modelled spatio-temporal variability with the known seasonal climatology of the regions, that determines carbon source and sink levels, is assessed.

## 10  2    Data and Methodology

### 2.1    Carbon Tracker Model

Carbon Tracker is an annually updated analysis of atmospheric carbon dioxide distributions and their surface fluxes (Peters et al., 2007). It is a data assimilation system that combines observed carbon dioxide concentrations from 81 sites around the world with model predictions of what concentrations would be based on a preliminary set of assumptions ("the first guess") about sources and sinks for carbon dioxide. Carbon Tracker compares the model predictions with reality and then systematically tweaks and evaluates the preliminary assumptions until it finds the combination that best matches the real world data. It has modules for atmospheric transport of carbon dioxide via weather systems, for photosynthesis and respiration, air-sea exchange, fossil fuel combustion and fires. Transport of atmospheric $CO_2$ is simulated by using the global two-way nested transport model (TM5). TM5 is an off line atmospheric tracer transport model (Krol et al., 2005) driven by meteorology from the European

20    Center for Medium-Range Weather Forecasts ($ECMWF$) operational forecast model and from the ERA Interim reanalysis (Dee et al., 2011) to propagate surface emissions. TM5 is based on a global $3^0 \times 2^0$ and at a $1^0 \times 1^0$ spatial grids over North America. Carbon Tracker CT2015 (http://carbontracker.noaa.gov; Peters et al. (2007) is used to extend aircraft profiles from the stratosphere to the top of the atmosphere (Inoue et al., 2013; Frankenberg et al., 2016) and to quantify co-location error (Kulawik et al., 2016). The older data versions have been used and also compared with different data sets over other parts of

25    the globe in previous studies (Peters et al., 2007; Nayak et al., 2014; Kulawik et al., 2016; Krishnapriya et al., 2017). Most of the studies confirm that CT $XCO_2$ captures observations reasonably well. In this study we use CarbonTracker release version CT2016 and CT-NRT.v2017, here after (CT2016 and CT2017 respectively). Both versions of NOAA CT2016 provides 3 hourly $CO_2$ mole-fractions data for global atmosphere at 25 pressure levels in a $3^0 \times 2^0$ spatial resolution for a period covering 2000 to 2016. The data can be accessed freely at the public domain (ftp://aftp.cmdl.noaa.gov/products/carbontracker).





## 2.2 GOSAT measurements

GOSAT is the world's first spacecraft to measure the concentrations of carbon dioxide and methane, the two major greenhouse gases, from space. The spacecraft was launched successfully on January 23, 2009, and has been operating properly since then. GOSAT records reflected sunlight using three near-infrared band sensors. The field of view at nadir allows a circular footprint of about 10.5 km diameter (Kuze et al., 2009; Yokota et al., 2009; Crisp et al., 2012). GOSAT consists of two instruments. The sensors for the two instruments can be broadly labeled as thermal, near infrared and imager. The first two sensors are used as part of Fourier Transform Spectrometer for carbon monitoring which is referred to as TANSO-FTS while the imager for cloud and aerosol observations is referred to as TANSO-CAI. The details on spectral coverage, resolution, field of view, and different products of TANSO-FTS in the three SWIR bands can be found in a number of previous studies (Kuze et al., 2009; Saitoh et al., 2009; Yokota et al., 2009, 2011; Crisp et al., 2012; Nayak et al., 2014; Deng et al., 2016, and references therein) . In this study $XCO_2$ from GOSAT Level 2 (L2) retrieval based on the SWIR spectra of FTS observations and made available by Atmospheric $CO_2$ Observations from Space (ACOS) of NASA is used. As noted in Section 1, ACOS has lower bias and better consistency than NIES GOSAT SWIR L2 $CO_2$ globally. Therefore, our choice of the ACOS $CO_2$ is motivated by these differences.

## 2.3 OCO-2 measurements

OCO-2 has three-band spectrometer, which measures reflected sunlight in three separate bands. The $O_2$ A-band measures molecular absorption of oxygen from reflected sunlight near 0.76 $\mu m$ while the $CO_2$ bands are located near 1.61 $\mu m$ and 2.06 $\mu m$ (Liang et al., 2017). In this study, $XCO_2$ from OCO-2 from January 2015 to December 2016 is used. The OCO-2 project team at Jet Propulsion Laboratory, California Institute of Technology, produced the OCO-2 data used in this study. The data can be accessed from NASA Goddard Earth Science Data and Information Service Center

## 2.4 Methods

The analyzed GOSAT and model $XCO_2$ time series spans five years, ranging from April 2009 to June 2014. Atmospheric $CO_2$ concentrations of NOAA Carbon-Tracker 2016 (CT2016) have a global coverage with a $3^0 \times 2^0$ Longitude/Latitude resolution which covers 428 grid points in our study area. Satellite observations, however, is different from model assimilation, and have gaps because of various reasons (e.g., cloud and the observational mode of satellite). As a result, there is no one to one spatio-temporal match between the two data sets. For example, $XCO_2$ products from the two dataset are not directly comparable since CT is a 3 hourly smooth and regular grid dataset whereas GOSAT $XCO_2$ is irregularly distributed in space and time. Thus, the CT2016 $XCO_2$ is extracted on the time and location of GOSAT-$XCO_2$ data. Using the grid point of CT2016 as a reference bin, the corresponding GOSAT L2 $XCO_2$ found with in a rectangle of $1.5^0 \times 1.5^0$ with center at the reference bin of CT2016 and with temporal mismatch of a maximum of 3 hrs is extracted. Correlation coefficients (R), bias and root mean square deviation (RMSD) are used to assess the level of agreement between the two data sets. The bias is calculated as the mean of the differences in this work. This study also applies extended categorical contingency table for evaluation of performance




of CT2016 in capturing the different parts of observed $XCO_2$ distribution. The suggested categorical metrics includes the Probability of Detection (POD), False Alarm Ratio (FAR), Critical Success Index (CSI) also known as the Threat Score and Categorical miss. POD quantifies the fraction of reference observations detected correctly by the simulation. Meaningful values of POD ranges from 0 (no skill) to 1 (perfect skill); FAR identifies the fraction of events captured by simulation but not available

in reference observations. Sound values of FAR is bounded by 0 to 1 with 0 implying perfect score. CSI combines different aspects of the POD and FAR to characterize the overall performance of the simulation in capturing observation. The CSI is constrained to have values between 0 (no skill) to 1 (perfect skill) by definition. Categorical miss quantifies events identified by reference observation but missed by the simulation. Therefore, by definition, categorical miss ranges from 0 (perfect score) to 1 (no skill). More details about these categorical statistical metrics can be found in works by other authors (e.g., AghaKouchak

et al., 2011; Wilks, 2011; AghaKouchak and Mehran, 2013,  and references therein) . Using similar coincidence criteria and statistical methods, CT2016 and OCO-2 $XCO_2$ are also compared. The analyzed $XCO_2$ time series spans two years, ranging from January 2015 to December 2016.

## 3   Results and discussions

### 3.1   Comparison of $XCO_2$ mean climatology from NOAA CT2016 and ACOS GOSAT

The mole fraction of $CO_2$ obtained from the NOAA carbon tracker model and ACOS GOSAT observation was compared. The results are based on 428 grid pints uniformly distributed to cover the whole Africa's land region in a $3^0 \times 2^0$ longitude/latitude resolution. The analysis was on five years daily data starting from April 2009 to June 2014. The $XCO_2$ comparison was done only when there are more than ten $XCO_2$ retrievals that fulfills the spatio-temporal matching criteria defined in Section 2.4

Fig. 1 shows the five-years average of CT2016 (Fig. 1a) and GOSAT (Fig. 1b) $XCO_2$ distribution. The major common

spatial feature in the mean map of $XCO_2$ from GOSAT and CT2016 reanalysis is dipole structure characterized by high $XCO_2$ northward of equator and low $XCO_2$ southward of equator with the exception of Congo basin which is characterized by spatially anomalous high $XCO_2$. The Southern Africa region is characterized by weak anthropogenic $CO_2$ emission and high $CO_2$ uptake by the vegetation. This contributed to the observed dipole distribution. Another important pattern is anomalous peak over annual average location of ICTZ which appears to fade over Eastern Africa. The two data sets are also characterized

by high spatial mean correlation of 0.73, a global offset of 0.43 ppm, which is the average bias, a regional precision of 3.49 ppm, and a relative accuracy of 1.22 ppm as depicted in Table 1.

**Table 1.** Summary of statistical relation between CT2016 and GOSAT observation. The statistical tools shown are the mean correlation coefficient (R), the average of bias, the average root mean square error (RMSE), the standard deviation in bias (std of Bias), mean GOSAT satellite retrieval error(GOSAT err), the standard deviation of CT2016 (std of CT2016) and the standard deviation of GOSAT (std of GOSAT). The number of data used in the statistics is 750.

| Statistical tool | R | Bias | RMSE | std of Bias | GOSAT err | std of CT2016 | std of GOSAT |
|---|---|---|---|---|---|---|---|
| Values | 0.73 | 0.43 | 3.47 | 1.22 | 0.9 | 5.24 | 4.32 |





Fig. 1c shows the mean differences (CT2016–GOSAT) from April 2009 to June 2014 which ranges from -3 to 4 ppm. The highest difference between the CT2016 and GOSAT (4 ppm) is observed over Congo basin. The region is known for its rain forest. The likely explanation could be $CO_2$ flux from respiration from forest in the region is overestimated in the reanalysis. On the other hand, CT2016 underestimate $XCO_2$ along the equator and southwestern Ethiopia and South Sudan which are

5    also known for near-year round rainfall and relatively dense vegetation. However, the mean five years climatology may also be slightly positively biased due to fewer observations as shown in Fig.1d. The number of datasets used for comparison range from 10 - 2000 with a mean of 682. The strategy and methods for cloud screening in ACOS retrievals could lead to smaller number of observation in the equatorial region (Crisp et al., 2012; Yoshida et al., 2013). Fig. 1c also shows GOSAT observations are overall less than the values of model simulations over regions northward of $13^0N$ while GOSAT observations are greater than

10    model simulations over regions southward of equator with the exception of Kalahari desert region. The spatial distribution of global atmospheric $CO_2$ is not uniform because of the irregularly distributed sources of $CO_2$ emissions, such as large power plant and forest fire, and biospherical assimilation as clearly noted above.

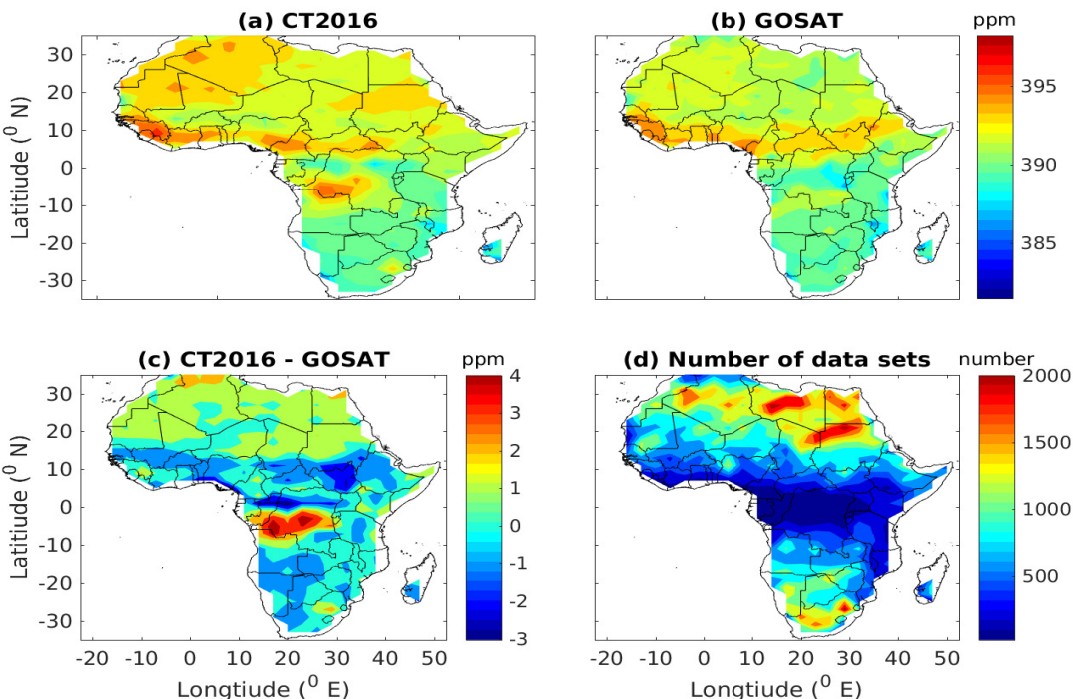

**Figure 1.** Distribution of five-years averages of CT2016 **(a)** and GOSAT **(b)** $XCO_2$ and their difference **(c)** gridded in $3^0 \times 2^0$ bins over Africa's Land mass; and the total number of datasets at each grid (d).

Fig. 2a shows the histograms of differences of CT2016 and GOSAT. The mean difference between CT2016 and GOSAT means is about 0.42 ppm with the standard deviation of 1.31 ppm indicating better consistency and less potential outliers.




Moreover, a positive mean of the difference implies that CT2016 $XCO_2$ mostly overestimates the $CO_2$ concentration in Africa.

Because of selection criteria which require a difference of upto 1.5 degree longitude and latitude, the two datasets are not exactly at the same point. The impact of the relative distance between them should be assessed before performing any

statistical comparison. Fig. 2b depicted color coded scatter plot of CT2016 model simulation verses GOSAT to determine if the discrepancy between the data sets arise from spatial mismatch. The color code indicates the relative distance between the model and observation datasets. For these datasets the $50^{th}$ percentile has a relative distance of $1.2^0$ which means 50% of the data has a relative distance of shorter than $1.2^0$. The maximum relative distance between them is $2.12^0$. However, there is no indication that this has been the case since the scatter is not a function of relative distance between the data sets. For

example, data points with blue color code with lowest location difference is scattered everywhere instead of along the 1:1 line. Furthermore, we found the bias of 0.71 ppm, correlation coefficient of 0.72 and RMSE of 3.77 ppm for datasets which has a relative distance below $1.2^0$. On the other hand, the bias , correlation coefficient and RMSE are 0.59 ppm, 0.72 and 3.70 ppm for those which are above $1.2^0$.

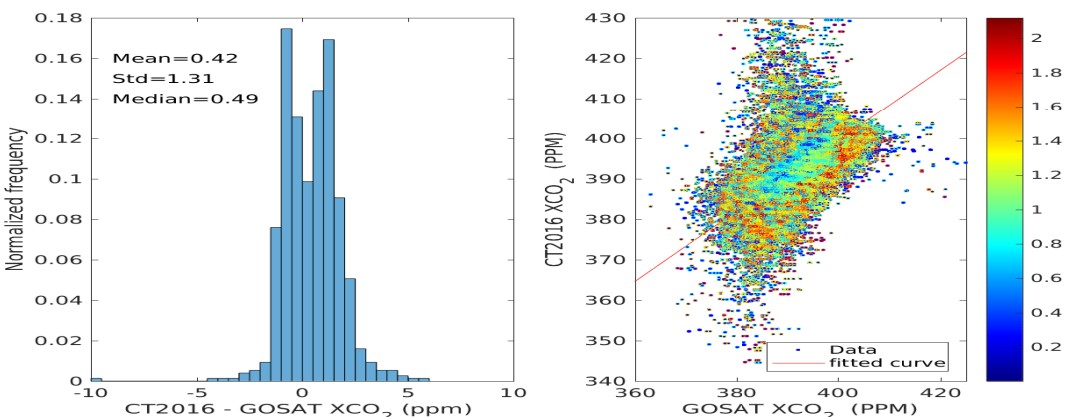

**Figure 2.** Histogram of the difference of CT2016 relative to GOSAT (left panel) and color code scatter diagram of $CO_2$ concentration as derived from CT2016 and GOSAT (right panel). Color indicates the relative distance as shown in colorbar between datasets.

Fig. 3 shows a statistical comparisons of $XCO_2$ from the CT2016 model simulation and GOSAT observation over Africa.

The number of data used are shown in Fig. 1d. There is a good consistency between them. As it is depicted in Fig. 3a, the bias ranges from -3 to 4 ppm with a mean bias of 0.43 ppm. Overall positive bias over northern half of Africa and normal to negative bias over Southern Africa except some isolated pockets over Congo and Angola. Fig. 3b depicts correlation coefficient between GOSAT and Carbon Tracker $XCO_2$. The correlation varies from zero over some isolated pockets over southern Congo and North Africa to 0.9 ppm over subsaharan Africa, Eastern Africa and most of Southern Africa. The region with poor correlation

also exhibit high RMSE as shown in Fig. 3c. To understand whether this discrepancy originates from model weakness alone, we have looked at the GOSAT retrieval errors which are high over the same regions with high bias and RMSE between GOSAT and



carbon tracker $XCO_2$ (Fig. 3d). Therefore, part of the discrepancy is clearly linked to satellite own uncertainty, which might have been amplified due to small number of data points used to calculate the mean error of GOSAT $XCO_2$ measurements.

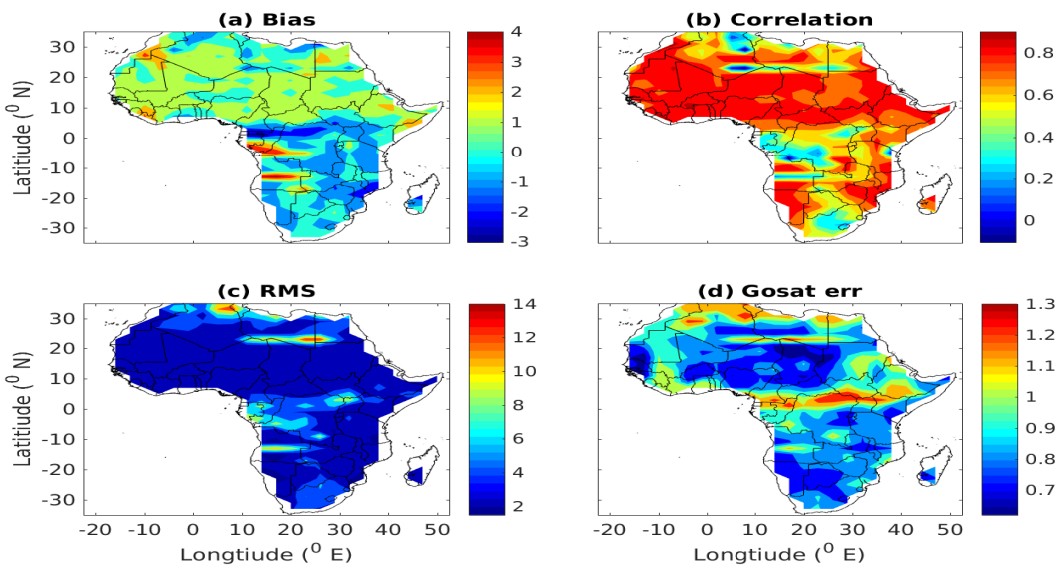

**Figure 3.** Spatial patterns of bias (a), correlation (b), RMSE (c) of the two data sets, and GOSAT $XCO_2$ total retrieval errors (d).

## 3.2   Categorical comparison of $XCO_2$ from NOAA CT and GOSAT

Following the methods described in AghaKouchak et al. (2011), quantile bias (QBias), POD, CSI, FAR and Categorical miss
(MISS) are determined from the coincident $XCO_2$ data to assess the skill of carbon tracker model in capturing different parts of
$XCO_2$ distribution. We filter out pixels in which the total number of observations are less than 10 to avoid unreliable statistics.
Fig. 4 displays values for Bias, POD, CSI, FAR and MISS for distribution exceeding 5% (first row), 75% (second row), 90%
(third row) and 95% (fourth row) quantiles. The thresholds are set based on the quantiles of the GOSAT observation. POD
and CSI decrease at higher quantiles. In contrast, Bias, FAR and MISS increase at higher quantiles. Specially the decrease in
critical success index is significant. It ranges from 0.8 to 1.0 at $5^{th}$ percentile ( i.e., CSI for values exceeding the $5^{th}$ percentile)
and smaller than 0.4 at $95^{th}$ percentile.

On the other hand, the false alarm ratio which is below 0.2 at threshold exceeding $5^{th}$ percentile shows a value above 0.6
for most regions at threshold exceeding the $95^{th}$ percentile. This indicates that the over all skill of the model deteriorates as
the comparison data range includes only higher extremes of the $XCO_2$ distribution. However, when the data covers lower
extremes, POD and CSI improve suggesting that the model performs well at the lower end of the $XCO_2$ distribution in
capturing observations. Nevertheless, on average over the continent at all quantiles the critical miss is lower than 0.3 indicating
70% to 97% (see also Table 2) of datasets detected by the observation are perfectly simulated by the model.





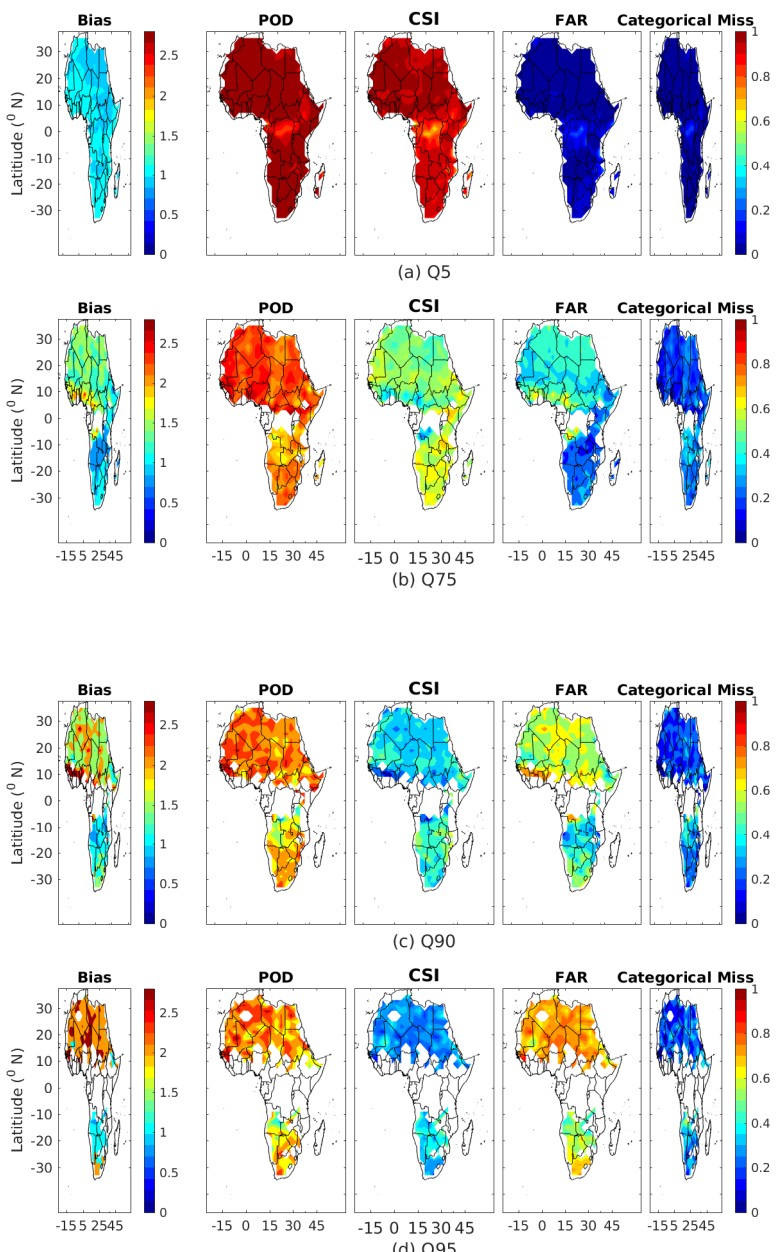

**Figure 4.** Distribution of categorical metrics over the study area for quantiles exceeding 5% (first row), 75% (second row), 90% (third row) and 95% (fourth row)

Figs. 4b - 4d show the scarce datasets in Equatorial Africa and northwestern Africa to certain extent. This is the main reason for large biases observed around the Equator in Fig. 1 and also the corresponding retrieval error in GOSAT $XCO_2$ in Fig. 3d





**Figure 5.** Summary of categorical metrics (Bias, POD, CSI, FAR and Categorical miss) averaged over Africa's land region for 5, 10, 25, 50, 75, 90 and 95 percentile.

**Table 2.** Summary of extended contingency metrics for the relation between CT2016 model simulation and GOSAT observation.

| Quantiles | Bias | POD | FAR | MISS | CSI | Number of coincident observations |
|-----------|------|------|------|------|------|-----------------------------------|
| Q5 | 1.02 | 0.972 | 0.044 | 0.027 | 0.932 | 719 |
| Q10 | 0.988 | 0.928 | 0.056 | 0.072 | 0.881 | 716 |
| Q25 | 0.996 | 0.886 | 0.099 | 0.114 | 0.803 | 701 |
| Q50 | 1.106 | 0.881 | 0.190 | 0.119 | 0.729 | 680 |
| Q75 | 1.293 | 0.784 | 0.370 | 0.216 | 0.532 | 645 |
| Q90 | 1.756 | 0.766 | 0.526 | 0.234 | 0.409 | 508 |
| Q95 | 2.423 | 0.738 | 0.662 | 0.262 | 0.297 | 367 |

since individual retrieval errors are smoothed out during averaging over large number of coincident observations. In this section we define bias as the ratio of sum of simulation to sum of observation. Fig. 4a shows the POD and CSI are above 0.88 for lower quantiles which indicates that the skill of the model to simulate the observation is above 88% (see also Table 2). However, at higher quantiles the change in contingency metrics show a clear deviation between the model simulation and observation. Fig. 4d shows the bias in $XCO_2$ values exceeding 95 percentile is such that Carbon Tracker $XCO_2$ is more than twice GOSAT $XCO_2$ values over regions northward of $10^oN$ while this figure ranges from 1 (no bias) over most of Southern Africa to 2 (positive bias) over South Africa. Consistent with this result, the critical success index is smaller than 0.2 and the false alarm ratio is above 0.6 over North Africa. This indicates that the ability of the model to simulate the observation is generally getting weak over the whole continent towards the extreme higher ends of the $XCO_2$ distribution although there is regional disparity as noted from weaker performance over North Africa than over Southern Africa.

In addition, quantile bias, POD, CSI, FAR categorical miss are calculated for all data, data that includes values higher than 0, 5, 10, 25, 50, 75, 90 and 95 percentile and averaged over the whole African land mass as shown in Fig. 5. There is one major conclusion that can be drawn from Fig. 5, i.e., the quantile bias, FAR and categorical miss increase with increase in the quantile thresholds while POD and CSI decrease.





### 3.3 Comparison of monthly average time series of NOAA CT2016 and GOSAT $XCO_2$

Africa is one of the largest continents covering both northern and southern hemispheres. As a result, the continent is under the influence of semi-permanent high pressure cells which led to the Sahara Desert in the North and the Kalahari in the South. The equatorial low pressure cell which allows formation of the seasonally migrating intertropical convergence zone is part

of the major large scale atmospheric circulation systems. These large scale pressure systems, Oceanic circulations and their interaction with the atmosphere coupled with diverse topographies of the region allow for the formation of different climates (e.g., equatorial, tropical wet, tropical dry, monsoon, semi desert (semi arid), desert (hyper arid), subtropical high climates). Geographically, the Sahel, a narrow steppe, is located just south of Sahara; the central part of the continent constitutes the largest rainforest next to Amazon whereas most southern areas contain savana plains. The continent get rainfall from migrating

intertropical convergence zones, west Africa monsoon, intrusion of mid-latitude frontal systems, traveling low pressure systems (Mitchell, 2001, and references therein). Since $CO_2$ fluxes exhibit seasonal variability and Africa experiences different seasons as noted above, it is important to divide Africa into three major regions, namely North Africa (10 to 35 $^0N$), Equatorial Africa (10 $^0S$ to 10 $^0N$), and Southern Africa (35 to 10 $^0S$) and conduct the comparison of the two $XCO_2$ datasets.

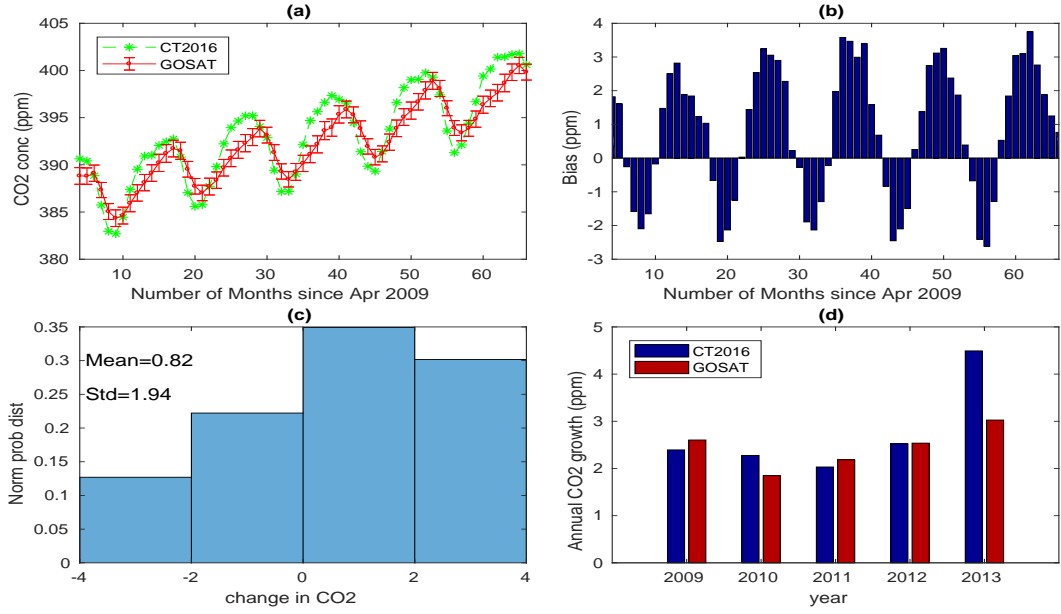

**Figure 6.** The monthly mean time series of CT2016 and ACOS-GOSAT from April 2009 to June 2014 averaged over North Africa (a), bias associated to the monthly means (b), the histogram of difference (c) and the annual growth rate obtained by subtracting the mean from the mean of the next year (d).

Figs. 6 - 8 shows time series of $XCO_2$ monthly means of five years data during the period of April 2009 to June 2014 for

both CT2016 and ACOS GOSAT over North Africa, Equatorial Africa and Southern Africa respectively. Figs. 6a - 8a depict





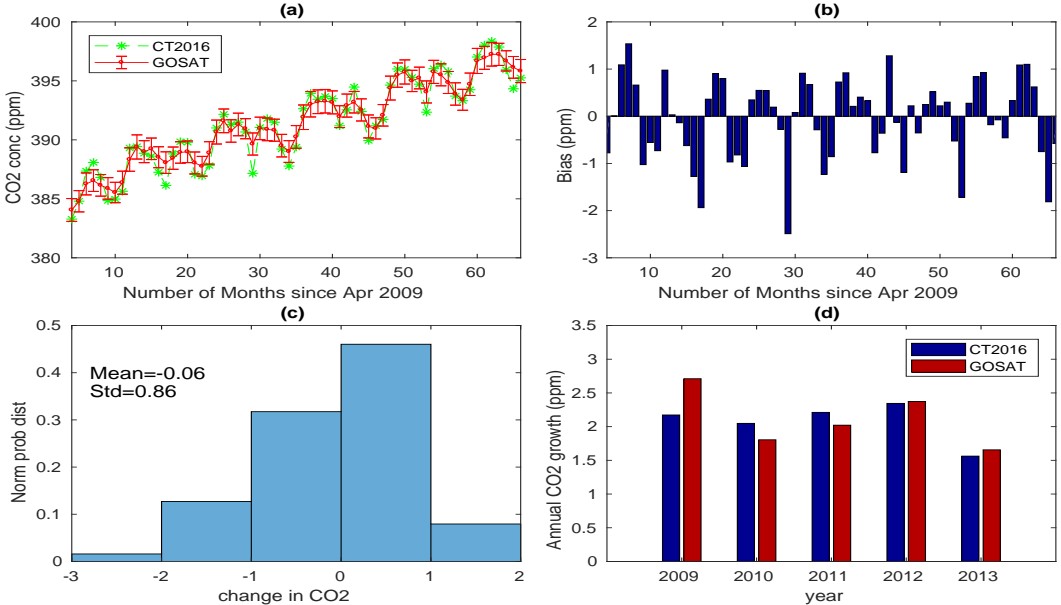

**Figure 7.** The monthly mean time series of CT2016 and ACOS-GOSAT from April 2009 to June 2014 averaged over Equatorial Africa (a), bias associated to the monthly means (b), the histogram of difference (c) and the annual growth rate obtained by subtracting the mean from the mean of the next year (d).

**Table 3.** Summary of statistical relation between CT2016 and GOSAT observation. The statistical analysis were made using monthly averaged time series based on 63 data points (i.e., months from April 2009 to June 2014).

| Statistics | R | $Bias$ | $RMSE$ | std in Ct | std in GOSAT |
|---|---|---|---|---|---|
| Africa | 0.96 | 0.44 | 1.17 | 3.95 | 3.53 |
| North Africa | 0.82 | 0.96 | 2.09 | 4.91 | 3.99 |
| Equatorial Africa | 0.9 | -0.05 | 0.86 | 3.71 | 3.46 |
| Southern Africa | 0.98 | -0.35 | 0.71 | 3.42 | 3.35 |

the existence of an overall good agreement for the monthly averages with respect to amplitudes and phase of $XCO_2$. With the exception over North Africa, CT2016 captures GOSAT's observation within its associated errors.

Fig. 6a shows that $CO_2$ concentration from both CT2016 and GOSAT reaches maximum in March (397.02 ppm for CT2016 and 394.36 ppm for GOSAT) and minimum in September (387.41 for CT2016 and 388.81 for GOSAT) over North
5 Africa. CT2016 also shows minimum in August (387.37 ppm). The largest monthly mean difference of 3.15 ppm between the two datasets is observed in March, while the smallest value of -2.2 ppm is found in September and August (see also Table 4). In addition, both datasets show concentration of $CO_2$ increases from October to March and decreases from June to September. However, the results from CT2016 shows a gradual decreasing trend from April to June. Conversely, GOSAT shows a gradual increasing trend. This is most likely CT2016 simulation respond to the growing size of sink following short rain season.





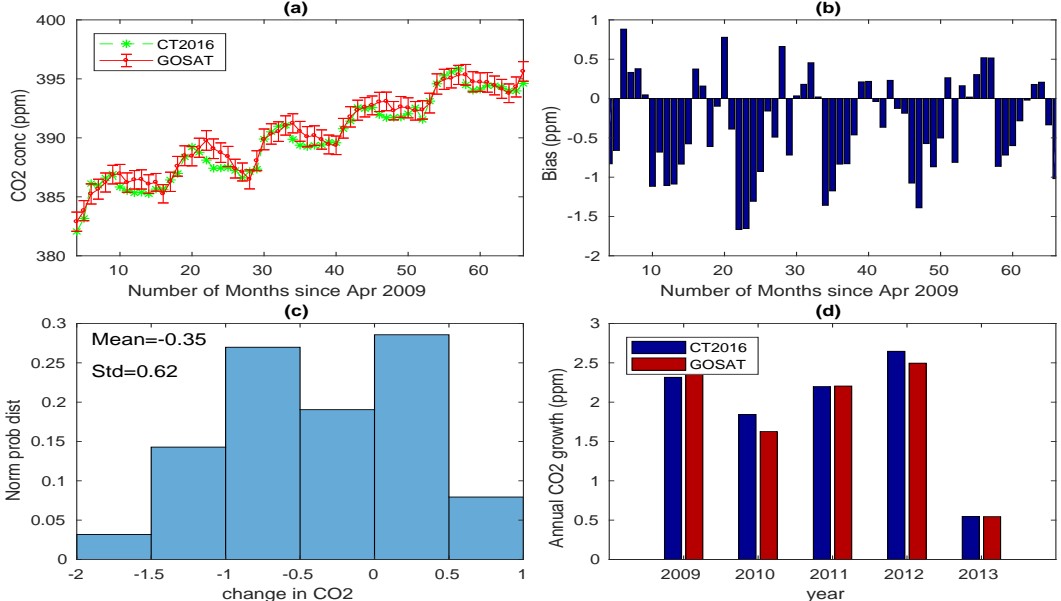

**Figure 8.** The monthly mean time series of CT2016 and ACOS-GOSAT from April 2009 to June 2014 averaged over Southern Africa (a), bias associated to the monthly means (b), the histogram of difference (c) and the annual growth rate obtained by subtracting the mean from the mean of the next year (d).

Fig. 7a shows $CO_2$ concentration reaches maximum (393.29 ppm for CT2016 and 393.29 ppm for GOSAT) in January and minimum (388.84 for CT2016 and 389.34 ppm for GOSAT) in October over Equatorial Africa. The largest monthly mean difference of 1.1 ppm between the two datasets observed in July and the smallest of -1.45 ppm in May (Table 4). Moreover, both datasets show that concentration of $CO_2$ increases from November to January and May to June while it decreases from

January to May and June to October. This similarity in the seasonal variability of the two datasets shows that the two datasets are in good agreement in terms of amplitude and phase. Fig. 8a shows maximum $CO_2$ concentration in September (391.04 ppm for CT2016) and October (392.27 for GOSAT), and minimum (388.3 ppm for both CT2016 and GOSAT) in April over Southern Africa. The largest monthly mean difference of 0.40 ppm between the two datasets is observed in August while the smallest (-1.22 ppm) in October (Table 4). Both datasets show concentration of $CO_2$ increases from May to September while it

decreases from November to April. Consistent with this evidence, other authors (e.g., Zhou et al., 2008) have indicated presence of a strong absorption of $CO_2$ by vegetations during August in the northern hemisphere. In contrast, there are evidences of a weak absorption by vegetation and a strong emission from human activities during winter in the same hemisphere (Liu et al., 2009; Kong et al., 2010). Because of the presence of strong source and sink, discrepancy between the model and observation is notable over North Africa.

Figs. 6b - 8b show regional averaged bias in the monthly mean time series between CT2016 and GOSAT covering the period of April 2009 to June 2014. The figures show presence of seasonally varying bias. Moreover, the bias is positive in dry seasons



**Table 4.** Five years monthly averaged $XCO_2$ concentration obtained from CT2016 (CT) and ACOS GOSAT (GO) and their difference $CT - GO$ (D) in ppm over Africa (A), North Africa (NA), Equatorial Africa(EA) and Southern Africa (SA).

| Month | A CT | A GO | A D | NA CT | NA GO | NA D | EA CT | EA GO | EA D | SA CT | SA GO | SA D |
|---|---|---|---|---|---|---|---|---|---|---|---|---|
| January | 393.81 | 392.15 | 1.67 | 395.77 | 392.62 | 3.15 | 393.92 | 393.29 | 0.62 | 389.68 | 390.48 | -0.80 |
| February | 394.26 | 393.59 | 1.67 | 396.53 | 393.55 | 2.99 | 393.59 | 393.20 | 0.39 | 389.67 | 390.06 | -0.40 |
| March | 394.93 | 393.27 | 1.65 | 397.02 | 394.36 | 2.65 | 393.39 | 393.21 | 0.18 | 389.76 | 389.84 | -0.08 |
| April | 393.86 | 392.77 | 1.09 | 396.12 | 394.34 | 1.78 | 390.89 | 391.44 | -0.55 | 388.33 | 388.36 | -0.03 |
| May | 393.16 | 392.85 | 0.31 | 395.81 | 394.94 | 0.86 | 389.33 | 390.78 | -1.45 | 389.13 | 389.37 | -0.24 |
| June | 392.56 | 392.81 | -0.25 | 394.17 | 394.50 | -0.32 | 391.84 | 391.70 | 0.15 | 390.60 | 390.78 | -0.18 |
| July | 390.14 | 391.10 | -0.97 | 389.44 | 391.60 | -2.16 | 392.11 | 390.20 | 1.10 | 390.54 | 390.35 | 0.19 |
| August | 389.07 | 390.05 | -0.98 | 387.37 | 389.59 | -2.22 | 391.25 | 390.67 | 0.59 | 390.97 | 390.57 | 0.40 |
| September | 388.53 | 389.48 | -0.96 | 387.41 | 388.81 | -1.39 | 388.96 | 389.69 | -0.73 | 391.04 | 391.04 | 0.00 |
| October | 389.46 | 389.75 | -0.23 | 389.40 | 389.32 | 0.08 | 388.84 | 389.34 | -0.49 | 390.05 | 391.27 | -1.22 |
| November | 391.11 | 390.43 | 0.68 | 391.96 | 390.34 | 1.62 | 389.77 | 390.46 | -0.69 | 389.61 | 390.74 | -1.12 |
| December | 393.22 | 391.51 | 1.70 | 394.50 | 391.61 | 2.89 | 392.93 | 392.40 | 0.53 | 389.58 | 390.46 | -0.88 |

and negative in wet seasons. This suggests that the model simulation tends to underestimate when the terrestrial biosphere becomes a stronger sink while it overestimates when the sink is weaker. Fig. 6b shows the monthly average bias is mostly above 2 ppm during dry seasons and below -1 ppm during wet seasons over the North Africa. On the other hand, Figs. 7b and 8b show the monthly average bias mostly ranges from -1 to 1 ppm in Equatorial Africa and Southern Africa.

Figs. 6c - 8c show the histogram of difference. The mean difference between CT2016 simulation and GOSAT observation of $XCO_2$ is 0.82 ppm with a standard deviation of 1.49 over North Africa (see Fig. 6c). This implies that CT2016 overestimates $XCO_2$ by 0.82 ppm over North Africa. Fig. 7c presents a mean difference of -0.06 ppm with standard deviation of 0.86 which indicates that CT2016 captures observed $XCO_2$ very well over Equatorial Africa. Fig. 8c reveals a mean difference of -0.35 ppm and a standard deviation of 0.6 which indicates that CT2016 underestimates observed $XCO_2$ over Southern Africa.

Figs. 6d - 8d display annual growth rate of $XCO_2$ which ranges from 1.5 - 2.7 ppm. Moreover, the two datasets are consistent in determining the annual growth rate with an exception during 2013. In 2013 CT2016 model simulates annual growth rate exceeding 4 ppm over North Africa. Conversely, GOSAT observation shows annual growth rate of 3 ppm. The results are found in good agreement with the observed variability in the global annual growth rate from surface measurements (http://www.esrl.noaa.gov/ gmd/ccgg/trends/global.html) which is 1.67, 2.39, 1.70, 2.40, 2.51 $ppm\ yr^{-1}$ global during 2009 -

2013 respectively, and 1.89, 2.42, 1.86,2.63, 2.06 $ppm\ yr^{-1}$ for Mauna Loa during 2009 - 2013 respectively, with error bars of 0.05 - 0.09 $ppm\ yr^{-1}$ for global and 0.11 $ppm\ yr^{-1}$ for Mauna Loa data sets. The results from both model and observation suggest that the growth rate over North Africa is greater than the global growth rate by about 1 ppm while it is lower than the global growth rate over Southern Africa by the same margin in 2013. This is most likely due to the observed high $CO_2$





emission in 2013. Olivier et al. (2017) reports 1.8% $CO_2$ emission in 2013 which is exceptionally high as compared to 0.6% and 0.8% in 2012 and 2014.

### 3.4 Comparison of seasonal climatology

Seasonal cycle has important implications for flux estimates (Keppel-Aleks et al., 2012). It is important to analyze whether

there are seasonally dependent biases that are affecting the seasonal cycle, and whether the data sets are capturing the same seasonal cycle. The four seasons considered here are: winter (December, January and February ), spring (March, April and May ), summer (June, July and August ), and autumn (September, October and November ). Fig. 9 shows the seasonal distributions of CT2016 (left panels) and GOSAT (middle panels) $XCO_2$ and their difference (CT2016 - GOSAT, right panels). The distribution clearly shows that $CO_2$ concentration is maximum during spring (MAM) over Africa land mass and minimum

during autumn (SON) over the North Africa and during winter (DJF) over the Southern Africa. These features are in good agreement with the rainfall climatology of northern and southern hemispheres. Moreover, Table 5 shows a seasonal varying biases. Seasonal biases affect the seasonal cycle and amplitudes, which are important for biospheric flux attribution.

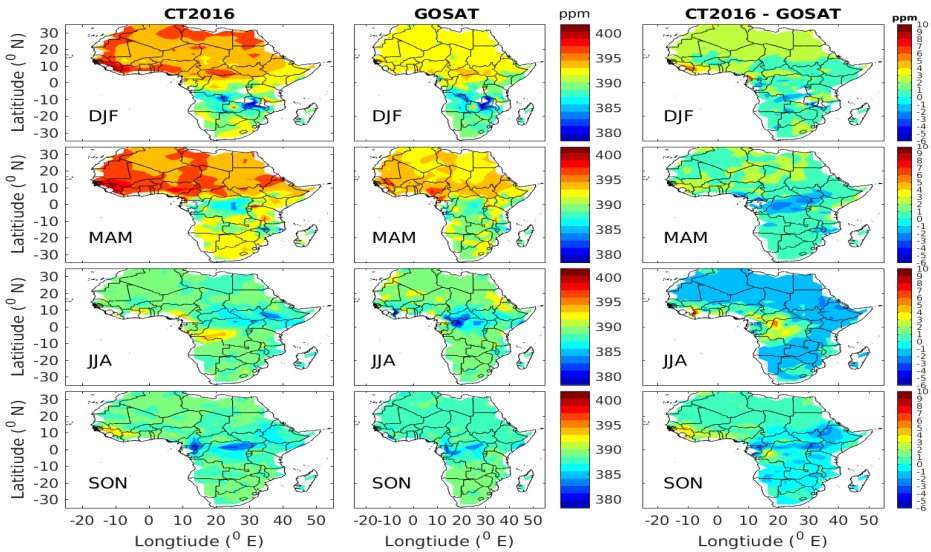

**Figure 9.** Seasonal climatology of $CO_2$ for NOAA CT2016 (left panels) and ACOS GOSAT (midel panels) and their difference (right panels).

The right panels in Fig. 9 show that the seasonal mean difference (CT2016 - GOSAT) ranges from -6 to 10 ppm. The highest spacial mean seasonal bias of 2.11 ppm between CT2016 and GOSAT is observed during winter (DJF). Fig. 9 (right panels)

also shows that a maximum difference of 10 ppm over the gulf of Guinea, above 2 ppm over North and Western Africa, a minimum of -5 ppm over Congo and within a range of -2 to 2 ppm over Eastern Africa and Southern Africa. This implies high





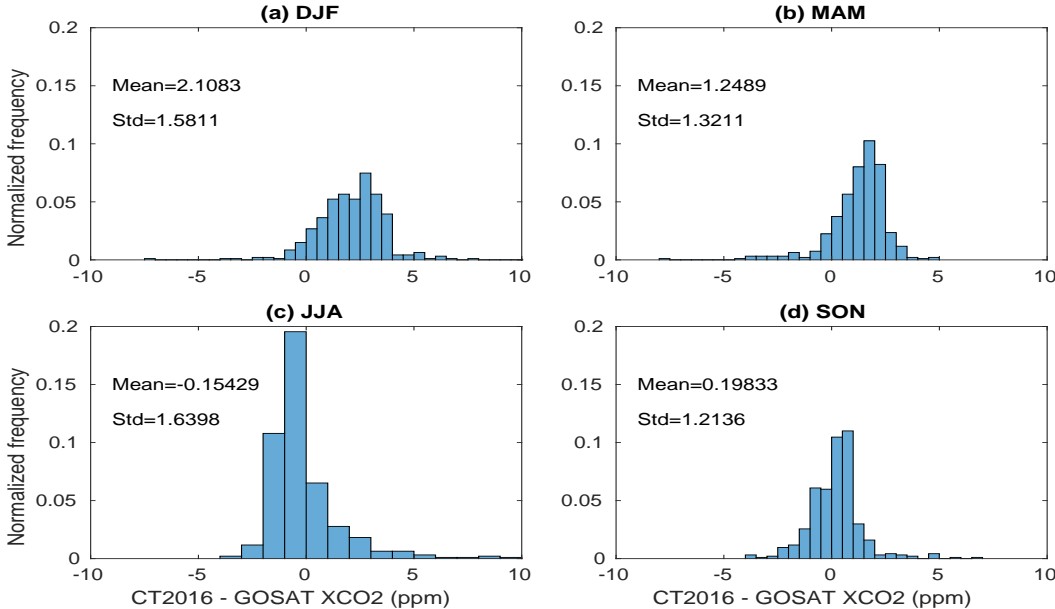

**Figure 10.** Histogram of difference for the seasonal $CO_2$ climatology for DJF (a), MAM(b), JJA (c) and SON (d) seasons.

**Table 5.** Summary of statistical relation between CT2016 and GOSAT $XCO_2$. The statistical analysis were made using monthly averaged time series based on 63 data and averaged spatially over all points within the continent.

| Statistics | Bias | $R$ | $RMSE$ | std in Ct | std in GOSAT |
|---|---|---|---|---|---|
| DJF | 2.11 | 0.86 | 2.63 | 3.04 | 2.39 |
| MAM | 1.25 | 0.90 | 1.82 | 2.94 | 2.28 |
| JJA | -0.15 | 0.68 | 1.65 | 1.68 | 2.24 |
| SON | 0.20 | 0.67 | 1.23 | 1.58 | 1.33 |

spatial variability of the seasonal mean difference during different seasons. It also suggests that the performance of the model becomes weak when vegetation cover fade out during dry seasons.

During JJA the seasonal difference in most Africa's land region ranges from -2 to -6 ppm. The result implies CT2016 simulates lower values than that of GOSAT observation. The minimum seasonal bias of -0.15 ppm indicates that there is a

5    better spatial consistency during this season. During SON and MAM when both the North and Southern Africa have a moderate vegetation cover following their respective summer season, the two datasets don't show a significant variation (i.e., only from -2 to 2 ppm) over most of Africa land mass. However, the Equatorial Africa exhibits the mean difference lower than -2 ppm. This indicates the model tends to overestimate $CO_2$ loss by vegetation absorption. Fig. 10 reveals that CT2016 measurements overestimate $CO_2$ except when there is a strong sink due to vegetation absorption during JJA when CT2016 $CO_2$ values are

10   smaller than GOSAT observation during summer season. The underestimation of observed $XCO_2$ by NOAA CT2016 model





is likely related with the skill of driving ERA-Interim data as noted from previous studies. For example, Mengistu Tsidu (2012) has shown that the ERA-Interim data has a wet bias over Ethiopian highlands. Mengistu Tsidu et al. (2015) have also shown that ERA-Interim precipitable water is higher than measurements from radio-sonde, FTIR and GPS observations. Therefore, such wet bias in the driving ERA-Interim GCM might have forced NOAA CT2016 to generate dense vegetation which serve

as $CO_2$ sink. The mean difference between CT2016 and GOSAT $XCO_2$ seasonal means ranges from -0.15 to 2.11 ppm with a standard deviation within a range of 1.21 to 1.64 ppm. The highest mean difference of $XCO_2$ (2.11 ppm) occurs during DJF and the lowest (-0.15 ppm) occurs during JJA. Table 5 presents the summary of statistical values for spacial mean of each season.

### 3.5   Comparison of mean $XCO_2$ from NOAA CT2016 and OCO-2

The strong El Niño event occurred during 2015-2016 provides an opportunity to compare the performance of CT2016 during strong El Niño events. Because of the decline in terrestrial productivity and enhancement of soil respiration, the concentration of $CO_2$ increases during El Niño events (Jones et al., 2001). In this section we compare mean $XCO_2$ of NOAA CT2016 and NASA's OCO-2 covering the period from January 2015 to December 2016. OCO-2 is the most recent full-time dedicated $CO_2$ measuring satellite with greater spatio-temporal resolution and better accuracy.

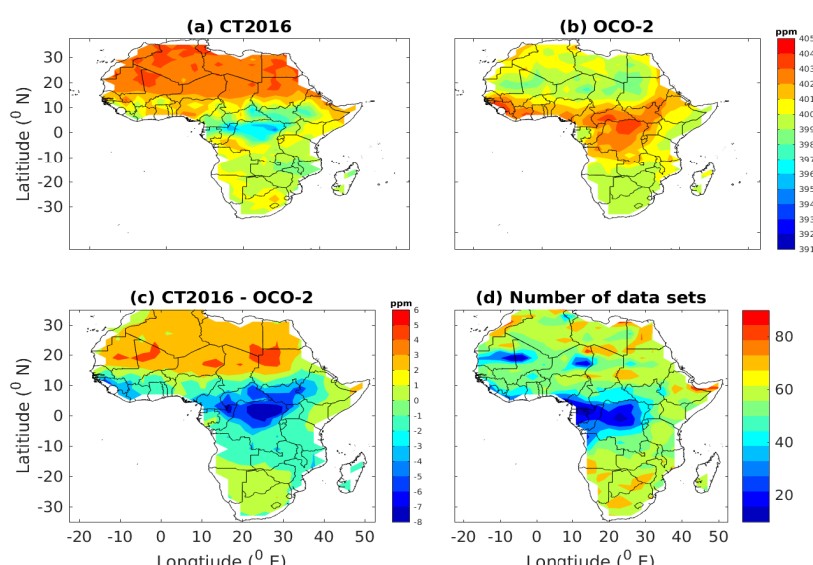

**Figure 11.** Distribution of two years average $XCO_2$ of CT2016 **(a)** and OCO-2 **(b)** $XCO_2$ and their difference **(c)** gridded in $3^0 \times 2^0$ bins; and **(d)** the total number of datasets at each grid

The comparison was done based on the selection criteria discussed in Section 2.4. Fig. 11a-11b show mean distribution of $XCO_2$ from CT2016 and OCO-2 over Africa's land mass respectively. CT2016 shows higher ( > 402 ppm) $XCO_2$ values





over the North Africa while these higher $XCO_2$ values are observed over Equatorial Africa in the case of OCO-2 observation. The two datasets show a discrepancy over Equatorial Africa, where CT2016 simulates lower $XCO_2$ values ($< 398$ ppm) while OCO-2 observes higher values of $XCO_2$ ($> 400$ ppm), especially over Congo basins, South Sudan and western Ethiopia. Both datasets show moderate $XCO_2$ values (which ranges from 398 to 401 ppm ) over Southern Africa. Fig. 11c shows the difference

5   between two years mean of $XCO_2$ from CT2016 and OCO-2, which is in the range from -3 to 4 ppm over most regions of Africa's land mass. Moreover, a negative mean difference over rain forest regions (Gulf of Guinea and Congo basin) and ITCZ zone of Eastern Africa (South Sudan and southwestern Ethiopia) between the two data sets is determined implying that CT2016 underestimates $XCO_2$ values over regions where vegetation uptake is strong. Conversely, a positive mean difference over the Sahara desert, Sahel, eastern Ethiopia, Somalia and the Kalahari desert implies CT2016 overestimates $XCO_2$ values where

10   the vegetation uptake is weak. Higher difference of -9 ppm observed in the rain forest regions of Congo and southwestern Ethiopia is most likely due to small number of data used in the comparison (see Fig. 11d). Overall, the two datasets show a fairly reasonable agreement with a correlation of 0.6 and offset of 0.93 ppm, and a regional precision of 3.77 ppm.

**Table 6.** Summary of statistical relation between CT2016 and OCO-2 observation. The statistical tools shown are the mean correlation coefficient (R), the average of bias, the average root mean square error (RMSE), the standard deviation in bias (std of Bias), mean OCO-2 satellite retrieval error (OCO-2 err), the standard deviation of CT2016 (std of CT2016) and the standard deviation of OCO-2 (std of OCO-2). The number of data used in the statistics is 750.

| Statistical tool | R | Bias | RMSE | std of Bias | OCO-2 err | std of CT2016 | std of OCO-2 |
|---|---|---|---|---|---|---|---|
| Values | 0.60 | 0.93 | 3.77 | 1.95 | 0.54 | 1.86 | 1.16 |

Fig. 12a shows the histogram of two years mean difference, which is characterized by a positive mean of 0.56 ppm and a standard deviation of 2.18 ppm. This suggests that CT2016 overestimates $XCO_2$ as compared to observations from OCO-2.

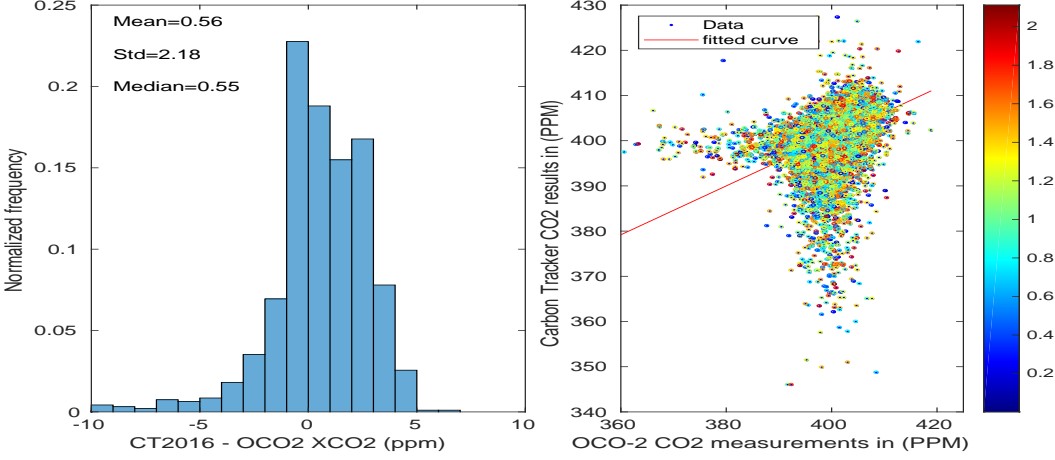

**Figure 12.** Histogram of the difference of CT2016 relative to OCO2 (left panel) and color code scatter diagram of $CO_2$ concentration as derived from CT2016 and OCO-2 (right panel). Color indicates the relative distance as shown in colorbar between datasets.



Because of presence of spatial and temporal mismatch of some level between CT2016 and OCO-2 datasets, it is important to assess the effect of relative distance between the datasets. Fig. 12b shows a color coded distribution of the two datasets. In the figure color codes indicate the relative distance. The random scatter of blue dots implies that the statistical discrepancies do not arise from the relative distance between the two datasets. More specifically, a statistical comparison of datasets below

and above the $50^{th}$ percentile ($1.2^0$) shows bias of 0.71 and 0.73 ppm, correlation of 0.43 and 0.41 and RMSE of 4.22 and 4.23 ppm respectively.

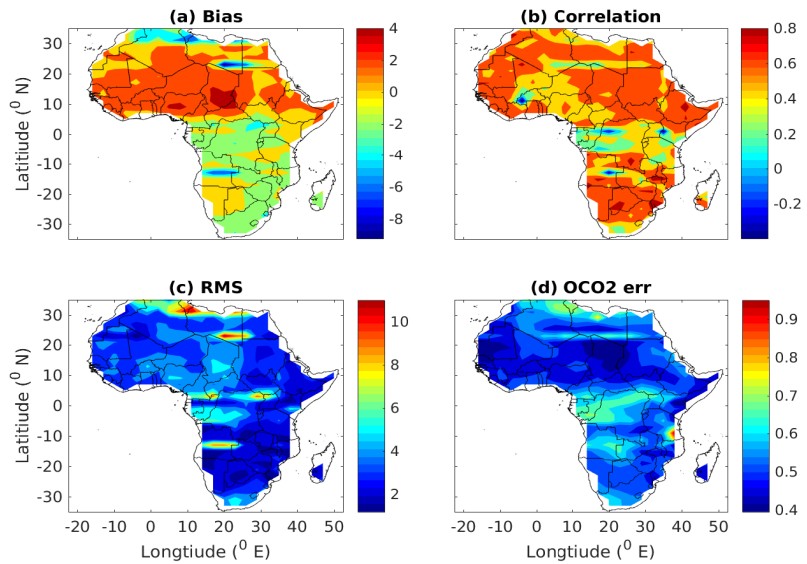

**Figure 13.** The bias (a), correlation (b), RMSE (c) of model and OCO-2 $XCO_2$ and mean total retrieval errors of $XCO_2$ from OCO-2 (d).

Fig. 13 shows the comparison of mean $XCO_2$ from CT2016 and OCO-2 covering the period from January 2015 to December 2016. The number of data used are displayed in Fig. 11d. Fig. 13a depicts the bias which ranges from -4 to 4 ppm with a mean bias of 0.93 ppm. However higher biases of -8 ppm are observed in small pockets over Egypt, Libya, Tunisia and Angola. Fig.

13b shows the correlation map with values from -0.2 to 0.8 over Africa's land mass. A good correlation of above 0.6 are seen over many regions of the continent while weak correlation of less than 0.2 and higher root mean square error ($> 8$ ppm ) are observed over Congo basin and Burkina Faso (see Fig. 13c). These regions also show a higher ($> 0.7$ ppm) error in satellite retrieval (see Fig. 13d). In addition, Fig. 11d shows the number of observations are small ($< 20$ ) over these regions. This may contribute to the observed discrepancy over these regions. However, weak correlations are also observed in many regions

of North Africa such as Burkina Faso, Tunisia, Mali, some regions of Niger and Algeria where satellite errors are low and sufficient data are obtained. This indicates the necessity of incorporating more measurement in CT assimilation over North Africa in order to tune the assimilation model such that it captures the sub-regional carbon cycles.





### 3.6 Categorical comparison of $XCO_2$ from NOAA CT2016 and OCO-2

Applying similar methodology as discussed in Sections 2.4 and 3.2, categorical comparison between CT2016 and OCO-2 datasets is made. Fig. 14 depicts the Bias, POD, CSI, FAR and Categorical Miss between CT2016 and OCO-2 at different thresholds. The analysis was made for the period covering from January 2015 to December 2016. The maps clearly show that

Bias, FAR and Categorical miss increase with increasing threshold. On the other hand, POD and CSI decrease with increasing thresholds (see also Fig. 15).

In our analysis threshold is determined based on OCO-2 observation. The value of $XCO_2$ at $90^{th}$ percentile is 403.7 ppm and most OCO-2 $XCO_2$ value are below 403 ppm over North Africa and Southern Africa. Therefore, categorical analysis shows white space over these regions for thresholds above $90^{th}$ percentile. Fig. 14 also shows that CT2016 simulates the observation

at lower quantiles. However, the performance of CT2016 is observed to be weak in simulating observations at higher quantiles. At $5^{th}$ quantile the POD and CSI values are less than 0.6 over Equatorial Africa which implies the performance of CT2016 is below 60% over the Equatorial Africa.

### 3.7 Comparison of monthly average time series of NOAA CT2016 and OCO-2 $XCO_2$

Figs. 16 - 18 show a two year (from January 2015 to December 2016) monthly average time series comparison of $XCO_2$ from

CT2016 and OCO-2 over North Africa, Equatorial Africa and Southern Africa respectively. Fig. 16a shows the existence of good agreement between the two datasets in describing pattern over North Africa. Moreover, both datasets show a decreasing trend of $XCO_2$ from May to September while increasing trend from October to April. On the other hand, consistent with the climate condition and associated $CO_2$ exchange, the monthly mean $CO_2$ shows a maximum value of 405.36 ppm in April for CT2016 and 401.88 ppm in May for OCO-2. Conversely, a minimum concentration of 397.13 ppm from CT2016 simulation

and 398.11 ppm from OCO-2 observation are found in September. In addition, both CT2016 and OCO-2 show maximum $XCO_2$ values (405.81 ppm for CT2016 and 402.27 ppm for OCO-2) in December. These pick values in December are not surprising, because the 2015-2016 El Niño started on March 2015 and reached pick in December 2015 which added extra $CO_2$ into the atmosphere (Chatterjee et al., 2017). Fig. 16a also shows that $XCO_2$ from CT2016 simulation are higher than OCO-2 observation over North Africa.

Fig. 16b shows the monthly difference between CT2016 and OCO-2 which ranges from 1 to 5 ppm with an exception in August and September. As a consequence of strong El Niño the region misses the short rain season during spring (MAM) when vegetations are still under the influence of the dormancy of winter (DJF). However, following the summer vegetation awakens from the long dormancy and becomes a sink of $CO_2$. In August and September the difference between the two datasets is negative; this implies that CT2016 underestimates $XCO_2$ when the vegetation uptake is strong. On the other hand,

a maximum difference of above 4 ppm was observed during DJF and MAM, implying CT2016 overestimates when vegetation uptake is weaker following the strong El Niño over North Africa. Moreover, Fig. 16c displays a mean bias of 2.23 ppm with a standard deviation of 1.98 ppm between CT2016 and OCO-2. This implies that CT2016 overestimates $XCO_2$ by 2.23 ppm than OCO-2 over North Africa. A strong anthropogenic emission from Nigeria, Egypt and Algeria together with the establishment





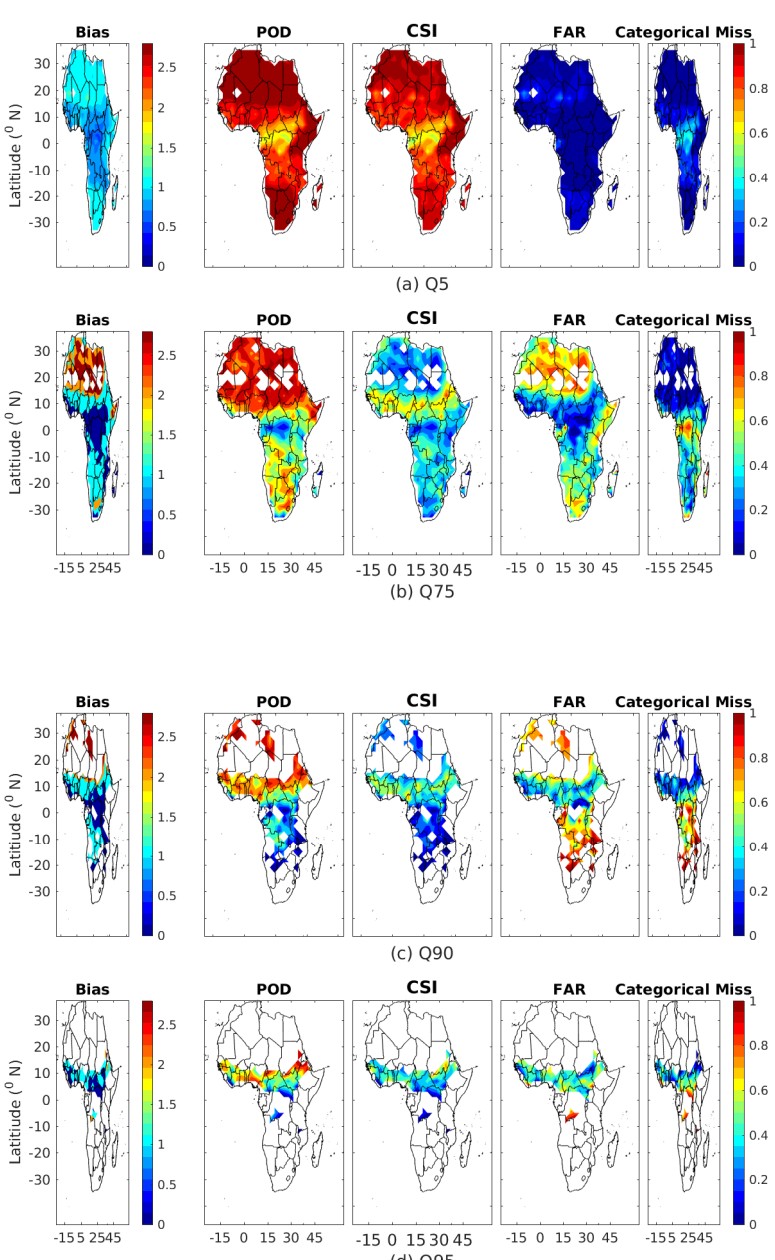

**Figure 14.** Distribution of categorical metrics over the study area for quantiles exceeding 5% (first row), 75% (second row), 90% (third row) and 95% (fourth row).

of plantation over North Africa, which recently exceeded deforestation, and resulted in net flux of carbon sink (Canadell et al., 2009). This might have contributed to the observed discrepancy over North Africa.



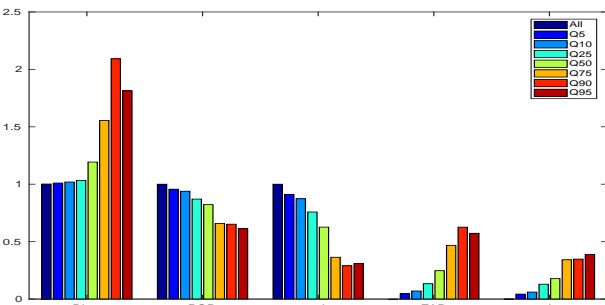

**Figure 15.** Summary of categorical metrics (Bias, POD, CSI, FAR and Categorical miss) averaged over Africa's land region for 0, 5, 10, 25, 50, 75, 90 and 95 percentiles.

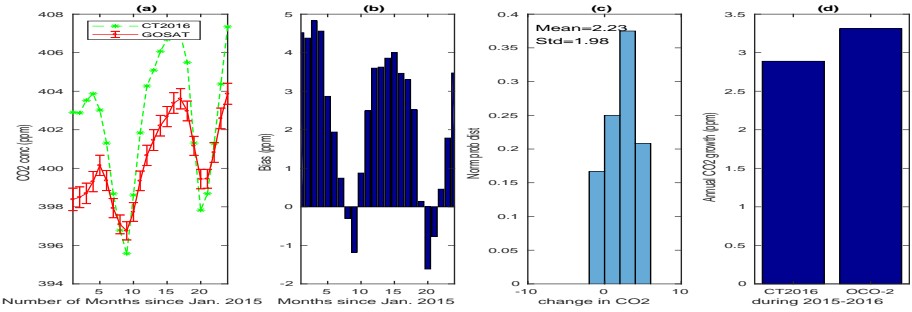

**Figure 16.** The monthly mean time series of CT2016 and OCO-2 from January 2015 to December 2016 averaged over North Africa (a), bias associated to the monthly means (b), the histogram of difference (c) and the annual growth rate obtained by subtracting the mean from the mean of the next year (d).

Figs. 17a - 18a show monthly mean time series of $XCO_2$ from the model and OCO-2 instrument over Equatorial Africa and Southern Africa which are in good agreement in terms of pattern and amplitude. The figures show that CT2016 can capture most of the OCO-2 monthly averages within the bounds of the satellite retrieval error.

Figs. 17b-18b depict a seasonal bias in the monthly time series over Equatorial Africa and Southern Africa. Positive biases are observed during dry seasons while negative biases are during wet seasons. Moreover, the datasets have low biases of -0.41 and 0.15 ppm and standard deviations of 1.3 and 0.53 ppm over Equatorial Africa and Southern Africa respectively. This implies that existence of better agreement between CT2016 and OCO-2 over these regions. Figs. 16d - 18d show both CT2016 and OCO-2 are in good agreement in estimating the annual growth rate. Patra et al. (2017) found a global mean of more than 3 ppm of $CO_2$ added to the atmosphere due to the strong El Niño event that occurred during 2015-2016. In agreement with this, both CT2016 and OCO-2 shows an annual growth rate that ranges from 2.89 to 3.41 ppm of $XCO_2$ over Africa's land mass (see also Table 7).




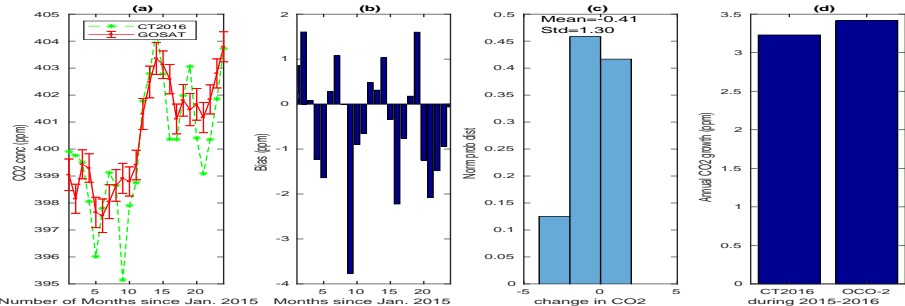

**Figure 17.** The monthly mean time series of CT2016 and OCO-2 from January 2015 to December 2016 averaged over Equatorial Africa (a), bias associated to the monthly means (b), the histogram of difference (c) and the annual growth rate obtained by subtracting the mean from the mean of the next year (d).

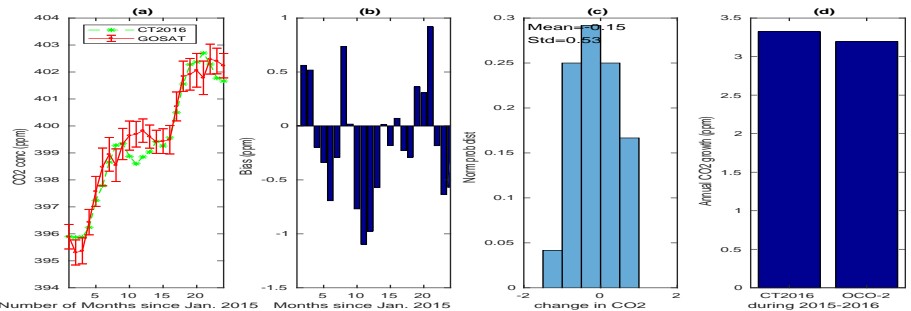

**Figure 18.** The monthly mean time series of CT2016 and OCO-2 from January 2015 to December 2016 averaged over Southern Africa (a), bias associated to the monthly means (b), the histogram of difference (c) and the annual growth rate obtained by subtracting the mean from the mean of the next year (d).

**Table 7.** Annual growth rate (AGR) of $XCO_2$ over Africa land mass from CT2016 and OCO-2. The results are obtained as the mean annual difference of 2015 and 2016 values

| Region | AGR of CT | AGR Of OCO-2 |
|---|---|---|
| North Africa | 2.89 | 3.31 |
| Equatorial Africa | 3.23 | 3.41 |
| Southern Africa | 3.32 | 3.19 |

## 3.8 Comparison of seasonal means of NOAA CT2016 and OCO-2 $XCO_2$

Fig. 19 depicts seasonal means of $XCO_2$ over Africa's land mass from CT2016 (left panels), OCO-2 (middle panels) and their difference (right panels) covering period of January 2015 to December 2016. The white space seen over some regions (e.g., Congo) is because there were no sufficient coincident satellite data according to the selection criteria during these seasons.



Both CT2016 and OCO-2 show maximum of $XCO_2$ during DJF and MAM over North Africa and minimum over Southern Africa. It attains maximum during SON over the Southern Africa.

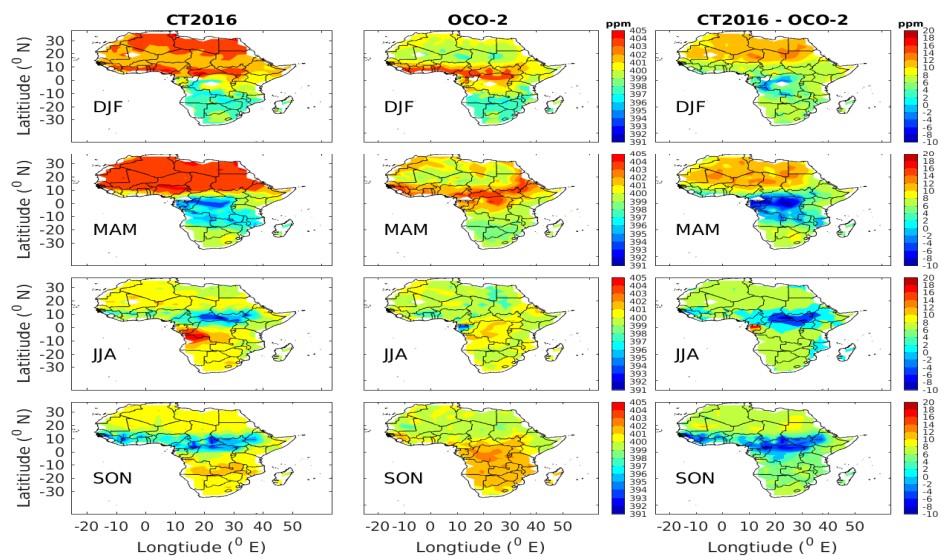

**Figure 19.** Seasonal mean of $CO_2$ for NOAA CT2016 (left panels) and OCO-2 (middle panels) and their difference (right panels).

Fig. 19 (right panels) shows the seasonal mean difference of CT2016 and OCO-2. A higher mean difference greater than 2 ppm observed over North Africa during DJF and MAM when the vegetation cover over the region decreases. This suggest that CT2016 overestimates $XCO_2$ values when vegetation uptake is weak. On the other hand, higher negative mean difference of less than -2 ppm are observed over Equatorial Africa and Eastern Africa during JJA and SON when the vegetation uptake is strong following their long rainy season.

Fig. 20 shows the histogram of seasonal mean difference of CT2016 and OCO-2. The smaller biases of 0.98 and -0.78 ppm and RMSE of 2.97 and 3.57 ppm are observed during JJA and SON respectively. On the other hand, higher biases of 2.16 and 1.01 ppm and RMSE of 3.3 and 4.52 ppm are also observed during DJF and MAM respectively. The results indicate that CT2016 and OCO-2 show a better consistency during wet seasons and this consistency decreases as the vegetation cover decreases over most regions of Africa land mass during dry seasons.

Consistent with report by Liang et al. (2017), very high seasonal variability was observed between CT2016 and GOSAT and low seasonal variability between CT2016 and OCO-2 with greater amplitude over North Africa than over Southern Africa.

## 4   Conclusions

Satellite observations are able to capture the $CO_2$ concentration truly and objectively. In the study area retrieved $CO_2$ data product, however, has average error of about 0.9 ppm for GOSAT and 0.54 ppm for OCO-2 due to the limitations of satellite



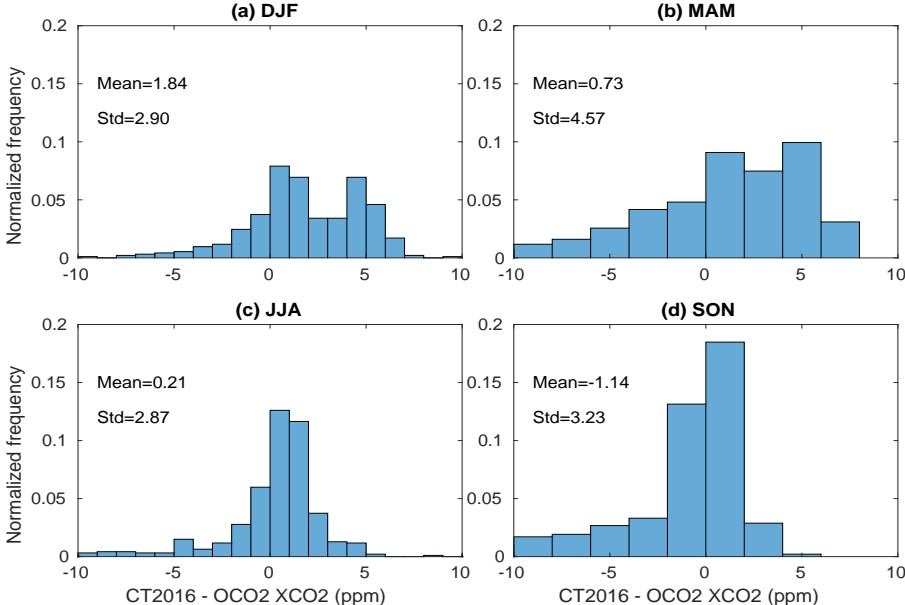

**Figure 20.** Histogram of difference for the seasonal $CO_2$ climatology for DJF (a), MAM(b), JJA (c) and SON (d) seasons.

sensors and retrieval algorithms. On the other hand, atmospheric transport models based on assimilation can provide global atmospheric $CO_2$ data in continuous spatio-temporal resolution.

In this study, the NOAA CT2016 $XCO_2$ values are compared with two full-time $CO_2$ dedicated satellites, GOSAT and OCO-2 over Africa land mass. Comparison between CT2016 and GOSAT were done using a five years datasets covering the period from April 2009 to June 2014. This comparison is important to test the performance of CT2016 in capturing climatology. Comparison of CT2016 with OCO-2 was done using two years data during the strong El Niño event from January 2015 to December 2016. This provide opportunity to asses the performance of CT2016 simulation during strong El Niño events. Comparison of CT2016 with the two satellites reveals biases of 0.44 and 0.98 ppm, correlations of 0.73 and 0.6 and an RMSE of 3.49 and 3.77 withe respect to GOSAT and OCO-2 respectively. The performance of the model in capturing the whole distribution is also assessed. It is found that, CT2016 can capture more than 88% of the observation at lower thresholds. However, the probability of detection and the critical success index decrease with increasing threshold implying CT2016 shows some weakness in capturing extreme values exceeding $95^{th}$ percentile.

The monthly average time series of CT2016 over North Africa, Equatorial Africa and Southern Africa separately compared with $XCO_2$ from the two satellites. CT2016 agrees well with measurements from the two instruments in terms of pattern and amplitude over Equatorial and Southern Africa. However, this agreement deteriorates over North Africa. It is also found that there is a seasonal dependent bias in between them which is positive during dry seasons while it is negative during wet seasons. High spacial mean biases of 2.11 and 1.84 ppm during DJF and low biases of -0.15 and 0.21 ppm during summer (JJA) in the model $XCO_2$ with respect to $XCO_2$ from GOSAT and OCO-2 respectively. CT2016 has the ability to capture monthly time





serious and seasonal cycles. However, CT2016 overestimates during dry seasons. In addition to this, CT2016 simulates lower $XCO_2$ than the observations over some regions (e.g., Congo and South Sudan and southwestern Ethiopia) and during summer season over the whole continent following larger vegetation uptake.

In general, $XCO_2$ from NOAA CT2016 show a very small bias with respect to GOSAT and OCO-2 observation over Africa's
5  land mass. Moreover, there is a good agreement between CT2016 simulation and observations in terms spatial distribution, monthly average time series and seasonal climatology. However, there are some discrepancies between the model and the two $XCO_2$ datasets from GOSAT and OCO-2 implying that the accuracy of the model data needs further improvements for the rain forest regions (e.g., Congo) through assimilation of in-situ observations and tuning the model through process studies.

*Acknowledgements.* The authors acknowledge NOAA Earth System Research Laboratory and NASA ACOS GOSAT for the data products.
10  Addis Ababa University, Addis Ababa Science and Technology University, Botswana International University of Science and Technology for their support through fellowship and access to the research facilities..



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
