# Peer review of "Comparison of $CO_2$ from NOAA Carbon Tracker reanalysis model and satellites over Africa"

_Atmospheric Measurement Techniques, 2018_

## Referee Comment (RC1) · Anonymous Referee #1 · 15 Jun 2018

The authors evaluated two versions of the CarbonTracker inversion system by comparing its posterior simulation with satellite $XCO_2$ retrievals over the African continent. This regional focus is interesting but the paper is loosely written and often reads like a technical report. Throughout the text, there are many vague or awkward expressions that induce misleading or erroneous statements. There are both some small repetitions and some out of scope paragraphs (e.g., the parts about growth rate). The content of the cited papers does not always correspond to what is said of them in the citing paper. Retrievals are used and cited without any reference to specific versions while they may

be quite different from one to the next (e.g., are the authors using v7 or v8 OCO-2 retrievals from NASA?). I argue that much effort is needed to check the content of each sentence and improve the general presentation, before the paper can be published.

I am therefore skipping the numerous details in order to concentrate here on general issues.

The introduction (p.2-3) is unnecessarily long and convoluted, and it poorly motivates the study. For instance, the paragraph about TCCON is disconnected from the rest. More importantly, there is a silent shift of meaning for "model" from "Earth system model" to "transport model" without any concern about consistency in the logic flow. Some statistics are given without any reference to specific retrieval versions.

The method (p. 5) does not mention the retrieval averaging kernels. The authors need to clearly state the fact that they have used them, or redo their study if they have not used them yet.

Statistical quantities are not mathematically defined and there are many in this paper. For instance, the False Alarm Ratio (FAR) is simply defined by "identifies the fraction of events captured by simulation but not available in reference observation", but there is no obvious "event" for $XCO_2$ (in contrast to rain for instance). An equation would explain what the authors mean, but at first glance FAR seems ill-suited for a continuous variable. From the values they find, the authors seem to be concerned by the fact that the model does not capture the higher end of the retrieval distribution, but what does it mean? The model could, e.g., slightly misplace fire plumes in time or space without affecting its overall realism. A simple scatter plot could efficiently replace the series of categorical indices that the authors use and simplify the message. Also about statistics, it is not even clear whether they are computed over 5 years, or over 5 years and 3 months. Last, some error bars are put on monthly-mean and area-mean retrieval values in the figures. We may suppose that they are standard deviations (the legend does not mention them), but they seem to be way to large for that (only random errors

are given in satellite products and by default they should decrease as 1/sqrt(n)).

Table 1 is very intriguing. First, the focus area is made of 428 grid points and the data spans about 5 years, but the statistics are made over 750 data points only. Second, if we assume that GOSAT errors and CT errors are uncorrelated, we deduce a 1-$\sigma$ error for CT $XCO_2$ over Africa of $(3.47^2\text{-}0.9^2)^{0.5}$ = 3.4 ppm, marginally smaller than the variability of the retrieved $XCO_2$ ($\sigma$ = 4.3 ppm). Such a poor skill is hard to believe. By comparison, I downloaded the CAMSv17r1 data (http://apps.ecmwf.int/datasets/data/cams-ghg-inversions/) and compared it with the ACOS GOSAT v3.5 retrievals for the same period (5 years and 3 months) with the proper averaging kernels. For individual values, I find a model-minus-retrieval bias of -0.41 ppm (similar to what is shown in the paper if we except the sign, that is undefined in the table legend) and a standard deviation of 2.2 ppm, for a number of data points of 266,662. That makes a model uncertainty of $(2.2^2\text{-}0.9^2)^{0.5}$ = 2 ppm, ie 35% less than for CT. If we account for the fact that the estimated retrieval precision may be wrong by a factor of ˜1.5 (see O' Dell et al 2012 for a previous ACOS release, http://dx.doi.org/10.5194/amt-5-99-2012), we find a model random uncertainty a bit better than the retrievals, as expected. The correlation also rises from 0.73 in the paper to 0.87. CAMS and CT are different products, but we do not expect such a difference in quality.

Talking about CAMS, there was a comparison between MACC13r1 and ACOS GOSAT v3.5 a few years ago with some focus over Africa savannahs, that suggested deficiencies in the retrievals (in terms of systemetic errors and in terms of averaging kernel shape) (Chevallier et al. 2015, http://dx.doi.org/10.5194/acp-15-11133-2015). If the authors use a more recent version, these artifacts may have disappeared, but this needs to be looked at. If the authors use the same version, this needs to be accounted for when using the retrievals as a reference.

---

## Short Comment (SC1) · 2 Aug 2018

D. Baker

dfbaker66@gmail.com

In this manuscript, $CO_2$ mixing ratio fields over the continent of Africa taken from the CarbonTracker-2016 (CT2016) $CO_2$ flux inversion system are compared to $CO_2$ measurements obtained from the GOSAT and OCO-2 satellites. Though it is not said explicitly, this $CO_2$ comparison is presumably done using column averages (commonly refered to as "$X_{CO2}$"), either using a straight pressure weighting or by sampling the CT model using the same vertical averaging kernel and prior $CO_2$ vertical profile that the satellite retrieval uses on the true atmosphere. The CT2016 system takes a set

of somewhat-realistic surface $CO_2$ fluxes (fossil fuel burning, land use change, wildfires, photosynthesis and respiration from the land biosphere, air-sea fluxes) as a first guess, runs them forward through the TM5 off-line atmospheric tracer transport model to obtain $CO_2$ mixing ratios in the interior of the atmosphere, samples these at the time and place of surface in situ data, and uses the measurement differences to improve the initial flux estimates using a fixed-lag ensemble Kalman smoother as the inversion method. CT2016 thus can be thought of as a model of $CO_2$ in the atmosphere that has been forced to agree with surface $CO_2$ measurements, or alternatively as a glorified interpolation between the in situ measurements. The difference between CT2016 and the satellite data is mainly viewed here as a deficiency in the CT model as compared to the more accurate satellite data. Explanations for the differences are suggested in terms of likely flux errors in the CT2016 model.

Comparisons of this sort are of value, but in my view are better viewed from a different perspective. First, if one is interested in understanding the processes driving $CO_2$ fluxes in Africa (fires, photosynthesis, respiration from plants or soils, etc), one might do better to look at the results of an inversion of the satellite $CO_2$ data compared to the fluxes obtained from a system like CT2016 that inverts in situ $CO_2$ data. The reason for this is that the column averages compared here have much information in the upper part of the column that blows in from outside the bounds of Africa and reflect the effect of fluxes from around the globe; thus, column $CO_2$ differences compared over Africa may not reflect the impact of local fluxes, but also of far-field fluxes that have blown in over Africa. The inversion systems, which model atmospheric transport, are supposed to sort this out and apportion the fluxes to the right locations – they are the tool that I would use if understanding fluxes is the goal.

The direct comparisons of column $CO_2$ do have value, but one should not assume that the model is wrong and the satellite measurements right, as has been done here for the most part. The retrieval of $CO_2$ from satellite data is beset with a variety of challenges (scattering due to clouds and aerosols, instrument issues, unknown spectroscopy, surface characterization issues, etc.) that often result in systematic errors that, at the moment, are usually removed in a separate step after the retrieval. A key component of this bias correction process is the comparison to independent data, such as column-averaged $CO_2$ measurements from the TCCON network, for example (using comparisons similar to those in this manuscript, but with the model being replaced by the TCCON data). However, comparison to models that have assimilated in situ data (like CT2016) have also been used in this bias correction step. It has been found that the systematic errors in the satellite retrievals are generally larger than those from an ensemble of models, so that the models may be used to help find errors in the satellite retrievals (instead of vice versa, as is done in this manuscript). Hopefully that situation will change as satellite retrieval schemes improve, but at the moment it is not a good assumption that most of the difference between model and satellite measurements can be attributed to model error. Really, there are errors in both model and measurement and one should quantify both with appropriate uncertainties. The authors do provide satellite $CO_2$ retrieval uncertainties here, but these do not capture the impact of systematic errors in the retrievals and are overly optimistic – one cannot assume that because they are small, the majority of the model-measurement difference must be due to model error.

In my view, the model-measurement comparisons done here are most valuable for understanding errors in the satellite retrievals. Africa is a continent with wide extremes in surface type (desert vs. rainforest vs savannah) and aerosol loading. These conditions have a strong impact on the satellite retrievals, so a study of this sort over Africa can tell much about how these systematic errors vary geographically. I would encourage the authors to consider whether their comparisons might be better viewed through that lens.

In general, the English punctuation and usage in this manuscript need extensive editing, which I have not attempted to do here. The authors need to clarify certain aspects of their method, most notably what $CO_2$ quantity they are plotting on their figures (presumably column-averaged $CO_2$) and how they do their vertical averaging. Measurement errors of 2 to 5 ppm are not small or reasonable – they are show-stoppers that prevent the satellite data from being useful for inferring surface $CO_2$ fluxes (since these measurement errors feed through to similar errors in the flux results). Some of the statistical metrics presented here are unconventional and are difficult to understand, and do not add much to the presentation – I suggest below that those sections of the document be removed.

Detailed comments follow below:

page 1, Line 1: "poor global coverage and resolution of satellite observations": Usually the satellites are thought to have good coverage over the globe, at least when compared to the in situ data.

Abstract: Here, the fidelity of the CarbonTracker model is being tested by comparing its a posteriori 3-D CO2 fields to $CO_2$ measured from satellites and by the TCCON network of ground-based sun-viewing spectrometers. That would make sense if the accuracy of those measurements was thought to be better than that of the model. But what if the reverse was true? Then it would make more sense to check the accuracy of the measurements by comparing them to the (more accurate) model. In the early days of satellite $CO_2$ data (and we are still in the early days, really) it was thought that that latter situation was actually the case, and models were used to correct the satellite data. Given that, some discussion of the assumptions underlying the comparison of CarbonTracker to the satellite and TCCON measurements would be useful in this paper before moving on to the comparison.

p1 L15: "relative accuracies of 1.22 and 1.95 ppm were found between the model and the two data sets"; these were judged to be "reasonably good". The in situ $CO_2$ data have relative accuracies an order of magnitude better than this (0.1 to 0.2 ppm) – why then should measurements an order of magnitude less accurate be considered "good"? Some guidelines concerning what is considered a good error versus what is considered

a bad error ought to be given to support this assertion. The density and coverage of a data type might also factor into this assessment, as well as the importance of random versus systematic errors.

p1 L17-18: "...probability of detection ranges from 0.6 to 1 and critical success index ranges from 0.4 to 1..." These statistics may not be familiar to the general reader. Maybe say "(see main text for definitions)" when using these in the abstract?

p1 L20-21: "GOSAT and OCO-2 $X_{CO2}$ are lower than that of CT2016 by upto 4 ppm over North Africa (10 N −35 N) whereas it exceeds CT2016 $X_{CO2}$ by 3 ppm over Equatorial Africa (10 S−10 N)."

It seems like the sign of this is wrong. Also, these satellite data are raw or bias-corrected?

p1 L25: "In these cases, the model overestimates the local emissions and underestimate $CO_2$ loss." Here, it is assumed that the satellite data are correct and the model is wrong. But what if the reverse was actually true? In reality, both model and satellite are wrong, to differing degrees that are quantified using the metric of uncertainty. You should give reasons why you think the uncertainty on the satellite measurements is lower than the uncertainty on the modeled $CO_2$, to defend why you think the model is wrong and the satellites right. The satellites have retrieval biases and random errors that could easily be larger than the modeling errors in $CO_2$ (errors due to transport, errors in the in situ data, inversion assumptions, etc).

p2 L25-28: In introducing the TCCON data, you should note that the TCCON network measures a column-integrated $CO_2$ mixing ratio; yes, it is "ground-based", since the sensor sits on the ground, but the measurement is a column average (unlike the ground-based in situ data) and that is why it is useful for validating the satellite data, which are also column averages.

p3 L3: "Other studies have revealed that significant improvement in estimation of

weekly and monthly $CO_2$ fluxes can be achieved subject to $CO_2$ retrieval error of less than 4 ppm" This may be true if the errors are purely random – with enough low-precision data of this sort, the errors may average down to something lower and more useful. However, if this is accompanied by systematic errors of similar magnitude, these will not average out. Therefore some discussion of random versus systematic error must be added when giving out these error quantities.

p3 L21: "V02.XX" – you need to complete the version numbers for two products on this line

p4 L3-4: "These findings suggest that it is important to assess the accuracy and uncertainty of $X_{CO2}$ from models with respect to observations" Again, the authors point to the differences between satellite data and models, then leap to the assumption that the models are responsible for most of those differences when the measurements might actually be more responsible. Some discussion of which is more error-prone (model or measurement) is needed.

p4 L27: "CT2016 and CT2017 respectively": you should not refer to the 2016 near-real-time CT as "CT2017", as that term is reserved for the subsequent release of the full data span (the release that would have been put out in 2017 if the releases were put out when indicated; the release that finishes at the end of 2016). The near-real-time releases are different from the standard releases in that they bring out a subset of the full data set, without the usual quality assurances, for early use by modelers. Even if that is only a label for use in this manuscript, it will confuse people.

p5 L1: Which version of ACOS GOSAT retrievals did you use in this study?

p5 L2-3: "GOSAT is the world's first spacecraft to measure the concentrations of carbon dioxide and methane, the two major greenhouse gases, from space." Not true: many satellites/instruments measured both in the thermal IR. Also SCIAMACHY, which came out much earlier, measured both in the near-IR. What you could say is that GOSAT was the first spacecraft dedicated solely to measuring $CO_2$ and $CH_4$.

p5 L15: When discussing the OCO-2 data, you must say what version of retrieval you are using: version 7? Also, you should indicate whether you are looking at raw $X_{CO2}$ values or bias-corrected values. If bias corrected, you should indicate whether you used the additional "s31" bias correction, designed to correct albedo-related errors not caught in the initial bias correction. This extra bias correction could be particularly important over Arica, with its large differences between desert and tropical rainforest. Also, you should discuss whether you sample your model with the OCO-2 averaging kernel and prior $CO_2$ profile when comparing to the OCO-2 retrievals.

p5 L21: (Methods): the satellites estimate both a profile of $CO_2$ on 20 levels, as well as a vertical weighted average, "$X_{CO2}$", computed from these. Which do you compare to? If the latter, you should discuss the vertical weighting kernel for $X_{CO2}$ and how you sample your model to account for this. This methods section seems to be the correct place to discuss this.

p6 L19: "Fig. 1 shows the five-years average of CT2016 (Fig. 1a) and GOSAT (Fig. 1b) $X_{CO2}$ distribution." It would have been more useful to do the 5-year average over a true 5-year span, rather than the "April 2009 to June 2014" span that you used. As it stands, you have an extra 3 month span (April to June) that is unbalanced by the remainder of an annual cycle. As a result, this three month span throws off what otherwise would be the average of five full annual cycles. If flux is positive north of the equator during April to June (which it probably is, this being right at the end of the burn season there) then you tend to see positive values there in Figure 1a and 1b. This is due to the seasonality of the flux rather than the annual mean value. It might be easier to discuss annual mean fluxes in the text if these plots reflected averages over pure annual cycles.

p7 L3: "respiration from forest in the region is overestimated" Why do you think that this is not due to photosynthesis being underestimated?

p7 last line: "standard deviation of 1.31 ppm indicating better consistency and less potential outliers." Better than what?

p8 L3: change "require" to "permit" Also, if the difference can be up to 1.5 degrees on either side, then we are talking about a box of 3 degrees on a side in both latitude and longitude. On p5 L29 you should then change the box from being 1.5 degrees long and wide to 3 degrees long and wide.

Fig 3 caption: please provide units (ppm)

p8 L19: A correlation should be unitless – remove "ppm"

Figure 3: It is not clear to me why Figures 3a and 1c are different – they are both showing the difference (or bias) between CT2016 and GOSAT on an annual mean basis. So why aren't they identical?

p8 L20: Good – here you are examining whether some of the difference might be due to the satellite data rather than the model.

Section 3.2: The statistics presented here are abstruse and require the reader to go back to the other references not only for what the statistics mean, but even what the acronyms stand for. It is not clear what the message is that these statistics convey. The only point I gleaned from the discussion was that the annual mean bias between model and data is higher where there are fewer data to average over – that is a point that could be made with Figure 1 alone. I would suggest deleting this section altogether.

p13 L9, p14 L11-14: Here you are suggesting that local fluxes are responsible for the changes in $CO_2$ seen over Africa, but it is quite possible that a large part of these changes are due to fluxes elsewhere, over different continents, blowing downwind over Africa. Without inversion results linking concentrations to the fluxes that caused them, your argument will remain weak using this approach.

p25 L16: "Satellite observations are able to capture the $CO_2$ concentration truly and objectively." I don't understand how you can say this based on the results that you have shown. You have shown large differences between CarbonTracker and the satellite data, on the order of 5 ppm on a seasonal and regional basis. This could be due

to satellite retrieval problems in addition to the model problems that have been empha-sized here. You haven't shown any data or comparisons that suggest the satellite data are better than the model, really. Maybe you could compare GOSAT and OCO-2 across the short time period that they overlap (late 2014, 2015) and if they agree closely then suggest they are both getting the right answer. From the plots you have showed in this manuscript, however, it looks like there are significant differences between OCO-2 and GOSAT... suggesting that the satellite errors are significant.

---

## Author Comment (AC1) · 23 Aug 2018

Authors' response to interactive comment on "Comparison of CO2 from NOAA Carbon Tracker reanalysis model and satellites over Africa" by Anonymous reviewer

We thank the anonymous reviewer for the valuable comments and time spent to provide the important comments and suggestions. They are enormously constructive and are used to improve the quality of the manuscript significantly. We will respond to all the comments in details as follows.

[Figure]

Reviewer's comments: The authors evaluated two versions of the Carbon Tracker inversion system by comparing its posterior simulation with satellite XCO2 retrievals over the African continent. This regional focus is interesting but the paper is loosely written and often reads like a technical report. Throughout the text, there are many vague or awkward expressions that induce misleading or erroneous statements. There are both some small repetitions and some out of scope paragraphs (e.g., the parts about growth rate).

Response We thank the reviewer for realizing the importance of this work, in particular, the focus on regional aspect. The revised manuscript have been edited substantially following the reviewer's comment. However, it is difficult to pin-pointedly indicate where these changes are made since we edited the whole manuscript because of the fact that the reviewer made generic comments to improve the overall manuscript quality.

The reviewer indicated presence of many vague expressions. We hope that some of those might have been already taken care of in the process of substantial editing as indicated in the previous paragraph. In the case of the content on the growth rate, for instance, we would like to emphasis that it is important as it shows the degree of agreement between the model simulation and satellite observations at interannual time scale. In other words, one can also argue assessing whether model prediction of growth rate is comparable to that captured by satellite retrievals is important. This means that growth rate in our context is meant for comparison purpose rather than understanding dynamics/processes that led to a specific growth rate. Therefore, we believe that the paragraphs on growth rate is within the scope of the paper which is to assess how CT model simulation compares with satellite observations.

Reviewer's comment: The content of the cited papers does not always correspond to what is said of them in the citing paper.

Response We have checked all our references and determined whether they are properly cited for the conclusions/results included in the cited papers. We thank the reviewer

for pointing out cases of inappropriate citations which are now removed. For example, on page 2, line 15, Deng et al. 2016 was cited as if it shows historical changes in XCO2 from pre-industrial period. This was wrong and it has been now replaced with two new references namely Tsutsumi et al. (2009) and Allison (2015). There was also few other cases that are also corrected.

Reviewer's comment: Retrievals are used and cited without any reference to specific versions while they may be quite different from one to the next (e.g., are the authors using v7 or v8 OCO-2 retrievals from NASA?). I argue that much effort is needed to check the content of each sentence and improve the general presentation, before the paper can be published. I am therefore skipping the numerous details in order to concentrate here on general issues

Response We thank the reviewer for pointing luck of specific information on the data version used in this study to us. We apologize for the oversight. Now, changes have been made in this particular case. Specifically, we want to indicate that the version of OCO-2 used is OCO-2 V7 lite level 2 products. This version of OCO-2 was compared to Carbon Tracker release CT2016 for 2015 and version CT-NRT.v2017 for 2016. Moreover, the version of GOSAT XCO2 used in the manuscript is ACOS B3.5 Lite. This information is incorporated in Section 2.2 and 2.3 of the revised manuscript.

Reviewer's comment: The introduction (p.2-3) is unnecessarily long and convoluted, and it poorly motivates the study. For instance, the paragraph about TCCON is disconnected from the rest. More importantly, there is a silent shift of meaning for "model" from "Earth system model" to "transport model" without any concern about consistency in the logic flow.

Response We have tried to shorten and add clarity to the two paragraphs mentioned by the reviewer's. Although the statement on the necessity of models to compliment observations is equally good for both kind of models namely complex 'Earth system model' and simple 'transport model' in the context of paragraph 2 (line 20-22), we

rephrased it such that it now refers to transport model to avoid similar misunderstanding by general reader. We also amended paragraph 3 to improve the flow of ideas reflected in it.

Reviewer's comment: Some statistics are given without any reference to specific retrieval versions. The method (p. 5) does not mention the retrieval averaging kernels. The authors need to clearly state the fact that they have used them, or redo their study if they have not used them yet.

Response We appreciate the reviewer for reminding us about the averaging kernel. In fact, we have ignored the difference in the resolution between the model and satellite XCO2 observations in the study after simple inspection of model and satellite vertical grids. As a result we did not use averaging kernel to smooth CT XCO2. We have now smoothed CT XCO2 as per procedure described by Rodgers and Connor (2003) and then compared the smoothed XCO2 with the observations from satellites. This has led to significant changes improving the agreement between the model simulations and satellite observations leading to lower RMSD, bias, high correlation as well improvement in the categorical statistics. Therefore, the analysis in the manuscript is updated (see the revised manuscript). We would like to thank the reviewer for the valuable inputs. Now, Section 2.4 in the revised manuscript includes the procedure on how to smooth the high resolution data to low resolution data using averaging kernel of low resolution data, a priori profile and the weighting functions.

Reviewer's comment: Statistical quantities are not mathematically defined and there are many in this paper. For instance, the False Alarm Ratio (FAR) is simply defined by "identifies the fraction of events captured by simulation but not available in reference observation", but there is no obvious "event" for XCO2 (in contrast to rain for instance). An equation would explain what the authors mean, but at first glance FAR seems ill-suited for a continuous variable. From the values they find, the authors seem to be concerned by the fact that the model does not capture the higher end of the retrieval distribution, but what does it mean? The model could, e.g., slightly misplace fire

plumes in time or space without affecting its overall realism. A simple scatter plot could efficiently replace the series of categorical indices that the authors use and simplify the message.

Response We have included equations that describes some of the known statistics such as RMSD, bias and pattern correlation including the categorical metrics although we felt that doing so does not add much value as there are ample references with details on these statistics cited in this manuscript. However, the reviewer has also questioned the importance and relevance of the categorical metrics. As correctly pointed by the reviewer, the scatter plots can provide qualitative information on the discrepancy between CT and satellites XCO2. However, we want to go beyond qualitative analysis such that we have some kind of metrics to assess the level of agreement between model and satellite observations in the extreme ends of XCO2 distribution. Indeed, it is true that XCO2 is a continuous variable whereas the categorical metrics apply to categorical variables. However, when we use quantiles as a threshold to determine whether the model predicts accurately the observed XCO2 lies above the given quantile threshold or not, we are not looking at the actual value. Instead, we are investigating whether the model and observations are in good agreement in indicating the correct category of the observed and model XCO2 i.e., higher or lower the quantile threshold. Details of the calculations are given in Section 2.4 of the revised manuscript. In this regard, we have also made some changes in the discussion of results to improve clarity of the manuscript.

Reviewer's comment: Also about statistics, it is not even clear whether they are computed over 5 years, or over 5 years and 3 months. Last, some error bars are put on monthly-mean and area-mean retrieval values in the figures. We may suppose that they are standard deviations (the legend does not mention them), but they seem to be way to large for that (only random errors are given in satellite products and by default they should decrease as 1/sqrt(n)).

Response We used 5 years and 3 months data in computing the statistics in the old

manuscript, while comparing ACOS GOSAT B3.5 and CT2016 covering the period from April 2009 to June 2014. However, comparison on the seasonal basis may affect the statistics between seasons due to unequal number of data points within seasons due to the extra three months as correctly indicated by Dr. Baker, the second reviewer. Therefore, we have used only 5 years in the revised manuscript. The error bars over the spatial mean and monthly mean are the corresponding spatial mean and monthly mean $XCO_2$ posterior error of satellite which is a combination of instrument noise, smoothing error and interference errors (Connor et al., 2008; O'Dell et al., 2012). Their value ranges from 0.74 to 0.88 ppm. Since the error bar does not include the complete systematic errors that arise, for example, due to forward radiative model error and Line of sight (pointing) error, our estimate is rather conservative. According to O'Dell et al., 2012 , the true error could be as large as twice the posteriori. Therefore, we have highlighted this drawback of the error bar used in this manuscript in Section 3.1 on page 10, lines 2-3.

Reviewer's comment: Table 1 is very intriguing. First, the focus area is made of 428 grid points and the data spans about 5 years, but the statistics are made over 750 data points only. Second, if we assume that GOSAT errors and CT errors are un-correlated, we deduce a 1-$\sigma$ error for CT XCO 2 over Africa of (3.47 2 -0.9 2) 0.5 = 3.4 ppm, marginally smaller than the variability of the retrieved XCO2 ($\sigma$ = 4.3 ppm). Such a poor skill is hard to believe. By comparison, I downloaded the CAMSv17r1 data (http://apps.ecmwf.int/datasets/data/cams-ghg-inversions/) and compared it with the ACOS GOSAT v3.5 retrievals for the same period (5 years and 3 months) with the proper averaging kernels. For individual values, I find a model-minus-retrieval bias of -0.41 ppm (similar to what is shown in the paper if we except the sign, that is undefined in the table legend) and a standard deviation of 2.2 ppm, for a number of data points of 266,662. That makes a model uncertainty of (2.2 2 -0.9 2 ) 0.5 = 2 ppm, ie 35% less than for CT. If we account for the fact that the estimated retrieval precision may be wrong by a factor of ÌČ1.5 (see O' Dell et al 2012 for a previous ACOS release, http://dx.doi.org/10.5194/amt-5-99-2012), we find a model random uncertainty a bit

better than the retrievals, as expected. The correlation also rises from 0.73 in the paper to 0.87. CAMS and CT are different products, but we do not expect such a difference in quality. Talking about CAMS, there was a comparison between MACC13r1 and ACOS GOSAT v3.5 a few years ago with some focus over Africa savannahs, that suggested deficiencies in the retrievals (in terms of systematic errors and in terms of averaging kernel shape) (Chevallier et al. 2015, http://dx.doi.org/10.5194/acp-15-11133-2015). If the authors use a more recent version, these artifacts may have disappeared, but this needs to be looked at. If the authors use the same version, this needs to be accounted for when using the retrievals as a reference.

Response We apologize for the oversight. The 750 indicated as being used in calculation was wrong. It is the initial number of grids that have coincident satellite observations when data over Africa is extracted from the global data set. However, our analysis focuses only on African landmass which reduces the 750 grids to 428. In addition, we excluded grids with less than 10 observations in the final analysis. This means that the actual analysis was based on 426 grids excluding 2 grids with less than 10 observations. Nevertheless, the actual analysis was not affected by the oversight and was based on the 426 grids in the old manuscript as well as in the revised manuscript. What was wrong is our erroneous reference to 750 and 428 in the different part of the manuscript. The statistics in section 3.1 of the paper were performed based the available time series data that passes the selection criteria as indicated above. The distribution at each pixel was indicated in the paper (Fig. 1d) and it was discussed in the manuscript. The aggregate number of data used in the statistics is 472,821 over 426 pixels covering the study period. We now also included the aggregate data points used in the revised manuscript in Table 1 caption of the revised manuscript.

As indicated in one of the previous responses, initially we have not used averaging kernel of GOSAT $CO_2$ retrieval to smooth CT data. After applying smoothing using retrieval averaging kernel, the results of statistics shows a substantial improvement. The correlation was changed from 0.73 to 0.83 which is improved by 13%, and RMSD

is changed from 3.4 to 2.3 ppm. Implying a model uncertainty of $(2.3^2 – 0.9^2)1/2 = 2.12$ ppm shows a 52% improvement. The bias is also changed from 0.43 to -0.28 ppm. The negative sign indicates that CT is lower than GOSAT by 0.28. Similarly, comparison between CT and OCO-2 shows improvement in bias and RMSD. The bias was changed from 0.93 to 0.34 and RMSD was changed from 3.77 to 2.57 ppm. These changes are quite significant and inline with what the reviewer expected. These changes are now reflected in various parts of the revised manuscript.

References: 1. Connor, B. J., Boesch, H., Toon, G., Sen, B., Miller, C., and Crisp, D.: Orbiting Carbon Observatory: Inverse method and prospective error analysis, Journal of Geophysical Research: Atmospheres, 113, 2008.

2. O'Dell, C., Connor, B., Bösch, H., O'Brien, D., Frankenberg, C., Castano, R., Christi, M., Eldering, A., Fisher, B., Gunson, M., et al.: The ACOS CO 2 retrieval algorithm-Part 1: Description and validation against synthetic observations, 2012.

3. Rodgers, C. D. and Connor, B. J.: Intercomparison of remote sounding instruments, Journal of Geophysical Research: Atmospheres, 108, 2003.

Please also note the supplement to this comment:
https://www.atmos-meas-tech-discuss.net/amt-2018-84/amt-2018-84-AC1-supplement.pdf

───────────────────

---

## Author Comment (AC2) · 23 Aug 2018

Anteneh Getachew Mengistu and Gizaw Mengistu Tsidu

anteneh.getachew7@aau.edu.et

Authors' response to interactive comment on "Comparison of CO2 from NOAA Carbon Tracker reanalysis model and satellites over Africa" by Dr. Baker

We thank Dr. Baker very much for meticulous and insightful comments. His comments are educational and helpful to improve the manuscript to its current level. We would also like to express our gratitude to Dr. Dietrich G.Feist, the editor, for giving us the chance to get feedback from the two knowledgeable reviewers. In the following, we will

respond to all the comments and questions of Dr. Baker in details indicating page and line numbers of the manuscript where changes are effected as much as it is possible.

Baker's comments: In this manuscript, $CO_2$ mixing ratio fields over the continent of Africa taken from the CarbonTracker-2016 (CT2016) $CO_2$ flux inversion system are compared to $CO_2$ measurements obtained from the GOSAT and OCO-2 satellites. Though it is not said explicitly, this $CO_2$ comparison is presumably done using column averages (commonly referred to as "XCO2 "), either using a straight pressure weighting or by sampling the CT model using the same vertical averaging kernel and prior $CO_2$ vertical profile that the satellite retrieval uses on the true atmosphere.

Response In the older manuscript, we have not used averaging kernel and a priopri $CO_2$ vertical columns to smoothout the model XCO2 based on our simple inspection of the vertical grids used by the model and the satellite retrievals and the judgment that followed to ignore the difference in vertical resolution. However, during this revision we applied the averaging kernel and a priori profile based on the comments from the anonymous reviewer. This led to substantial improvements in the degree of agreement between the two data sets. We thank both reviewers for bringing it to our attention. As a result, we have now updated the manuscript with the new results. As indicated in our response to the anonymous reviewer's comments, we have also included the procedure used for smoothing as proposed by Rodgers and Connor (2003) and Connor et al. (2008) in the revised manuscript in Section 2.4.

D. Baker's comments: The CT2016 system takes a set of somewhat-realistic surface $CO_2$ fluxes (fossil fuel burning, land use change, wildfires, photosynthesis and respiration from the land biosphere, air-sea fluxes) as a first guess, runs them forward through the TM5 off-line atmospheric tracer transport model to obtain $CO_2$ mixing ratios in the interior of the atmosphere, samples these at the time and place of surface in situ data, and uses the measurement differences to improve the initial flux estimates using a fixed-lag ensemble Kalman smoother as the inversion method. CT2016 thus can be thought of as a model of $CO_2$ in the atmosphere that has been forced to agree with

surface CO2 measurements, or alternatively as a glorified interpolation between the in situ measurements. The difference between CT2016 and the satellite data is mainly viewed here as a deficiency in the CT model as compared to the more accurate satellite data. Explanations for the differences are suggested in terms of likely flux errors in the CT2016 model. Comparisons of this sort are of value, but in my view are better viewed from a different perspective. First, if one is interested in understanding the processes driving CO2 fluxes in Africa (fires, photosynthesis, respiration from plants or soils, etc), one might do better to look at the results of an inversion of the satellite CO2 data compared to the fluxes obtained from a system like CT2016 that inverts in situ CO2 data. The reason for this is that the column averages compared here have much information in the upper part of the column that blows in from outside the bounds of Africa and reflect the effect of fluxes from around the globe; thus, column CO2 differences compared over Africa may not reflect the impact of local fluxes, but also of far-field fluxes that have blown in over Africa. The inversion systems, which model atmospheric transport, are supposed to sort this out and apportion the fluxes to the right locations – they are the tool that I would use if understanding fluxes is the goal.

Response We agree that the tone of the discussion tends to portray accurate satellite observations as compared to CT model. We have made several changes to highlight the work is about comparison of two data sets with their own weaknesses and strengths. Clearly, we are not doing model validation. Our intention is to assess the extent of agreement between satellite retrievals and CT model so that in subsequent work which is currently in progress, we can use CT XCO2 with confidence to address some scientific problems including variability at different temporal scales and their possible drivers. This is due to the fact that CT provides synoptic data and additional parameters such as flux. Understanding fluxes is not our goal in this particular work. Some of our suggestions to explain the differences between XCO2 from CT and Satellite are difficult to substantiate in current context. For example, as suggested by Dr. Baker, some of our explanation can be best supported if inversion of the satellite CO2 data is compared to the fluxes obtained from a system like CT2016. However, that is not our

intention. In situation like this, we changed our wording such that our suggestion indicate only a few of many factors as a possible source of difference. But, there are also instance where CT clearly shows weakness to capture XCO2 distribution over Africa as compared to satellite. For example, because of the climate of Africa, it is generally expected to observe high XCO2 along equator and low XCO2 as we move away from equator towards arid regions over Sahel and Kalahari (Williams et al., 2007). This is generally captured by satellite in contrast to CT implying some level of weakness in CT.

D. Baker's comments: The direct comparisons of column CO2 do have value, but one should not assume that the model is wrong and the satellite measurements right, as has been done here for the most part. The retrieval of CO2 from satellite data is beset with a variety of challenges (scattering due to clouds and aerosols, instrument issues, unknown spectroscopy, surface characterization issues, etc.) that often result in systematic errors that, at the moment, are usually removed in a separate step after the retrieval. A key component of this bias correction process is the comparison to independent data, such as column-averaged CO2 measurements from the TCCON network, for example (using comparisons similar to those in this manuscript, but with the model being replaced by the TCCON data). However, comparison to models that have assimilated in situ data (like CT2016) have also been used in this bias correction step. It has been found that the systematic errors in the satellite retrievals are generally larger than those from an ensemble of models, so that the models may be used to help find errors in the satellite retrievals (instead of vice versa, as is done in this manuscript). Hopefully that situation will change as satellite retrieval schemes improve, but at the moment it is not a good assumption that most of the difference between model and satellite measurements can be attributed to model error. Really, there are errors in both model and measurement and one should quantify both with appropriate uncertainties. The authors do provide satellite CO2 retrieval uncertainties here, but these do not capture the impact of systematic errors in the retrievals and are overly optimistic – one cannot assume that because they are small, the majority of the model-measurement difference must be due to model error.

Response We appreciate the comments from Dr. Baker. As indicated in the previous response, we have now soften the tone of several statements in the manuscript but we have still kept some of the possible factors for the difference between the two XCO2 data sets. For example, as correctly indicated here in the review comments by Dr. Baker himself, satellite XCO2 accuracy is affected by a number of factors including clouds and aerosols, pointing errors, spectroscopy, forward modeling error etc. These have been indicated wherever appropriate in both the old and revised manuscript. On the other hand, CT, as a model, has its own limitation including accuracy of reanalysis data used and TM5 tracer. For example, reanalysis data from ECMWF is not well constrained by in-situ observations over Africa due to gaps in existing in-situ data and insufficient number of observations. There are several studies on the performance of reanalysis data over Africa that supports the above understanding (e.g., Nagarajan and Aiyyer, 2004; Mengistu Tsidu, 2012). The assessment of the impact of these inaccuracies in input data is work on its own but certainly it will severely affect the accuracy of CT XCO2 since it is difficult to merely assume CT to be just a simple glorified interpolation of surface fluxes in particular over Africa with just handful of in-situ observations. Therefore, these are points worth discussing when the two data sets are compared. Discussions that reflects our response here have been added in different parts of the manuscript.

D. Baker's comments: In my view, the model-measurement comparisons done here are most valuable for understanding errors in the satellite retrievals. Africa is a continent with wide extremes in surface type (desert vs. rainforest vs savannah) and aerosol loading. These conditions have a strong impact on the satellite retrievals, so a study of this sort over Africa can tell much about how these systematic errors vary geographically. I would encourage the authors to consider whether their comparisons might be better viewed through that lens.

Response We are afraid that it might not be a good idea since we have indicated in the preceding responses, that there are a number factors that affect both satellite and

CT model. Moreover, the agreement between the two in terms of absolute magnitude (low bias and RMSD) and phase (strong correlation) suggests worth using them for further scientific problems on equal footing or one of them depending on suitability to the problem at hand. For example, where the spatial and temporal resolution as well as synoptic nature of the data are important consideration to address a scientific problem, one might prefer CT over Satellite XCO2. We can not dispute the importance of what Dr. Baker suggested but our mere interest in this work is to assess how CT performs as compared to satellite and from there to use CT to solve other scientific problems since it has better spatial and temporal resolution and agrees reasonably well with the only available data over the region i.e., satellite XCO2.

D. Baker's comments: In general, the English punctuation and usage in this manuscript need extensive editing, which I have not attempted to do here. The authors need to clarify certain aspects of their method, most notably what CO2 quantity they are plotting on their figures (presumably column-averaged CO2) and how they do their vertical averaging. Measurement errors of 2 to 5 ppm are not small or reasonable – they are show-stoppers that prevent the satellite data from being useful for inferring surface CO2 fluxes (since these measurement errors feed through to similar errors in the flux results). Some of the statistical metrics presented here are unconventional and are difficult to understand, and do not add much to the presentation – I suggest below that those sections of the document be removed.

Response In the revised manuscript we have made substantial editing to improve the language. The quantity used in all our plots is XCO2. We understand that there was lack of clarity which is now improved. The statistical metrics used here are common in other discipline when the interest is to compare how the distribution of physical quantity as determined from two observations or observation and model agrees at the extreme ends of the distribution. In several cases including current study, the discrepancy between medians of the distribution determined from two observations usually agree very well as mainly reflected by correlation, bias and RMSE. However, a notable difference

between two measurements of the same quantity exists at the two tails of the distribution. This can be qualitatively assessed using scatter plots. The categorical metrics based on qunatile thresholds can disclose the agreement between model and satellite observations in more quantitative manner. The information from these statistics can be used either by modelers or those working on satellite retrievals. For example, in satellite retrievals, smoothing constraint/regularization (e.g. Tikhonov first or second order-so called shape constraint) heavily penalizes the portion of the profile at either low extreme or high extremes. Moreover as the regularization strength depends on the a priori profile constructed usually from climatology (with strong smoothing of the extremes), the part of the distribution in the tails may not represent true observations. That could create huge discrepancy between the model and satellite observations in the tail region of the XCO2 distribution. Therefore, information from these part based on categorical statistics could provide useful insights to experts in satellite remote sensing for further improvements of retrieval strategy. This is just a single example but one can think of similar benefits for modelers to improve accuracy of XCO2. Therefore, we retain section on categorical statistics but enhanced the methodology and discussion sections to improve clarity. Content that reflects parts of this response is included in the manuscript on page 13, lines 2-12 of revised manuscript.

D. Baker's comments: page 1, Line 1: "poor global coverage and resolution of satellite observations": Usually the satellites are thought to have good coverage over the globe, at least when compared to the in-situ data.

Response Indeed, satellites have good global coverage as compared to in suit observations. However, the global coverage from the model simulation is far better than satellite observations. We put this statement to highlight that the global coverage from models are better than that of satellite observations. Since it might create similar wrong impression by the general reader, we have rephrased it such that it now reads as "poor spatial and temporal resolution of satellite observations" in the revised manuscript on page 1 line 1.

D. Baker's comments: Abstract: Here, the fidelity of the CarbonTracker model is being tested by comparing its a posteriori 3-D $CO_2$ fields to $CO_2$ measured from satellites and by the TCCON network of ground-based sun-viewing spectrometers. That would make sense if the accuracy of those measurements was thought to be better than that of the model. But what if the reverse was true? Then it would make more sense to check the accuracy of the measurements by comparing them to the (more accurate) model. In the early days of satellite $CO_2$ data (and we are still in the early days, really) it was thought that latter situation was actually the case, and models were used to correct the satellite data. Given that, some discussion of the assumptions underlying the comparison of CarbonTracker to the satellite and TCCON measurements would be useful in this paper before moving on to the comparison.

Response We appreciate your suggestions and guidance how we should perform the comparison in the best way. In fact, in our introduction, we put some discussions that the Carbon Tracker is more accurate than satellite observations. For example, see this statement "Kulawik et al. (2016) found root mean square deviation of 1.7, and 0.9 ppm in GOSAT and CT2013b XCO2 relative to TCCON respectively" on page 3 line 18 of old and lines 22-23 of revised manuscript) over TCOON sites. However, this could not guarantee that Carbon Tracker performs best at continental and large scales in particular over Africa with extremely limited number of in-situ flux observations in contrast to region covered by TCOON sites. Dr. Baker's argument may work for region with relatively dense network of flux measurements. However, we are dealing with Africa land mass with hardly a handful of flux measuring sites. So, we are in a difficult position to accept the suggestion in the case of this study which focuses on Africa landmass. It is indeed difficult to convincingly take the suggestion by Dr. Baker which amounts to assume flux measurement in the surrounding regions (e.g. Europe and others) can be extrapolated to produce more accurate XCO2 than satellites. Therefore, we still assume that the satellite is better than CT on certain aspect such as producing expected decrease in XCO2 away from equator (see our previous response). A long as the current data gaps in flux measurements over Africa is concerned, the satellite

might have some advantages over CT in reproducing true XCO2. Therefore, we prefer to treat the analysis in this manuscript focusing only on how the satellite and CT XCO2 compare and suggesting all possible scenarios for the difference. Our work is a follow up of many other studies (e.g., Chevallier et al.,2010; Feng et al., 2011; Yingying Jing et al.,2018)

D. Baker's comments: p1 L15: "relative accuracies of 1.22 and 1.95 ppm were found between the model and the two data sets"; these were judged to be "reasonably good". The in situ CO2 data have relative accuracies an order of magnitude better than this (0.1 to 0.2 ppm) – why then should measurements an order of magnitude less accurate be considered "good"? Some guidelines concerning what is considered a good error versus what is considered a bad error ought to be given to support this assertion. The density and coverage of a data type might also factor into this assessment, as well as the importance of random versus systematic errors.

Response This comment is based on our older figures. There is significant improvement after CT is smoothed using retrieval averaging kernel and apriori profiles. For example in the revised manuscript, the relative accuracies is now 1.01 instead of 1.22 in the older version of the manuscript for the comparison between CT2016 and GOSAT. Similarly, the relative accuracies changes from 1.95 (older manuscript version) to 1.18 ppm (revised manuscript) for comparison between CT16NRT17 and OCO-2). These are reasonably good in view of results from previous studies. For example, Deng et al. (2016) founds a regional accuracy of 0.62 ppm between ACOS and TCCON and 0.93 ppm between NIES and TCCON for comparison over 11 TCCON sites. These figures are comparable and reasonable given that our comparison is over 426 pixels and region with diverse climate in contrast to limited number of sites (e.g., 11 TCCON sites). Moreover, TCCON is more accurate as compared to satellites whereas in this study we are dealing with less accurate satellite and model, both of which contribute to the relative accuracy. Apart from these factors, other authors have indicated vertical transport is more variable among transport models (Gurney et al., 2002) and probably

more error-prone. Hungershoefer et al. (2010) have shown that an error of up to 3 ppm may be observed over site with complex circulation and fluxes based on numerical experiment (simulation). The above information is highlighted in the manuscript to strengthen our conclusion and help the readers understand how good is good in the context of this study.

D. Baker's comments: p1 L17-18: "...probability of detection ranges from 0.6 to 1 and critical success index ranges from 0.4 to 1..." These statistics may not be familiar to the general reader. Maybe say "(see main text for definitions)" when using these in the abstract?

Response We accepted the comment and made changes in the abstract accordingly. Moreover, we have made a lot of changes that includes basic definitions, equations and the physical acceptable limits of the individual statistical parameter in Section 2.4. These changes are crucial to understand and interpret the results.

D. Baker's comments: p1 L20-21: "GOSAT and OCO-2 XCO2 are lower than that of CT2016 by upto 4 ppm over North Africa (10 N −35 N) whereas it exceeds CT2016 X CO2 by 3 ppm over Equatorial Africa (10 S−10 N)." It seems like the sign of this is wrong. Also, these satellite data are raw or bias-corrected?

Response Dr. Baker is right that these figures are too high. We have now realized that the discrepancy arises mainly from ignoring the difference in the vertical resolution of the CT and satellite retrievals. Following one of previous comments, we have already recalculated XCO2 from CT using averaging kernel weighting function and a priopri profiles of the retrievals. The comparison with smoothed XCO2 has resulted in much lower difference. Moreover, GOSAT XCO2 is lower than that of CT. Therefore, the statement is amended such that it now reads as "Spatially, OCO-2 XCO2 are lower than that of CT16NRT17 by 3 ppm over some regions in North Africa (e.g., Egypt, Libya Mali) whereas it exceeds CT16NRT17 by 2 ppm over Equatorial Africa (100S − 100N)." This change is made on page 1 lines 23-25 of the revised manuscript. We

would like to confirm that data from both satellite are biased corrected. This information is also included in the revised manuscript on page 3 line 11 and 20.

D. Baker's comments: p1 L25: "In these cases, the model overestimates the local emissions and underestimate CO2 loss." Here, it is assumed that the satellite data are correct and the model is wrong. But what if the reverse was actually true? In reality, both model and satellite are wrong, to differing degrees that are quantified using the metric of uncertainty. You should give reasons why you think the uncertainty on the satellite measurements is lower than the uncertainty on the modeled CO2, to defend why you think the model is wrong and the satellites right. The satellites have retrieval biases and random errors that could easily be larger than the modeling errors in CO2 (errors due to transport, errors in the in situ data, inversion assumptions, etc).

Response The comment is well taken and the statement is removed from the revised manuscript.

D. Baker's comments: p2 L25-28: In introducing the TCCON data, you should note that the TCCON network measures a column-integrated CO2 mixing ratio; yes, it is "ground-based", since the sensor sits on the ground, but the measurement is a column average (unlike the ground-based in situ data) and that is why it is useful for validating the satellite data, which are also column averages.

Response We accept your comment and rephrased the statement on page 2 line 31 such that it now reads as "Total Carbon Column Observing Network (TCCON) is a notable one since it provides accurate and high–frequency measurements of column-integrated CO2 mixing ratio"

D. Baker's comments: p3 L3: "Other studies have revealed that significant improve-ment in estimation of weekly and monthly CO2 fluxes can be achieved subject to CO2 retrieval error of less than 4 ppm" This may be true if the errors are purely random – with enough low precision data of this sort, the errors may average down to something lower and more useful. However, if this is accompanied by systematic errors of similar

magnitude, these will not average out. Therefore some discussion of random versus systematic error must be added when giving out these error quantities.

Response This is a mere reference to other works.

D. Baker's comments: p3 L21: "V02.XX" – you need to complete the version numbers for two products on this line

Response We have indicated the specific version of GEOS-Chem used from the cited paper on page 3 line 31 of the revised manuscript. However, the NEIS L2 V02.xx remained the same. According to Yoshida et al. (2013), "xx" in NIES L2 V02.xx and NIES L2 V01.xx initially planned for the specific versions of NIES L1 used in the retrievals to generate L2 data. However, the L1 data version remains the same in all NIES L2 and therefore it remains xx in all versions to the best of our knowledge.

D. Baker's comments: p4 L3-4: "These findings suggest that it is important to assess the accuracy and uncertainty of XCO2 from models with respect to observations" Again, the authors point to the differences between satellite data and models, then leap to the assumption that the models are responsible for most of those differences when the measurements might actually be more responsible. Some discussion of which is more error-prone (model or measurement) is needed.

Response We have rephrased it such that it now reads as "These findings suggest that it is important to assess how satellite and model XCO2 compare with each other over other regions". This change is made on page 4 lines 4-5 of the revised manuscript.

D. Baker's comments: p4 L27: "CT2016 and CT2017 respectively": you should not refer to the 2016 near real-time CT as "CT2017", as that term is reserved for the subsequent release of the full data span (the release that would have been put out in 2017 if the releases were put out when indicated; the release that finishes at the end of 2016). The near-real-time releases are different from the standard releases in that they bring out a subset of the full data set, without the usual quality assurances, for early

use by modelers. Even if that is only a label for use in this manuscript, it will confuse people.

Response Thank for your suggestion. CT2017 was replaced by CT16NRT17 in the revised manuscript.

D. Baker's comments: p5 L1: Which version of ACOS GOSAT retrievals did you use in this study?

Response Thank you for pointing out the missing information which was also indicated by the anonymous reviewer. The version used in this study is ACOS B3.5 Lite. This information is incorporated in Section 2.2 of the revised manuscript.

D. Baker's comments: p5 L2-3: "GOSAT is the world's first spacecraft to measure the concentrations of carbon dioxide and methane, the two major greenhouse gases, from space." Not true: many satellites/instruments measured both in the thermal IR. Also SCIAMACHY, which came out much earlier, measured both in the near-IR. What you could say is that GOSAT was the first spacecraft dedicated solely to measuring $CO_2$ and $CH_4$.

Response The comment is well taken and included on page 5 line 1.

D. Baker's comments: p5 L15: When discussing the OCO-2 data, you must say what version of retrieval you are using: version 7? Also, you should indicate whether you are looking at raw $XCO_2$ values or bias-corrected values. If bias corrected, you should indicate whether you used the additional "s31" bias correction, designed to correct albedo-related errors not caught in the initial bias correction. This extra bias correction could be particularly important over Africa, with its large differences between desert and tropical rainforest. Also, you should discuss whether you sample your model with the OCO-2 averaging kernel and prior $CO_2$ profile when comparing to the OCO-2 retrievals.

Response This was also a comment from the anonymous reviewer. So, it has already

been corrected. The version of OCO-2 (i.e., OCO-2 V7 lite level 2) used is biased-corrected. However, we have not used albedo bias correction. We have smoothed the CT $XCO_2$ using averaging kernel and a priori profiles with respect both satellite data sets following your previous comments and that of anonymous reviewer's comments (see also one of the previous responses).

D. Baker's comments: p5 L21: (Methods): the satellites estimate both a profile of $CO_2$ on 20 levels, as well as a vertical weighted average, "$XCO_2$", computed from these. Which do you compare to? If the latter, you should discuss the vertical weighting kernel for $XCO_2$ and how you sample your model to account for this. This methods section seems to be the correct place to discuss this.

Response We used the column average $XCO_2$ from the satellites and compared it to the $XCO_2$ computed from the model $CO_2$ profile. This $CO_2$ profile is extracted and interpolated to resolution and vertical level of the satellites. Then following the procedures described in Coner et al., (2008) we transform the model $CO_2$ profile to model averaged values ($XCO_2$). Brief discussion and mathematical expression of the technique are included in section 2.4 of the revised manuscript.

D. Baker's comments: p6 L19: "Fig. 1 shows the five-years average of CT2016 (Fig. 1a) and GOSAT (Fig. 1b) $XCO_2$ distribution." It would have been more useful to do the 5-year average over a true 5-year span, rather than the "April 2009 to June 2014" span that you used. As it stands, you have an extra 3 month span (April to June) that is unbalanced by the remainder of an annual cycle. As a result, this three month span throws off what otherwise would be the average of five full annual cycles. If flux is positive north of the equator during April to June (which it probably is, this being right at the end of the burn season there) then you tend to see positive values there in Figure 1a and 1b. This is due to the seasonality of the flux rather than the annual mean value. It might be easier to discuss annual mean fluxes in the text if these plots reflected averages over pure annual cycles.

Response We agree that the extra 3 months may destroy the symmetry of the annual cycles over Africa. In the revised manuscript we change the study period to span only 5 years (from May 2009 to April 2014).

D. Baker's comments: p7 L3: "respiration from forest in the region is overestimated" Why do you think that this is not due to photosynthesis being underestimated?

Response It is possible that both or one of the two can cause the observed discrepancy. It might be equally possible cloud contamination may affect the satellite $XCO_2$ such that the satellite estimates are unusually high (O'Dell et al., 2012). The equatorial region, where the difference is observed, is known to high altitude thin cirrus clouds. The revised manuscript was edited to reflect these possibilities on page 8 lines 18-23.

D. Baker's comments: p7 last line: "standard deviation of 1.31 ppm indicating better consistency and less potential outliers." Better than what?

Response In the revised manuscript this standard deviation was only 0.98 and the statement was reworded as "standard deviation of 0.98 ppm indicating good regional consistency and low potential outliers" on page 8 lines 30-31.

D. Baker's comments: p8 L3: change "require" to "permit" Also, if the difference can be up to 1.5 degrees on either side, then we are talking about a box of 3 degrees on a side in both latitude and longitude.

Response Accepted. Change is made on page 8 line 33.

D. Baker's comments: On p5 L29 you should then change the box from being 1.5 degrees long and wide to 3 degrees long and wide.

Response Accepted. Change is made on page 8 line 33.

D. Baker's comments: Fig 3 caption: please provide units (ppm)

Response The unit is now indicated as the title of the color bars in the revised manuscript.
D. Baker's comments: p8 L19: A correlation should be unitless – remove "ppm"

Response Removed

D. Baker's comments: Figure 3: It is not clear to me why Figures 3a and 1c are different – they are both showing the difference (or bias) between CT2016 and GOSAT on an annual mean basis. So why aren't they identical?

Response We agree that the mean difference and the bias should be the same. But Fig.1c is based on the full coincident data sets while in Fig.3a, grid points with less than 10 data points are removed to establish reliable statistics. Although such restriction is not a requirement for simple difference which is temporally averaged in Fig.1c, we have now excluded those points in Fig.1c to avoid similar misunderstanding in the revised manuscript.

D. Baker's comments: p8 L20: Good – here you are examining whether some of the difference might be due to the satellite data rather than the model.

Response Appreciated the complement and changes in other parts of the manuscript have similar tone in the revised manuscript.

D. Baker's comments: Section 3.2: The statistics presented here are abstruse and require the reader to go back to the other references not only for what the statistics mean, but even what the acronyms stand for. It is not clear what the message is that these statistics convey. The only point I gleaned from the discussion was that the annual mean bias between model and data is higher where there are fewer data to average over – that is a point that could be made with Figure 1 alone. I would suggest deleting this section altogether.

Response As the statistical tools used in this section are uncommon in our science community, it is not at all surprising to see more inquiry on the tools. We have clearly indicated why it is important by citing specific application on page 6 of this author response and in the manuscript in Section 2.4 as well as in the discussion section from

page 13 lines 2-12. Moreover, amendment in different parts of the manuscript will now provide more information that clarify the importance of the statistical tools (see also the author response to anonymous reviewer).

D. Baker's comments: p13 L9, p14 L11-14: Here you are suggesting that local fluxes are responsible for the changes in CO2 seen over Africa, but it is quite possible that a large part of these changes are due to fluxes elsewhere, over different continents, blowing downwind over Africa. Without inversion results linking concentrations to the fluxes that caused them, your argument will remain weak using this approach.

Response We have taken note of your suggestion and included it as one of possible scenarios in the revised manuscript on page 18 lines 1-3.

D. Baker's comments: p25 L16: "Satellite observations are able to capture the CO2 concentration truly and objectively." I don't understand how you can say this based on the results that you have shown. You have shown large differences between Carbon-Tracker and the satellite data, on the order of 5 ppm on a seasonal and regional basis. This could be due to satellite retrieval problems in addition to the model problems that have been emphasized here. You haven't shown any data or comparisons that suggest the satellite data are better than the model, really. Maybe you could compare GOSAT and OCO-2 across the short time period that they overlap (late 2014, 2015) and if they agree closely then suggest they are both getting the right answer. From the plots you have showed in this manuscript, however, it looks like there are significant differences between OCO-2 and GOSAT... suggesting that the satellite errors are significant.

Response We have accepted that we are a bit biased in favor of satellites. We removed the statement. However, since we do not have overlap between the two satellites (GOSAT and OCO-2) in the version used in this study, we did not compare them.

References 1. Chevallier, F.; Feng, L.; Bösch, H.; Palmer, P.I.; Rayner, P.J. On the impact of transport model errors for the estimation of CO 2 surface fluxes from GOSAT observations. Geophys. Res. Lett. 2010, 37. 2. Connor, B. J., Boesch, H., Toon,

G., Sen, B., Miller, C., and Crisp, D.: Orbiting Carbon Observatory: Inverse method and prospective error analysis, Journal of Geophysical Research: Atmospheres, 113, 2008. 3. Deng, A., Yu, T., Cheng, T., Gu, X., Zheng, F., and Guo, H.: Intercomparison of Carbon Dioxide Products Retrieved from GOSAT Short-Wavelength Infrared Spectra for Three Years (2010–2012), Atmosphere, 7, 109, 2016. 4. Feng, L.; Palmer, P.I.; Yang, Y.; Yantosca, R.M.; Kawa, S.R.; Paris, J.D.; Matsueda, H.; Machida, T. Evaluating a 3-D transport model of atmospheric CO 2 using ground-based, aircraft, and space-borne data. Atmos. Chem. Phys. 2011, 11, 2789–2803. 5. Gurney, K. R., Law, R. M., Denning, A. S., Rayner, P. J., Baker, D., Bousquet, P., Bruhwiler, L., Chen, Y.-H., Ciais, P., Fan, S., et al.: Towards robust regional estimates of CO2 sources and sinks using atmospheric transport models, Nature, 415, 626–630, 2002.

6. Hungershoefer, K., Breon, F.-M., Peylin, P., Chevallier, F., Rayner, P., Klonecki, A., Houweling, S., and Marshall, J.: Evaluation of various observing systems for the global monitoring of CO 2 surface fluxes, Atmospheric chemistry and physics, 10, 10 503–10 520, 2010. 7. Jing, Y., Wang, T., Zhang, P., Chen, L., Xu, N., and Ma, Y.: Global Atmospheric CO2 Concentrations Simulated by GEOS-Chem: Comparison with GOSAT, Carbon Tracker and Ground-Based Measurements, Atmosphere, 9, 175, 2018. 8. O'Dell, C., Connor, B., Bösch, H., O'Brien, D., Frankenberg, C., Castano, R., Christi, M., Eldering, A., Fisher, B., Gunson, M., et al.: The ACOS CO 2 retrieval algorithm-Part 1: Description and validation against synthetic observations, 2012.

9. Mengistu Tsidu, G.: High-resolution monthly rainfall database for Ethiopia: Homogenization, reconstruction, and gridding, Journal of Climate,25 , 8422–8443, 2012. 10. Nagarajan, B. and Aiyyer, A. R.: Performance of the ECMWF operational analyses over the tropical Indian Ocean, Monthly weather review, 132, 2275–2282, 2004. 11. Rodgers, C. D. and Connor, B. J.: Intercomparison of remote sounding instruments, Journal of Geophysical Research: Atmospheres, 108, 2003.

12. Yoshida, Y., Kikuchi, N., Morino, I., Uchino, O., Oshchepkov, S., Bril, A., Saeki, T., Schutgens, N., Toon, G., Wunch, D., et al.: Improvement of the retrieval algorithm for

GOSAT SWIR XCO2 and XCH4 and their validation using TCCON data, 2013.

13. Williams, C. A., Hanan, N. P., Neff, J. C., Scholes, R. J., Berry, J. A., Denning, A. S., and Baker, D. F.: Africa and the global carbon cycle, Carbon balance and management, 2, 3, 2007.

Please also note the supplement to this comment:
https://www.atmos-meas-tech-discuss.net/amt-2018-84/amt-2018-84-AC2-supplement.pdf

---

## Author Comment (AC4) · 10 Sep 2018

Authors' response to interactive comment on "Comparison of CO2 from NOAA Carbon Tracker reanalysis model and satellites over Africa" by Dr. Baker

We thank Dr. Baker very much for meticulous and insightful comments. His comments are educational and helpful to improve the manuscript to its current level. We would also like to express our gratitude to Dr. Dietrich G.Feist, the editor, for giving us the chance to get feedback from the two knowledgeable reviewers. In the following, we will

respond to all the comments and questions of Dr. Baker in details indicating page and line numbers of the manuscript where changes are effected as much as it is possible. Please look at the manuscript attached as a supplement.

Baker's comments: In this manuscript, CO2 mixing ratio fields over the continent of Africa taken from the CarbonTracker-2016 (CT2016) CO2 flux inversion system are compared to CO2 measurements obtained from the GOSAT and OCO-2 satellites. Though it is not said explicitly, this CO2 comparison is presumably done using column averages (commonly referred to as "XCO2 "), either using a straight pressure weighting or by sampling the CT model using the same vertical averaging kernel and prior CO2 vertical profile that the satellite retrieval uses on the true atmosphere.

Response: In the older manuscript, we have not used averaging kernel and a priopri CO2 vertical columns to smoothout the model XCO2 based on our simple inspection of the vertical grids used by the model and the satellite retrievals and the judgment that followed to ignore the difference in vertical resolution. However, during this revision we applied the averaging kernel and a priori profile based on the comments from the anonymous reviewer. This led to substantial improvements in the degree of agreement between the two data sets. We thank both reviewers for bringing it to our attention. As a result, we have now updated the manuscript with the new results. As indicated in our response to the anonymous reviewer's comments, we have also included the procedure used for smoothing as proposed by Rodgers and Connor (2003) and Connor et al. (2008) in the revised manuscript in Section 2.4.

D. Baker's comments: The CT2016 system takes a set of somewhat-realistic surface CO2 fluxes (fossil fuel burning, land use change, wildfires, photosynthesis and respiration from the land biosphere, air-sea fluxes) as a first guess, runs them forward through the TM5 off-line atmospheric tracer transport model to obtain CO2 mixing ratios in the interior of the atmosphere, samples these at the time and place of surface in situ data, and uses the measurement differences to improve the initial flux estimates using a fixed-lag ensemble Kalman smoother as the inversion method. CT2016 thus can be

thought of as a model of CO2 in the atmosphere that has been forced to agree with surface CO2 measurements, or alternatively as a glorified interpolation between the in situ measurements. The difference between CT2016 and the satellite data is mainly viewed here as a deficiency in the CT model as compared to the more accurate satellite data. Explanations for the differences are suggested in terms of likely flux errors in the CT2016 model. Comparisons of this sort are of value, but in my view are better viewed from a different perspective. First, if one is interested in understanding the processes driving CO2 fluxes in Africa (fires, photosynthesis, respiration from plants or soils, etc), one might do better to look at the results of an inversion of the satellite CO2 data compared to the fluxes obtained from a system like CT2016 that inverts in situ CO2 data. The reason for this is that the column averages compared here have much information in the upper part of the column that blows in from outside the bounds of Africa and reflect the effect of fluxes from around the globe; thus, column CO2 differences compared over Africa may not reflect the impact of local fluxes, but also of far-field fluxes that have blown in over Africa. The inversion systems, which model atmospheric transport, are supposed to sort this out and apportion the fluxes to the right locations – they are the tool that I would use if understanding fluxes is the goal.

Response: We agree that the tone of the discussion tends to portray accurate satellite observations as compared to CT model. We have made several changes to highlight the work is about comparison of two data sets with their own weaknesses and strengths. Clearly, we are not doing model validation. Our intention is to assess the extent of agreement between satellite retrievals and CT model so that in subsequent work which is currently in progress, we can use CT XCO2 with confidence to address some scientific problems including variability at different temporal scales and their possible drivers. This is due to the fact that CT provides synoptic data and additional parameters such as flux. Understanding fluxes is not our goal in this particular work. Some of our suggestions to explain the differences between XCO2 from CT and Satellite are difficult to substantiate in current context. For example, as suggested by Dr. Baker, some of our explanation can be best supported if inversion of the satellite CO2 data is

compared to the fluxes obtained from a system like CT2016. However, that is not our intention. In situation like this, we changed our wording such that our suggestion indicate only a few of many factors as a possible source of difference. But, there are also instance where CT clearly shows weakness to capture XCO2 distribution over Africa as compared to satellite. For example, because of the climate of Africa, it is generally expected to observe high XCO2 along equator and low XCO2 as we move away from equator towards arid regions over Sahel and Kalahari (Williams et al., 2007). This is generally captured by satellite in contrast to CT implying some level of weakness in CT.

D. Baker's comments: The direct comparisons of column CO2 do have value, but one should not assume that the model is wrong and the satellite measurements right, as has been done here for the most part. The retrieval of CO2 from satellite data is beset with a variety of challenges (scattering due to clouds and aerosols, instrument issues, unknown spectroscopy, surface characterization issues, etc.) that often result in systematic errors that, at the moment, are usually removed in a separate step after the retrieval. A key component of this bias correction process is the comparison to independent data, such as column-averaged CO2 measurements from the TCCON network, for example (using comparisons similar to those in this manuscript, but with the model being replaced by the TCCON data). However, comparison to models that have assimilated in situ data (like CT2016) have also been used in this bias correction step. It has been found that the systematic errors in the satellite retrievals are generally larger than those from an ensemble of models, so that the models may be used to help find errors in the satellite retrievals (instead of vice versa, as is done in this manuscript). Hopefully that situation will change as satellite retrieval schemes improve, but at the moment it is not a good assumption that most of the difference between model and satellite measurements can be attributed to model error. Really, there are errors in both model and measurement and one should quantify both with appropriate uncertainties. The authors do provide satellite CO2 retrieval uncertainties here, but these do not capture the impact of systematic errors in the retrievals and are overly optimistic – one cannot assume that because they are small, the majority of the model-measurement

difference must be due to model error.

Response: We appreciate the comments from Dr. Baker. As indicated in the previous response, we have now soften the tone of several statements in the manuscript but we have still kept some of the possible factors for the difference between the two XCO2 data sets. For example, as correctly indicated here in the review comments by Dr. Baker himself, satellite XCO2 accuracy is affected by a number of factors including clouds and aerosols, pointing errors, spectroscopy, forward modeling error etc. These have been indicated wherever appropriate in both the old and revised manuscript. On the other hand, CT, as a model, has its own limitation including accuracy of reanalysis data used and TM5 tracer. For example, reanalysis data from ECMWF is not well constrained by in-situ observations over Africa due to gaps in existing in-situ data and insufficient number of observations. There are several studies on the performance of reanalysis data over Africa that supports the above understanding (e.g., Nagarajan and Aiyyer, 2004; Mengistu Tsidu, 2012). The assessment of the impact of these inaccuracies in input data is work on its own but certainly it will severely affect the accuracy of CT XCO2 since it is difficult to merely assume CT to be just a simple glorified interpolation of surface fluxes in particular over Africa with just handful of in-situ observations. Therefore, these are points worth discussing when the two data sets are compared. Discussions that reflects our response here have been added in different parts of the manuscript.

D. Baker's comments: In my view, the model-measurement comparisons done here are most valuable for understanding errors in the satellite retrievals. Africa is a continent with wide extremes in surface type (desert vs. rainforest vs savannah) and aerosol loading. These conditions have a strong impact on the satellite retrievals, so a study of this sort over Africa can tell much about how these systematic errors vary geographically. I would encourage the authors to consider whether their comparisons might be better viewed through that lens.

Response: We are afraid that it might not be a good idea since we have indicated in

the preceding responses, that there are a number factors that affect both satellite and CT model. Moreover, the agreement between the two in terms of absolute magnitude (low bias and RMSD) and phase (strong correlation) suggests worth using them for further scientific problems on equal footing or one of them depending on suitability to the problem at hand. For example, where the spatial and temporal resolution as well as synoptic nature of the data are important consideration to address a scientific problem, one might prefer CT over Satellite XCO2. We can not dispute the importance of what Dr. Baker suggested but our mere interest in this work is to assess how CT performs as compared to satellite and from there to use CT to solve other scientific problems since it has better spatial and temporal resolution and agrees reasonably well with the only available data over the region i.e., satellite XCO2.

D. Baker's comments: In general, the English punctuation and usage in this manuscript need extensive editing, which I have not attempted to do here. The authors need to clarify certain aspects of their method, most notably what CO2 quantity they are plotting on their figures (presumably column-averaged CO2) and how they do their vertical averaging. Measurement errors of 2 to 5 ppm are not small or reasonable – they are show-stoppers that prevent the satellite data from being useful for inferring surface CO2 fluxes (since these measurement errors feed through to similar errors in the flux results). Some of the statistical metrics presented here are unconventional and are difficult to understand, and do not add much to the presentation – I suggest below that those sections of the document be removed.

Response: In the revised manuscript we have made substantial editing to improve the language. The quantity used in all our plots is XCO2. We understand that there was lack of clarity which is now improved. The statistical metrics used here are common in other discipline when the interest is to compare how the distribution of physical quantity as determined from two observations or observation and model agrees at the extreme ends of the distribution. In several cases including current study, the discrepancy between medians of the distribution determined from two observations usually agree very

well as mainly reflected by correlation, bias and RMSE. However, a notable difference between two measurements of the same quantity exists at the two tails of the distribution. This can be qualitatively assessed using scatter plots. The categorical metrics based on qunatile thresholds can disclose the agreement between model and satellite observations in more quantitative manner. The information from these statistics can be used either by modelers or those working on satellite retrievals. For example, in satellite retrievals, smoothing constraint/regularization (e.g. Tikhonov first or second order-so called shape constraint) heavily penalizes the portion of the profile at either low extreme or high extremes. Moreover as the regularization strength depends on the a priori profile constructed usually from climatology (with strong smoothing of the extremes), the part of the distribution in the tails may not represent true observations. That could create huge discrepancy between the model and satellite observations in the tail region of the XCO2 distribution. Therefore, information from these part based on categorical statistics could provide useful insights to experts in satellite remote sensing for further improvements of retrieval strategy. This is just a single example but one can think of similar benefits for modelers to improve accuracy of XCO2. Therefore, we retain section on categorical statistics but enhanced the methodology and discussion sections to improve clarity. Content that reflects parts of this response is included in the manuscript on page 13, lines 2-12 of revised manuscript.

D. Baker's comments: page 1, Line 1: "poor global coverage and resolution of satellite observations": Usually the satellites are thought to have good coverage over the globe, at least when compared to the in-situ data.

Response: Indeed, satellites have good global coverage as compared to in suit observations. However, the global coverage from the model simulation is far better than satellite observations. We put this statement to highlight that the global coverage from models are better than that of satellite observations. Since it might create similar wrong impression by the general reader, we have rephrased it such that it now reads as "poor spatial and temporal resolution of satellite observations" in the revised manuscript on

page 1 line 1.

D. Baker's comments: Abstract: Here, the fidelity of the CarbonTracker model is being tested by comparing its a posteriori 3-D CO2 fields to CO2 measured from satellites and by the TCCON network of ground-based sun-viewing spectrometers. That would make sense if the accuracy of those measurements was thought to be better than that of the model. But what if the reverse was true? Then it would make more sense to check the accuracy of the measurements by comparing them to the (more accurate) model. In the early days of satellite CO2 data (and we are still in the early days, really) it was thought that latter situation was actually the case, and models were used to correct the satellite data. Given that, some discussion of the assumptions underlying the comparison of CarbonTracker to the satellite and TCCON measurements would be useful in this paper before moving on to the comparison.

Response: We appreciate your suggestions and guidance how we should perform the comparison in the best way. In fact, in our introduction, we put some discussions that the Carbon Tracker is more accurate than satellite observations. For example, see this statement "Kulawik et al. (2016) found root mean square deviation of 1.7, and 0.9 ppm in GOSAT and CT2013b XCO2 relative to TCCON respectively" on page 3 line 18 of old and lines 22-23 of revised manuscript) over TCOON sites. However, this could not guarantee that Carbon Tracker performs best at continental and large scales in particular over Africa with extremely limited number of in-situ flux observations in contrast to region covered by TCOON sites. Dr. Baker's argument may work for region with relatively dense network of flux measurements. However, we are dealing with Africa land mass with hardly a handful of flux measuring sites. So, we are in a difficult position to accept the suggestion in the case of this study which focuses on Africa landmass. It is indeed difficult to convincingly take the suggestion by Dr. Baker which amounts to assume flux measurement in the surrounding regions (e.g. Europe and others) can be extrapolated to produce more accurate XCO2 than satellites. Therefore, we still assume that the satellite is better than CT on certain aspect such as producing

expected decrease in XCO2 away from equator (see our previous response). A long as the current data gaps in flux measurements over Africa is concerned, the satellite might have some advantages over CT in reproducing true XCO2. Therefore, we prefer to treat the analysis in this manuscript focusing only on how the satellite and CT XCO2 compare and suggesting all possible scenarios for the difference. Our work is a follow up of many other studies (e.g., Chevallier et al.,2010; Feng et al., 2011; Yingying Jing et al.,2018)

D. Baker's comments: p1 L15: "relative accuracies of 1.22 and 1.95 ppm were found between the model and the two data sets"; these were judged to be "reasonably good". The in situ CO2 data have relative accuracies an order of magnitude better than this (0.1 to 0.2 ppm) – why then should measurements an order of magnitude less accurate be considered "good"? Some guidelines concerning what is considered a good error versus what is considered a bad error ought to be given to support this assertion. The density and coverage of a data type might also factor into this assessment, as well as the importance of random versus systematic errors.

Response: This comment is based on our older figures. There is significant improvement after CT is smoothed using retrieval averaging kernel and apriori profiles. For example in the revised manuscript, the relative accuracies is now 1.01 instead of 1.22 in the older version of the manuscript for the comparison between CT2016 and GOSAT. Similarly, the relative accuracies changes from 1.95 (older manuscript version) to 1.18 ppm (revised manuscript) for comparison between CT16NRT17 and OCO-2). These are reasonably good in view of results from previous studies. For example, Deng et al. (2016) founds a regional accuracy of 0.62 ppm between ACOS and TCCON and 0.93 ppm between NIES and TCCON for comparison over 11 TCCON sites. These figures are comparable and reasonable given that our comparison is over 426 pixels and region with diverse climate in contrast to limited number of sites (e.g., 11 TCCON sites). Moreover, TCCON is more accurate as compared to satellites whereas in this study we are dealing with less accurate satellite and model, both of which contribute

to the relative accuracy. Apart from these factors, other authors have indicated vertical transport is more variable among transport models (Gurney et al., 2002) and probably more error-prone. Hungershoefer et al. (2010) have shown that an error of up to 3 ppm may be observed over site with complex circulation and fluxes based on numerical experiment (simulation). The above information is highlighted in the manuscript to strengthen our conclusion and help the readers understand how good is good in the context of this study.

D. Baker's comments: p1 L17-18: "...probability of detection ranges from 0.6 to 1 and critical success index ranges from 0.4 to 1..." These statistics may not be familiar to the general reader. Maybe say "(see main text for definitions)" when using these in the abstract?

Response: We accepted the comment and made changes in the abstract accordingly. Moreover, we have made a lot of changes that includes basic definitions, equations and the physical acceptable limits of the individual statistical parameter in Section 2.4. These changes are crucial to understand and interpret the results.

D. Baker's comments: p1 L20-21: "GOSAT and OCO-2 XCO2 are lower than that of CT2016 by upto 4 ppm over North Africa (10 N −35 N) whereas it exceeds CT2016 X CO2 by 3 ppm over Equatorial Africa (10 S−10 N)." It seems like the sign of this is wrong. Also, these satellite data are raw or bias-corrected?

Response Dr. Baker is right that these figures are too high. We have now realized that the discrepancy arises mainly from ignoring the difference in the vertical resolution of the CT and satellite retrievals. Following one of previous comments, we have already recalculated XCO2 from CT using averaging kernel weighting function and a priopri profiles of the retrievals. The comparison with smoothed XCO2 has resulted in much lower difference. Moreover, GOSAT XCO2 is lower than that of CT. Therefore, the statement is amended such that it now reads as "Spatially, OCO-2 XCO2 are lower than that of CT16NRT17 by 3 ppm over some regions in North Africa (e.g., Egypt,

Libya Mali) whereas it exceeds CT16NRT17 by 2 ppm over Equatorial Africa (100S − 100N)." This change is made on page 1 lines 23-25 of the revised manuscript. We would like to confirm that data from both satellite are biased corrected. This information is also included in the revised manuscript on page 3 line 11 and 20.

D. Baker's comments: p1 L25: "In these cases, the model overestimates the local emissions and underestimate CO2 loss." Here, it is assumed that the satellite data are correct and the model is wrong. But what if the reverse was actually true? In reality, both model and satellite are wrong, to differing degrees that are quantified using the metric of uncertainty. You should give reasons why you think the uncertainty on the satellite measurements is lower than the uncertainty on the modeled CO2, to defend why you think the model is wrong and the satellites right. The satellites have retrieval biases and random errors that could easily be larger than the modeling errors in CO2 (errors due to transport, errors in the in situ data, inversion assumptions, etc).

Response: The comment is well taken and the statement is removed from the revised manuscript.

D. Baker's comments: p2 L25-28: In introducing the TCCON data, you should note that the TCCON network measures a column-integrated CO2 mixing ratio; yes, it is "ground-based", since the sensor sits on the ground, but the measurement is a column average (unlike the ground-based in situ data) and that is why it is useful for validating the satellite data, which are also column averages.

Response: We accept your comment and rephrased the statement on page 2 line 31 such that it now reads as "Total Carbon Column Observing Network (TCCON) is a notable one since it provides accurate and high–frequency measurements of column-integrated CO2 mixing ratio"

D. Baker's comments: p3 L3: "Other studies have revealed that significant improvement in estimation of weekly and monthly CO2 fluxes can be achieved subject to CO2 retrieval error of less than 4 ppm" This may be true if the errors are purely random −

with enough low precision data of this sort, the errors may average down to something lower and more useful. However, if this is accompanied by systematic errors of similar magnitude, these will not average out. Therefore some discussion of random versus systematic error must be added when giving out these error quantities.

Response: This is a mere reference to other works.

D. Baker's comments: p3 L21: "V02.XX" – you need to complete the version numbers for two products on this line

Response: We have indicated the specific version of GEOS-Chem used from the cited paper on page 3 line 31 of the revised manuscript. However, the NEIS L2 V02.xx remained the same. According to Yoshida et al. (2013), "xx" in NIES L2 V02.xx and NIES L2 V01.xx initially planned for the specific versions of NIES L1 used in the retrievals to generate L2 data. However, the L1 data version remains the same in all NIES L2 and therefore it remains xx in all versions to the best of our knowledge.

D. Baker's comments: p4 L3-4: "These findings suggest that it is important to assess the accuracy and uncertainty of XCO2 from models with respect to observations" Again, the authors point to the differences between satellite data and models, then leap to the assumption that the models are responsible for most of those differences when the measurements might actually be more responsible. Some discussion of which is more error-prone (model or measurement) is needed.

Response: We have rephrased it such that it now reads as "These findings suggest that it is important to assess how satellite and model XCO2 compare with each other over other regions". This change is made on page 4 lines 4-5 of the revised manuscript.

D. Baker's comments: p4 L27: "CT2016 and CT2017 respectively": you should not refer to the 2016 near real-time CT as "CT2017", as that term is reserved for the subsequent release of the full data span (the release that would have been put out in 2017 if the releases were put out when indicated; the release that finishes at the end of

2016). The near-real-time releases are different from the standard releases in that they bring out a subset of the full data set, without the usual quality assurances, for early use by modelers. Even if that is only a label for use in this manuscript, it will confuse people.

Response: Thank for your suggestion. CT2017 was replaced by CT16NRT17 in the revised manuscript.

D. Baker's comments: p5 L1: Which version of ACOS GOSAT retrievals did you use in this study?

Response: Thank you for pointing out the missing information which was also indicated by the anonymous reviewer. The version used in this study is ACOS B3.5 Lite. This information is incorporated in Section 2.2 of the revised manuscript.

D. Baker's comments: p5 L2-3: "GOSAT is the world's first spacecraft to measure the concentrations of carbon dioxide and methane, the two major greenhouse gases, from space." Not true: many satellites/instruments measured both in the thermal IR. Also SCIAMACHY, which came out much earlier, measured both in the near-IR. What you could say is that GOSAT was the first spacecraft dedicated solely to measuring CO2 and CH4.

Response: The comment is well taken and included on page 5 line 1.

D. Baker's comments: p5 L15: When discussing the OCO-2 data, you must say what version of retrieval you are using: version 7? Also, you should indicate whether you are looking at raw XCO2 values or bias-corrected values. If bias corrected, you should indicate whether you used the additional "s31" bias correction, designed to correct albedo-related errors not caught in the initial bias correction. This extra bias correction could be particularly important over Africa, with its large differences between desert and tropical rainforest. Also, you should discuss whether you sample your model with the OCO-2 averaging kernel and prior CO2 profile when comparing to the OCO-2 retrievals.

Response: This was also a comment from the anonymous reviewer. So, it has already been corrected. The version of OCO-2 (i.e., OCO-2 V7 lite level 2) used is biased-corrected. However, we have not used albedo bias correction. We have smoothed the CT XCO2 using averaging kernel and a priori profiles with respect both satellite data sets following your previous comments and that of anonymous reviewer's comments (see also one of the previous responses).

D. Baker's comments: p5 L21: (Methods): the satellites estimate both a profile of CO2 on 20 levels, as well as a vertical weighted average, "XCO2", computed from these. Which do you compare to? If the latter, you should discuss the vertical weighting kernel for XCO2 and how you sample your model to account for this. This methods section seems to be the correct place to discuss this.

Response: We used the column average XCO2 from the satellites and compared it to the XCO2 computed from the model CO2 profile. This CO2 profile is extracted and interpolated to resolution and vertical level of the satellites. Then following the procedures described in Coner et al., (2008) we transform the model CO2 profile to model averaged values (XCO2). Brief discussion and mathematical expression of the technique are included in section 2.4 of the revised manuscript.

D. Baker's comments: p6 L19: "Fig. 1 shows the five-years average of CT2016 (Fig. 1a) and GOSAT (Fig. 1b) XCO2 distribution." It would have been more useful to do the 5-year average over a true 5-year span, rather than the "April 2009 to June 2014" span that you used. As it stands, you have an extra 3 month span (April to June) that is unbalanced by the remainder of an annual cycle. As a result, this three month span throws off what otherwise would be the average of five full annual cycles. If flux is positive north of the equator during April to June (which it probably is, this being right at the end of the burn season there) then you tend to see positive values there in Figure 1a and 1b. This is due to the seasonality of the flux rather than the annual

mean value. It might be easier to discuss annual mean fluxes in the text if these plots reflected averages over pure annual cycles.

Response: We agree that the extra 3 months may destroy the symmetry of the annual cycles over Africa. In the revised manuscript we change the study period to span only 5 years (from May 2009 to April 2014).

D. Baker's comments: p7 L3: "respiration from forest in the region is overestimated" Why do you think that this is not due to photosynthesis being underestimated?

Response: It is possible that both or one of the two can cause the observed discrepancy. It might be equally possible cloud contamination may affect the satellite XCO2 such that the satellite estimates are unusually high (O'Dell et al., 2012). The equatorial region, where the difference is observed, is known to high altitude thin cirrus clouds. The revised manuscript was edited to reflect these possibilities on page 8 lines 18-23.

D. Baker's comments: p7 last line: "standard deviation of 1.31 ppm indicating better consistency and less potential outliers." Better than what?

Response: In the revised manuscript this standard deviation was only 0.98 and the statement was reworded as "standard deviation of 0.98 ppm indicating good regional consistency and low potential outliers" on page 8 lines 30-31.

D. Baker's comments: p8 L3: change "require" to "permit" Also, if the difference can be up to 1.5 degrees on either side, then we are talking about a box of 3 degrees on a side in both latitude and longitude.

Response: Accepted. Change is made on page 8 line 33.

D. Baker's comments: On p5 L29 you should then change the box from being 1.5 degrees long and wide to 3 degrees long and wide.

Response: Accepted. Change is made on page 8 line 33.

D. Baker's comments: Fig 3 caption: please provide units (ppm)

Response: The unit is now indicated as the title of the color bars in the revised manuscript.

D. Baker's comments: p8 L19: A correlation should be unitless – remove "ppm"

Response: Removed

D. Baker's comments: Figure 3: It is not clear to me why Figures 3a and 1c are different – they are both showing the difference (or bias) between CT2016 and GOSAT on an annual mean basis. So why aren't they identical?

Response: We agree that the mean difference and the bias should be the same. But Fig.1c is based on the full coincident data sets while in Fig.3a, grid points with less than 10 data points are removed to establish reliable statistics. Although such restriction is not a requirement for simple difference which is temporally averaged in Fig.1c, we have now excluded those points in Fig.1c to avoid similar misunderstanding in the revised manuscript.

D. Baker's comments: p8 L20: Good – here you are examining whether some of the difference might be due to the satellite data rather than the model.

Response: Appreciated the complement and changes in other parts of the manuscript have similar tone in the revised manuscript.

D. Baker's comments: Section 3.2: The statistics presented here are abstruse and require the reader to go back to the other references not only for what the statistics mean, but even what the acronyms stand for. It is not clear what the message is that these statistics convey. The only point I gleaned from the discussion was that the annual mean bias between model and data is higher where there are fewer data to average over – that is a point that could be made with Figure 1 alone. I would suggest deleting this section altogether.

Response: As the statistical tools used in this section are uncommon in our science community, it is not at all surprising to see more inquiry on the tools. We have clearly

indicated why it is important by citing specific application on page 6 of this author response and in the manuscript in Section 2.4 as well as in the discussion section from page 13 lines 2-12. Moreover, amendment in different parts of the manuscript will now provide more information that clarify the importance of the statistical tools (see also the author response to anonymous reviewer).

D. Baker's comments: p13 L9, p14 L11-14: Here you are suggesting that local fluxes are responsible for the changes in CO2 seen over Africa, but it is quite possible that a large part of these changes are due to fluxes elsewhere, over different continents, blowing downwind over Africa. Without inversion results linking concentrations to the fluxes that caused them, your argument will remain weak using this approach.

Response: We have taken note of your suggestion and included it as one of possible scenarios in the revised manuscript on page 18 lines 1-3.

D. Baker's comments: p25 L16: "Satellite observations are able to capture the CO2 concentration truly and objectively." I don't understand how you can say this based on the results that you have shown. You have shown large differences between Carbon-Tracker and the satellite data, on the order of 5 ppm on a seasonal and regional basis. This could be due to satellite retrieval problems in addition to the model problems that have been emphasized here. You haven't shown any data or comparisons that suggest the satellite data are better than the model, really. Maybe you could compare GOSAT and OCO-2 across the short time period that they overlap (late 2014, 2015) and if they agree closely then suggest they are both getting the right answer. From the plots you have showed in this manuscript, however, it looks like there are significant differences between OCO-2 and GOSAT... suggesting that the satellite errors are significant.

Response: We have accepted that we are a bit biased in favor of satellites. We removed the statement. However, since we do not have overlap between the two satellites (GOSAT and OCO-2) in the version used in this study, we did not compare them.

References: 1. Chevallier, F.; Feng, L.; Bösch, H.; Palmer, P.I.; Rayner, P.J. On the

impact of transport model errors for the estimation of CO 2 surface fluxes from GOSAT observations. Geophys. Res. Lett. 2010, 37. 2. Connor, B. J., Boesch, H., Toon, G., Sen, B., Miller, C., and Crisp, D.: Orbiting Carbon Observatory: Inverse method and prospective error analysis, Journal of Geophysical Research: Atmospheres, 113, 2008. 3. Deng, A., Yu, T., Cheng, T., Gu, X., Zheng, F., and Guo, H.: Intercomparison of Carbon Dioxide Products Retrieved from GOSAT Short-Wavelength Infrared Spectra for Three Years (2010–2012), Atmosphere, 7, 109, 2016. 4. Feng, L.; Palmer, P.I.; Yang, Y.; Yantosca, R.M.; Kawa, S.R.; Paris, J.D.; Matsueda, H.; Machida, T. Evaluating a 3-D transport model of atmospheric CO 2 using ground-based, aircraft, and space-borne data. Atmos. Chem. Phys. 2011, 11, 2789–2803. 5. Gurney, K. R., Law, R. M., Denning, A. S., Rayner, P. J., Baker, D., Bousquet, P., Bruhwiler, L., Chen, Y.-H., Ciais, P., Fan, S., et al.: Towards robust regional estimates of CO2 sources and sinks using atmospheric transport models, Nature, 415, 626–630, 2002.

6. Hungershoefer, K., Breon, F.-M., Peylin, P., Chevallier, F., Rayner, P., Klonecki, A., Houweling, S., and Marshall, J.: Evaluation of various observing systems for the global monitoring of CO 2 surface fluxes, Atmospheric chemistry and physics, 10, 10 503–10 520, 2010. 7. Jing, Y., Wang, T., Zhang, P., Chen, L., Xu, N., and Ma, Y.: Global Atmospheric CO2 Concentrations Simulated by GEOS-Chem: Comparison with GOSAT, Carbon Tracker and Ground-Based Measurements, Atmosphere, 9, 175, 2018. 8. O'Dell, C., Connor, B., Bösch, H., O'Brien, D., Frankenberg, C., Castano, R., Christi, M., Eldering, A., Fisher, B., Gunson, M., et al.: The ACOS CO 2 retrieval algorithm-Part 1: Description and validation against synthetic observations, 2012.

9. Mengistu Tsidu, G.: High-resolution monthly rainfall database for Ethiopia: Homogenization, reconstruction, and gridding, Journal of Climate,25 , 8422–8443, 2012. 10. Nagarajan, B. and Aiyyer, A. R.: Performance of the ECMWF operational analyses over the tropical Indian Ocean, Monthly weather review, 132, 2275–2282, 2004. 11. Rodgers, C. D. and Connor, B. J.: Intercomparison of remote sounding instruments, Journal of Geophysical Research: Atmospheres, 108, 2003.

12. Yoshida, Y., Kikuchi, N., Morino, I., Uchino, O., Oshchepkov, S., Bril, A., Saeki, T., Schutgens, N., Toon, G., Wunch, D., et al.: Improvement of the retrieval algorithm for GOSAT SWIR XCO2 and XCH4 and their validation using TCCON data, 2013.

13. Williams, C. A., Hanan, N. P., Neff, J. C., Scholes, R. J., Berry, J. A., Denning, A. S., and Baker, D. F.: Africa and the global carbon cycle, Carbon balance and management, 2, 3, 2007.

Please also note the supplement to this comment:
https://www.atmos-meas-tech-discuss.net/amt-2018-84/amt-2018-84-AC4-supplement.pdf

**Supplement:**

**Comparison of $CO_2$ from NOAA Carbon Tracker reanalysis model and satellites over Africa**

Anteneh Getachew Mengistu [1] and Gizaw Mengistu Tsidu[1,2]

[1]Addis Ababa University, Addis Ababa, Ethiopia
[2]Botswana International University of Science and Technology, Palapye, Botswana

**Correspondence:** Anteneh G (antenehgetachew7@gmail.com)

**Abstract.** The scarcity of ground-based observations, poor spatial and temporal resolution of satellite observations necessitate the use of data generated from models to assess spatio-temporal variations of atmospheric $CO_2$ concentrations in a near continuous manner in a global and regional scale. Africa is one of the most data scarce region as satellite observation at the equator is limited by cloud cover and there are very limited number of ground based measurements. As a result, use of simulations from models are mandatory to fill this data gap. However, the first step before the use of data from models requires assessment of model skill in simulating limited existing observations reasonably well. Even though, the NOAA Carbon Tracker model is evaluated using TCCON and satellite observations at a global level, its performance should be assessed at a regional scale, specifically in a regions like Africa with a highly varying climatic responses and a growing number of local sources. In this study, NOAA CT2016 $XCO_2$ is compared with the GOSAT observation over Africa using five years data covering the period from May 2009 to April 2014. In addition, NOAA CT16NRT17 $XCO_2$ is compared with OCO-2 observation over Africa using two years data covering the period from January 2015 to December 2016. The analysis shows that the $XCO_2$ simulated from CT2016 is lower than GOSAT retrievals by 0.28 ppm whereas CT16NRT17 $XCO_2$ is higher than OCO-2 retrievals by 0.34 ppm on African landmass on average, which are within the mean $XCO_2$ posterior errors of 0.91 and 0.55 ppm associated with the GOSAT and OCO-2 $XCO_2$ retrievals respectively. The mean correlations of 0.83 and 0.60, regional precisions of 2.30 and 2.57 ppm, and the relative accuracies of 1.05 and 1.21 ppm are found between the model and the two data sets implying the existence of reasonably good agreement between CT and the two satellites over Africa's land region given that very limited number of in-situ observations over the region is assimilated in the model. These differences, however, exhibit spatial and seasonal scale variations. Moreover, the model shows some differences with the observations at extreme ends of $XCO_2$ distribution. For example, the quantile probability of detection ranges from 0.42 to 0.99 and quantile critical success index ranges from 0.37 to 0.93 over the continent when the analysis includes data above the $90^{th}$ and $5^{th}$ percentiles respectively (see main text for definitions). This shows that the model disagrees with observations on 58% of $XCO_2$ with values higher than $90^{th}$ percentile. This discrepancy increases to approximately 80% at $95^{th}$ percentile indicating the agreement between CT and the satellites deteriorates towards the high extreme ends of the $XCO_2$ distribution. Spatially, OCO-2 $XCO_2$ are lower than that of CT16NRT17 by upto 3 ppm over some regions in North Africa (e.g., Egypt, Libya and Mali ) whereas it exceeds CT16NRT17 $XCO_2$ by 2 ppm over Equatorial Africa $(10\,^0 S - 10\,^0 N)$. CT2016 shows high spatial mean of seasonal mean RMSD of 1.91 ppm during DJF with respect to GOSAT while CT16NRT17 shows 1.75 ppm during

MAM with respect to OCO-2. On the other hand, low RMSD of 1.00 and 1.07 ppm during SON in the model $XCO_2$ with respect to GOSAT and OCO-2 are determined respectively indicating better agreement during autumn. The model simulation and satellite observations exhibit similar seasonal cycles of $XCO_2$ with a small discrepancy over the Southern Africa and during wet seasons over all regions.

**1  Introduction**

An understanding of the regional contributions and trends of carbon dioxide ($CO_2$) is critical to design mitigation strategies aimed at stabilizing atmospheric greenhouse gases. The present-day concentration of atmospheric $CO_2$ continues to rise. Ongoing emissions of $CO_2$ and other greenhouse gases will influence the global climate system during the next decades and centuries. Approximately one-half of the $CO_2$ emissions from human activities is accumulating in the Earth's atmosphere, whereas the remaining portion of emitted $CO_2$ is absorbed by sink on land and in the ocean (Raupach et al., 2007).

Several studies (e.g., Tsutsumi et al., 2009; Stocker, 2014; Allison, 2015) have shown that the column averaged dry air mole fractions of $CO_2$ ($XCO_2$) has undergone rapid changes from pre-industrial period value of 280 ppm to 396 ppm as recently as 2013. This change has been attributed to anthropogenic factors such as fossil fuel combustion, land use change, biomass burning, emission from industries such as cement. Notably, this value exceeds the highest $XCO_2$ level retrieved from ice cores representing the past 800,000 years. This increasing trend has been persistent. For example according to the World Meteorological Organization (WMO) 2016 report, the average concentrations of $CO_2$ hit 403.3 ppm, up from 400 ppm in 2015 surpassing an annual positive increasing rate of $1.99 \pm 0.43\ ppmyr^{-1}$ (http://www.esrl.noaa. gov/gmd/ccgg/trends/global.html Houghton, 2007). These positive trends have led to imbalance of $0.58 \pm 0.15\ Wm^2$ in energy budget between 2005 and 2010 at the top of the atmosphere (Hansen et al., 2011). To this end, changes in atmospheric temperature, hydrology, sea ice, and sea levels are attributed to climate forcing agents dominated by $CO_2$ (Santer et al., 2013; Stocker et al., 2013). However, understanding the climate response to anthropogenic forcing in a more traceable manner is still difficult due to a major uncertainty in carbon-climate feedbacks (Friedlingstein et al., 2006). Part of this uncertainty is due to lack of sufficient data on regional and global carbon cycle. This is compounded with inappropriate modelling practices to capture spatio-temporal variability of carbon cycle. These problems can be solved through strengthening carbon monitoring networks and setting up proper modelling. A transport model, with appropriate physical and mathematical formulations and sufficiently tuned by observations, can be used to understand the spatio-temporal nature of atmospheric $CO_2$ source and sink as well its associated drivers.

[revised manuscript text omitted]

**2.3 OCO-2 measurements**

OCO-2, the second world's full-time dedicated $CO_2$ measurement satellite, was successfully launched by National Aeronautics and Space Administration (NASA) on 2 July 2014. OCO-2 measures atmospheric carbon dioxide with the accuracy, resolution, and coverage required to detect $CO_2$ source and sink on global and regional scale. OCO-2 has three-band spectrometer, which measures reflected sunlight in three separate bands. The $O_2$ A-band measures molecular absorption of oxygen from reflected sunlight near 0.76 $\mu m$ while the $CO_2$ bands are located near 1.61 $\mu m$ and 2.06 $\mu m$ (Liang et al., 2017). In this study, bias corrected OCO-2 $XCO_2$ V7 lite level 2 covering the period from January 2015 to December 2016, here after referred to as OCO-2 $XCO_2$ are used. Due to scarcity of data, CT values from the two releases CT2016 for the year 2015 and CT-NRT.v2017 for the year 2016, here after (CT16NRT17) are employed in this study. The OCO-2 project team at Jet Propulsion Laboratory, California Institute of Technology, produced the OCO-2 $XCO_2$ data used in this study. The data can be accessed from NASA Goddard Earth Science Data and Information Service Center.

**2.4 Methods**

The GOSAT and CT model $XCO_2$ time series used in this investigation spans five years, ranging from May 2009 to April 2014. Atmospheric $CO_2$ concentrations of NOAA Carbon-Tracker have a global coverage with a $3^0 \times 2^0$ Longitude/Latitude resolution which covers 428 grid points in our study area. Satellite observations, however, is different from model assimilation, and have gaps because of various reasons (e.g., cloud and the observational mode of satellite). As a result, there is no one to one spatio-temporal match between the two data sets. For example, $CO_2$ products from the two dataset are not directly comparable since CT is a 3 hourly smooth and regular grid dataset whereas GOSAT $XCO_2$ is irregularly distributed in space and time.

Thus, the CT $CO_2$ is extracted on the time and location of GOSAT $XCO_2$ data. Using the grid point of CT as a reference bin, the corresponding GOSAT $XCO_2$ found with in a rectangle of $1.5^0 \times 1.5^0$ with centre at the reference bin and with temporal mismatch of a maximum of 3 hrs is extracted. Moreover, CT has higher vertical resolutions than GOSAT. As a result, the two can not be directly compared. It is customary to smooth the high resolution data (in this case CT) with averaging kernels and a priori profiles of the low resolution satellite measurements (in this case GOSAT). In addition, due to a difference between CT and GOSAT on the number vertical levels, CT $CO_2$ is interpolated to vertical levels of GOSAT. The CT $XCO_2$ ($XCO_2^{model}$) used in the comparison is computed from the interpolated CT $CO_2$ ($CO_2^{interp}$), pressure weighting function ($w$), $XCO_2$ a priori ($XCO_{2a}$), column averaging kernel of the satellites retrievals ($A$) and a priori profile ($CO_{2a}$) of the retrievals as per procedure discussed by Rodgers and Connor (2003); Connor et al. (2008); O'Dell et al. (2012); Chevallier (2015); Jing et al. (2018) and given as:

$$XCO_2^{model} = XCO_{2a} + \sum_i w_i A_i * (CO_2^{interp} - CO_{2a})_i \tag{1}$$

where $i$ is the index of the satellite retrieval vertical level.

Correlation coefficients (R), bias and root mean square deviation (RMSD) are used to assess the level of agreement between the two data sets. The mean bias determines the average deviations in $XCO_2$ between Carbon Tracker simulation and satellite observations. In this work the bias at the $j^{th}$ grid point is computed as:

$$Bias_j = \frac{1}{n} \sum_{i=1}^{n} (S_i - O_i) \tag{2}$$

where $S_i$ and $O_i$ are CT and GOSAT $XCO_2$ values over the $j^{th}$ pixel at the $i^{th}$ time respectively. To quantify the extent to which $XCO_2$ of CT and GOSAT agree, the pattern correlations at the $j^{th}$ grid point are computed as:

$$R_j = \frac{\frac{1}{n} \sum_{i=1}^{n} (S_i - \bar{S})(O_i - \bar{O})}{\sqrt{\frac{1}{n} \sum_{i=1}^{n} (S_i - \bar{S})^2} \sqrt{\frac{1}{n} \sum_{i=1}^{n} (O_i - \bar{O})^2}} \tag{3}$$

where $\bar{S}$ and $\bar{O}$ are the mean values of $S_i$ and $O_i$ over the $j^{th}$ pixel. The root mean square deviation (RMSD) which shows the standard error of the model with respect the observation at the $j^{th}$ grid point is computed as:

$$RMSD_j = \sqrt{\frac{1}{n} \sum_{i=1}^{n} ((S_i - \bar{S}) - (O_i - \bar{O}))^2} \tag{4}$$

This study also applies categorical contingency table for evaluation of performance of CT2016 in capturing the different parts of observed $XCO_2$ distribution. $XCO_2$ is a continuous physical quantity for which categorical metrics are not applicable. In such cases, scatter plots are used as a means of visual inspection of the model skill with no quantitative information in terms of spatial distribution and magnitude of the scatter apart from a single standard deviation. Recently, extended categorical contingency table is proposed to overcome this drawback and assess whether a model simulation/satellite retrieval can capture or fail to capture observed physical quantity exceeding a specified quantile threshold. This procedure effectively reduces the

continuous physical quantity with two outcomes i.e., yes (capture) or no (fail). In this study, the skill of CT in correctly simulating whether the observed $XCO_2$ values are above a selected threshold will be determined using the categorical metrics. The categorical metrics includes Quantile Bias (QBias), Quantile Probability of Detection (QPOD), Quantile False Alarm Ratio (QFAR), Quantile Critical Success Index (QCSI) also known as the Threat Score and quantile Categorical miss (QMISS). The

5  QBias is defined as the ratio of number of observations (satellite data, OBS) to simulations (Carbon Tracker, SIM) above a certain threshold. Mathematically, QPOD, QFAR, QMISS and QCSI are defined as

$$QPOD = \frac{NoHit}{NoMiss + NoHit} \tag{5}$$

$$QMISS = \frac{NoMiss}{NoMiss + NoHit} \tag{6}$$

$$QFAR = \frac{NoFalse}{NoHit + NoFalse} \tag{7}$$

and

$$QCSI = \frac{NoHit}{(NoHit + NoFalse + NoMiss)} \tag{8}$$

where $NoHit = \sum_i^n (SIM_i | (SIM_i > t \& OBS_i > t))$ is the number of data detected by both simulation ($SIM$) and ob-

15  servation ($OBS$) above a threshold $t$, $NoMiss = \sum_i^n (OBS_i | (SIM_i \leq t \& OBS_i > t))$ is the number of data detected by observation but missed by simulation and $NoFalse = \sum_i^n (SIM_i | (SIM_i > t \& OBS_i \leq t))$ refers to number of data available only from model above the threshold $t$ and n is the number of exceedances.

QPOD quantifies the fraction of reference observations (in this case GOSAT observations) detected correctly by the simulation above a selected threshold. Meaningful values of QPOD ranges from 0 (zero agreement) to 1 (perfect agreement); QFAR

20  identifies the fraction of events captured by simulation but not available in reference observations above the threshold. Sound values of QFAR is bounded by 0 to 1 with 0 implying perfect score. QCSI combines different aspects of the QPOD and QFAR to characterize the overall performance of the simulation in capturing observation. The QCSI is constrained to have values between 0 (zero agreement) to 1 (perfect agreement) by definition. QMISS quantifies events identified by reference observation but missed by the simulation. Therefore, by definition, quantile categorical miss ranges from 0 (perfect score) to 1 (zero

25  agreement). More details about these categorical statistical metrics can be found in works by other authors (e.g., AghaKouchak et al., 2011; Wilks, 2011; AghaKouchak and Mehran, 2013, and references therein). Using similar coincidence criteria and statistical methods, CT16NRT17 and OCO-2 $XCO_2$ are also compared.

**3 Results and discussions**

**3.1 Comparison of $XCO_2$ mean climatology from NOAA CT2016 and GOSAT**

The mole fraction of $CO_2$ obtained from the NOAA Carbon Tracker model and GOSAT observation was compared. The results are based on 426 grid pints uniformly distributed to cover the whole Africa's land region. The analysis was based on five years daily data starting from May 2009 to April 2014. The $XCO_2$ comparison was done only when there are more than ten $XCO_2$ retrievals that fulfils the spatio-temporal matching criteria defined in Section 2.4.

Fig. 1 shows temporal average of CT2016 (Fig. 1a) and GOSAT (Fig. 1b) $XCO_2$ distribution. The major common spatial feature in the mean map of $XCO_2$ from GOSAT and CT2016 reanalysis is dipole structure characterized by high $XCO_2$ northward of equator and low $XCO_2$ southward of equator with the exception of Congo basin which is characterized by spatially anomalous high $XCO_2$. The Southern Africa region is characterized by weak anthropogenic $CO_2$ emission and high $CO_2$ uptake by the vegetation. This contributed to the observed dipole distribution. Another important pattern is anomalous peak over annual average location of Inter-tropical convergence zone (ITCZ) (Fig. 1b) which appears to fade over Eastern Africa. This is in agreement with fact that carbon stocks and net primary production per unit land area are higher over Equatorial Africa and decrease towards northward and southward of the equator over arid environments (Williams et al., 2007). However, Fig. 1a shows that CT2016 has some limitations in simulating this spatial pattern in comparison to GOSAT.

Fig. 1c shows mean difference (CT2016–GOSAT) $XCO_2$ which ranges from -4 to 2 ppm. The highest difference between the CT2016 and GOSAT $XCO_2$ (as high as -4 ppm) is observed over Equatorial Africa, western Ethiopia and South Sudan which are also known for near-year-round rainfall and relatively dense vegetation. The regions are known for their rain forest. The likely explanation could be $CO_2$ flux from respiration (photosynthesis) of forest in the region which is underestimated (overestimated) in the reanalysis. However, the mean (over five years) may also be slightly positively biased due to fewer observations as shown in Fig.1d. The strategy and methods for cloud screening in GOSAT retrievals could lead to smaller number of observation in the equatorial region (Crisp et al., 2012; O'Dell et al., 2012; Yoshida et al., 2013; Chevallier, 2015; Deng et al., 2016b). The number of datasets used for comparison range from 14 to 4288 from grid to grid with a spatial mean of 1109 data over the continent. Fig. 1c also shows CT2016 simulations are overall lower than the values of GOSAT observation over most regions with an exception in Gabon, Congo, southern Kenya and southern Tanzania where CT2016 simulations are higher than GOSAT observation by more than 1 ppm. The spatial distribution of global atmospheric $CO_2$ is not uniform because of the irregularly distributed sources of $CO_2$ emissions, such as large power plant and forest fire, and biospherical assimilation as clearly noted above.

Fig. 2a shows the histograms of differences of CT2016 and GOSAT $XCO_2$. The mean difference between CT2016 and GOSAT means is about -0.27 ppm with the standard deviation of 0.98 ppm indicating good regional consistency and low potential outliers. Moreover, a negative mean of the difference implies that $XCO_2$ simulated from CT2016 is lower than that of GOSAT retrievals over Africa land mass.

Because of selection criteria which permits a difference of 3 degree long and wide, the two datasets are not exactly at the same point. The impact of the relative distance between them should be assessed before performing any statistical comparison.

[Figure]

**Figure 1.** Distribution of five-years averages of CT2016 **(a)** and GOSAT **(b)** $XCO_2$ and their difference **(c)** gridded in $3^0 \times 2^0$ bins over Africa's Land mass; and the total number of datasets at each grid (d).

Fig. 2b depicted color-coded scatter plot of CT2016 model simulation verses GOSAT to determine if the discrepancy between the data sets arise from spatial mismatch. The color code indicates the relative distance between the model and observation datasets. For these datasets the $50^{th}$ percentile has a relative distance of $1.19^0$ which means 50% of the data has a relative distance shorter than $1.19^0$. The maximum relative distance between them is $2.12^0$. However, there is no indication that this

5   has been the case since the scatter is not a function of relative distance between the data sets. For example, data points with blue color with lowest location difference is scattered everywhere instead of along the 1:1 line. Furthermore, we found the bias of -0.26 ppm, correlation coefficient of 0.86 and RMSD of 2.19 ppm for datasets which has a relative distance shorter than $1.19^0$. On the other hand, the bias , correlation coefficient and RMSD are -0.33 ppm, 0.86 and 2.22 ppm for those which are longer than $1.19^0$. The above statistics was performed merely to test the influence of location mismatch.

10     Fig. 3 shows a statistical comparison of $XCO_2$ from the CT2016 and GOSAT over Africa. The number of data used in this comparison are shown in Fig. 1d. As it is depicted in Fig. 3a, the bias ranges from -4 to 2 ppm with a mean bias of -0.28 ppm (see Table 1). A larger negative bias of about -2 ppm was found along the annual mean position of ITCZ. The correlation varies from 0.4 over some isolated pockets in Congo, Tanzania, Mozambique, Uganda and western Ethiopia to 0.9 over northern half of Africa northward of $13^0 N$, Eastern Ethiopia and Kalahari Desert. Fig. 3b depicts correlation coefficient between GOSAT

15   and Carbon Tracker $XCO_2$. The region with poor correlation also exhibits high RMSD as shown in Fig. 3c. To understand whether this discrepancy originates from model weakness alone, we have looked at the GOSAT posteriori estimate of $XCO_2$ error, which are high over the same regions with high bias and RMSD between GOSAT and Carbon Tracker $XCO_2$ (Fig. 3d).

[Figure]

**Figure 2.** Histogram of the difference of CT2016 relative to GOSAT (left panel) and color code scatter diagram of $XCO_2$ concentration as derived from CT2016 and GOSAT (right panel). Color indicates the relative distance as shown in colorbar between datasets.

GOSAT's posteriori estimate of $XCO_2$ error is a combination of instrument noise, smoothing error and interference errors (Connor et al., 2008; O'Dell et al., 2012). This posteriori estimate of $XCO_2$ error does not include forward model error which may lead to underestimation of the true error of satellite $XCO_2$ by a factor of two (O'Dell et al., 2012). Therefore, part of the discrepancy is clearly linked to satellite own uncertainty, which might have been amplified due to small number of data points used to calculate the mean error of GOSAT $XCO_2$ measurements (see Fig. 1d). In general, the two data sets are characterized by high spatial mean correlation of 0.83, a global offset of -0.28 ppm, which is the average bias, a regional precision of 2.30 ppm, and a relative accuracy of 1.05 ppm as depicted in Table 1.

**Table 1.** Summary of statistical relation between CT2016 and GOSAT observation. The statistical tools shown are the mean correlation coefficient (R), the spatial average of bias (Bias), the spatial average root mean square deviation (RMSD), the standard deviation in bias (std of Bias), GOSAT posteriori estimate of $XCO_2$ error (GOSAT err), the standard deviation in CT2016 $XCO_2$ (CT2016 std) and the standard deviation in GOSAT $XCO_2$ (GOSAT std). The number of data used in the statistics is 472,821 over 426 pixels covering the study period; distribution at each grid point is shown in Fig. 1d. Negative bias indicates that CT2016 $XCO_2$ is lower than GOSAT $XCO_2$ values.

| Statistical tool | R | Bias (ppm) | RMSD (ppm) | std of Bias (ppm) | GOSAT err (ppm) | CT2016 std (ppm) | GOSAT std(ppm) |
|---|---|---|---|---|---|---|---|
| Values | 0.83 | -0.28 | 2.30 | 1.05 | 0.91 | 0.90 | 1.55 |

[Figure]

**Figure 3.** Spatial patterns of bias (a), correlation (b), RMSD (c) of the two data sets, and mean posteriori estimate of $XCO_2$ error from GOSAT (d).

**3.2 Categorical comparison of $XCO_2$ from NOAA CT and GOSAT**

Following the methods described in AghaKouchak et al. (2011), QBias, QPOD, QCSI, QMISS and QFAR are determined from the coincident $XCO_2$ data to assess how Carbon Tracker model and satellite observations perform with respect each other in capturing different parts of $XCO_2$ distribution. It is worth noting that these categorical metrics are used to evaluate the level of agreement between CT2016 and GOSAT $XCO_2$ in certain part of $XCO_2$ distribution (e.g., above a given quantile threshold). In this way, $XCO_2$ distribution is effectively rendered to be dichotomous variable for which we can use the categorical metrics.

Fig. 4 displays values for QBias, QPOD, QCSI, QMISS and QFAR for distribution exceeding 5% (first row), 75% (second row), 90% (third row) and 95% (fourth row) quantiles. We filter out pixels in which the total number of observations are less than 10 to avoid unreliable statistics. The thresholds are set based on the quantiles of the GOSAT observation. QPOD and QCSI decrease at higher quantiles. In contrast, QFAR and QMISS increase at higher quantiles. Specially the decrease in QCSI is significant. It ranges from 0.8 to 1.0 at $5^{th}$ percentile ( i.e., QCSI for values exceeding the $5^{th}$ percentile) and smaller than 0.4 at $90^{th}$ percentile.

On the other hand, the QMISS which is below 0.07 at threshold of $5^{th}$ percentile shows a value above 0.58 at threshold value of the $90^{th}$ percentile for most of the regions (Fig. 4 ). This indicates that the agreement between CT2016 and GOSAT $XCO_2$ deteriorates as the comparison data range includes only higher extremes of the $XCO_2$ distribution. For example, on average over the continent at $95^{th}$ percentile the QFAR is lower than 0.24 indicating that 24% (see also Table 2) of datasets

[Figure]

**Figure 4.** Distribution of categorical metrics over the study area for quantiles exceeding 5% (first row), 75% (second row), 90% (third row) and 95% (fourth row)

[Figure]

**Figure 5.** Summary of categorical metrics (QBias, QPOD, QCSI, QFAR and QMISS) averaged over Africa's land region for 5, 10, 25, 50, 75, 90 and 95 percentile.

simulated by the model to be above $95^{th}$ percentile threshold were not actually in the observation. This indicates that a notable difference between CT2016 and GOSAT $XCO_2$ exists at the tails of the distribution. The information from these statistics are crucial for modelers and/or scientists working on satellite remote sensing to improve the model and/or the retrieval strategy. For example, in satellite retrievals, smoothing constraint or regularization (e.g., Tikhonov first or second order, the so called

5   shape constraint) heavily penalizes the portion of the profile at low and high extremes thereby restricting the possibility of observing profiles unusually different from the a priori profile. It is worth-noting that the a priori profile is usually constructed from climatology of $CO_2$ profile. In some case, only a handful of a priori profile represents a region (e.g., tropical model atmosphere, mid-latitude model atmosphere, polar model atmosphere) with wide range of variability. This could create huge discrepancy between the model and satellite observations in the tail region of the $XCO_2$ distribution. The observed difference

10   between CT and GOSAT may well be partly attributed to such factors. However, when the data covers lower extremes, QPOD and QCSI have substantially improved indicating the existence of better agreement between CT2016 and GOSAT at the lower end of the $XCO_2$ distribution.

    Fig. 4a shows the QPOD and QCSI are above 0.93 for lower quantiles which indicates that the model simulates above 93% of the observations (see also Table 2). However, at higher quantiles the change in contingency metrics show a clear deviation

15   between the model simulation and observation. Figs. 4b show the scarce datasets in Equatorial Africa which, to certain extent, is the main reason for large biases observed around the Equator in Fig. 3a and also for the corresponding posteriori retrieval error in GOSAT $XCO_2$ in Fig. 3d. Note that individual posteriori retrieval errors are smoothed out during averaging over large number of coincident observations. Fig. 4 shows that QBias is 1 at lower quantiles and decreases with increasing quantiles implying that the number of CT2016 data that matches the GOSAT observation decreases with increasing threshold. Fig. 4b

**Table 2.** Summary of extended contingency metrics for the relation between CT2016 model simulation and GOSAT observation.

| Quantiles | QBias | QPOD | QCSI | QMISS | QFAR | threshold (ppm) | Number of grid points |
|---|---|---|---|---|---|---|---|
| Q5 | 1.064 | 0.993 | 0.93 | 0.007 | 0.064 | 384.32 | 424 |
| Q10 | 1.073 | 0.976 | 0.896 | 0.024 | 0.084 | 385.73 | 421 |
| Q25 | 1.016 | 0.902 | 0.816 | 0.098 | 0.107 | 388.17 | 419 |
| Q50 | 0.936 | 0.801 | 0.711 | 0.199 | 0.128 | 391.27 | 405 |
| Q75 | 0.852 | 0.64 | 0.548 | 0.36 | 0.198 | 394.52 | 386 |
| Q90 | 0.551 | 0.418 | 0.361 | 0.582 | 0.202 | 397.31 | 284 |
| Q95 | 0.351 | 0.259 | 0.231 | 0.741 | 0.239 | 398.98 | 225 |

shows a spatial mean bias of 0.85 (see also Table 2). Fig. 4b also depicts that QCSI is lower than 0.5 and the QMISS is above 0.3 over most regions of Southern Africa. However, QCSI exceeds 0.6 over regions northward of $10^o N$. The results indicates that, the agreement between CT2016 and GOSAT $XCO_2$ shows a regional disparity which is better over Northern Africa than the Southern Africa. In general, the discrepancy between CT2016 and GOSAT $XCO_2$ is significant over the whole continent

5   towards the extreme high ends of the $XCO_2$ distribution.

In addition, QBias, QPOD, QCSI, QMISS and QFAR are calculated for all data, i.e., data that includes values higher than 5, 10, 25, 50, 75, 90 and 95 percentiles and averaged over the whole African land mass, as shown in Fig. 5. There is one major conclusion that can be drawn from Fig. 5, i.e., the QFAR and QMISS increase with increase in the quantile thresholds while QBias, QPOD and QCSI decrease.

10   **3.3   Comparison of monthly average time series of NOAA CT2016 and GOSAT $XCO_2$**

Africa is one of the largest continents covering both northern and southern hemispheres. As a result, the continent is under the influence of semi-permanent high pressure cells which led to the Sahara Desert in the North and the Kalahari in the South. The equatorial low pressure cell which allows formation of the seasonally migrating inter-tropical convergence zone is part of the major large scale atmospheric circulation systems. These large scale pressure systems, Oceanic circulations and their

15   interaction with the atmosphere coupled with diverse topographies of the region allow for the formation of different climates (e.g., equatorial, tropical wet, tropical dry, monsoon, semi desert (semi arid), desert (hyper arid), subtropical high climates). Geographically, the Sahel, a narrow steppe, is located just south of Sahara; the central part of the continent constitutes the largest rainforest next to Amazon whereas most southern areas contain savana plains. The continent get rainfall from migrating ITCZ, west Africa monsoon, intrusion of mid-latitude frontal systems, travelling low pressure systems (Mitchell, 2001, and

20   references therein). Since $CO_2$ fluxes exhibit seasonal variability and Africa experiences different seasons as noted above, it is important to divide Africa into three major regions, namely North Africa (10 to 35 $^0N$), Equatorial Africa (10 $^0S$ to 10 $^0N$), and Southern Africa (35 to 10 $^0S$) and conduct the comparison of the two $XCO_2$ datasets.

Figs. 6 - 8 show time series of $XCO_2$ monthly means covering the period from May 2009 to April 2014 for both CT2016 and GOSAT over North Africa, Equatorial Africa and Southern Africa respectively. Figs. 6a - 8a depict the existence of an

**Table 3.** Summary of statistical relation between CT2016 and GOSAT observation. The statistical analysis were made using monthly average of time series of 60 months (i.e., months from May 2009 to April 2014).

| Statistics | R | Bias (ppm) | RMSD (ppm) | number of data |
|---|---|---|---|---|
| Africa | 0.997 | -0.254 | 0.265 | 698505 |
| North Africa | 0.996 | -0.361 | 0.345 | 424070 |
| Equatorial Africa | 0.977 | -0.172 | 0.708 | 101660 |
| Southern Africa | 0.964 | 0.006 | 0.841 | 172775 |

[Figure]

**Figure 6.** The monthly mean time series of CT2016 and GOSAT from May 2009 to April 2014 averaged over North Africa (a), bias associated to the monthly means (b), the histogram of difference (c) and the annual growth rate obtained by subtracting the mean from the mean of the next year (d). The error bars in (a) shows the GOSAT a posteriori $XCO_2$ uncertainty.

overall very good agreement for the monthly averages with respect to amplitudes and phase of $XCO_2$. However, $XCO_2$ from the two data sets slightly disagree in capturing seasonal cycle over Southern Africa.

Fig. 6a shows that $XCO_2$ concentration reaches maximum (394.79 ppm) for CT2016 and (395.35 ppm) for GOSAT in April and minimum in September (388.66 ppm) for CT2016 and (388.75 ppm) for GOSAT over North Africa. Consistent with this

5    evidence, other authors (e.g., Zhou et al., 2008) have indicated presence of a strong absorption of $CO_2$ by vegetation during August in the northern hemisphere. This is also likely cause for minimum $XCO_2$ observed during September over North Africa. The largest monthly mean difference of -0.90 ppm between the two datasets is observed in June, while the smallest

[Figure]

**Figure 7.** The same as Fig. 6 but over Equatorial Africa.

value of -0.04 ppm is found in August (see also Table 4). In addition, both datasets show $XCO_2$ increases from October to April and decreases from May to September. Moreover, the two dataset shows a monthly mean regional mean bias of -0.36 ppm with a correlation of 1.0 and root mean square deviation of 0.36 ppm (see Table 3).

Fig. 7a shows $XCO_2$ concentration reaches maximum (392.99 ppm) for CT2016 in March and (393.53 ppm) for GOSAT in

5  January while minimum (389.56 ppm for CT2016 and 389.32 ppm for GOSAT) in October over Equatorial Africa. The largest monthly mean difference of -1.34 ppm and the smallest of -0.05 ppm between the two datasets observed in December in April respectively (Table 4). Moreover, both datasets show that $XCO_2$ increases from October to March while it decreases from June to October. This similarity in the seasonal variability of the two datasets shows that they are in good agreement in terms of amplitude and phase. In addition, the two data sets show a monthly average regional average bias of -0.17 ppm, correlation

10  of 0.98 and a small root mean square deviation 0.71 ppm over Equatorial Africa (see Table 3). Fig. 8a shows maximum $XCO_2$ concentration in April (391.04 ppm) for CT2016 and in October (391.28 ppm) for GOSAT, while minimum in May (389.30 ppm) for CT2016 and ( 388.46 ppm) for GOSAT over Southern Africa. The largest monthly mean difference of 1.53 ppm and 0.03 ppm between the two datasets is observed in April and in July (Table 4) respectively. Both datasets show concentration of $CO_2$ increases from May to July while it decreases from October to November. However, the $XCO_2$ from CT2016 shows

15  a gradual increasing trend from January to April. Conversely, GOSAT $XCO_2$ shows a decreasing trend. This is most likely

[Figure]

**Figure 8.** The same as Fig. 6 but over Southern Africa.

CT2016 simulation respond to the growing size of sinks following the rainy season. Moreover, the two data sets shows a monthly mean regional mean bias of 0.07 ppm, correlation of 0.97 and RMSD of 0.87 ppm over southern Africa (see Table 3).

Figs. 6b - 8b show regional averaged bias in the monthly mean $XCO_2$ time series between CT2016 and GOSAT covering the period of May 2009 to April 2014. Fig. 6b shows the presence of seasonally varying negative bias over North Africa. A
5  high (<-0.5 ppm) negative bias in dry seasons (April to June) and low (>=-0.1 ppm) negative bias in wet seasons (August to September) are observed. Moreover, the strength of bias increases from February to June. Conversely, the bias decreases from June to September. Similarly, Figs. 7b and 8b show seasonally fluctuating bias over Equatorial and Southern Africa regions. For example, Fig. 8b shows a positive bias from February to July and negative bias from August to December over Southern Africa.
10  Figs. 6c - 8c show the histogram of difference. The mean difference between CT2016 simulation and GOSAT observation of $XCO_2$ is -0.36 ppm with a standard deviation of 0.35 ppm over North Africa (see Fig. 6c); Fig. 7c presents a mean difference of -0.17 ppm with standard deviation of 0.71 ppm over Equatorial Africa; and Fig. 8c reveals a mean difference of 0.01 ppm and a standard deviation of 0.85 ppm which indicate that $XCO_2$ from CT2016 was slightly higher than that of GOSAT over Southern Africa on average. In addition, the low standard deviation of monthly mean difference over North Africa typically
15  indicates good regional consistency between CT2016 and GOSAT. This is mainly because Northern Africa is dominated by the Sahara desert which is known for its weak source/sink of $CO_2$. However, the spatial mean of monthly mean bias is slightly

**Table 4.** Five years monthly averaged $XCO_2$ concentration in ppm obtained from CT2016 (CT) and GOSAT (GO) and their difference $CT - GO$ (D) in ppm over Africa (A), North Africa (NA), Equatorial Africa(EA) and Southern Africa (SA).

| Month | A CT | A GO | A D | NA CT | NA GO | NA D | EA CT | EA GO | EA D | SA CT | SA GO | SA D |
|-------|------|------|-----|-------|-------|------|-------|-------|------|-------|-------|------|
| January | 391.81 | 392.17 | -0.36 | 392.43 | 392.61 | -0.18 | 392.22 | 393.53 | -1.31 | 390.28 | 390.49 | -0.21 |
| February | 392.48 | 392.58 | -0.1 | 393.27 | 393.5 | -0.23 | 392.72 | 393.21 | -0.49 | 390.52 | 390.06 | 0.46 |
| March | 393.25 | 393.28 | -0.03 | 394.02 | 394.29 | -0.27 | 392.99 | 393.19 | -0.2 | 390.82 | 389.81 | 1.01 |
| April | 393.81 | 393.91 | -0.1 | 394.79 | 395.35 | -0.56 | 392.87 | 392.92 | -0.05 | 391.04 | 389.51 | 1.53 |
| May | 391.65 | 391.85 | -0.21 | 392.92 | 393.73 | -0.81 | 390.47 | 389.93 | 0.54 | 389.3 | 388.46 | 0.84 |
| June | 391.49 | 391.94 | -0.45 | 392.43 | 393.33 | -0.9 | 391.12 | 390.89 | 0.23 | 389.95 | 389.85 | 0.11 |
| July | 390.92 | 391.1 | -0.18 | 391.09 | 391.5 | -0.41 | 391.44 | 391.03 | 0.41 | 390.43 | 390.4 | 0.03 |
| August | 389.89 | 389.96 | -0.07 | 389.4 | 389.44 | -0.04 | 390.92 | 390.72 | 0.21 | 390.37 | 390.61 | -0.25 |
| September | 389.26 | 389.4 | -0.14 | 388.65 | 388.75 | -0.1 | 390.02 | 389.67 | 0.35 | 390.39 | 391.01 | -0.61 |
| October | 389.19 | 389.71 | -0.51 | 388.85 | 389.26 | -0.41 | 389.56 | 389.32 | 0.24 | 389.95 | 391.28 | -1.32 |
| November | 389.97 | 390.43 | -0.46 | 390.06 | 390.32 | -0.26 | 389.86 | 390.52 | -0.66 | 389.8 | 390.76 | -0.96 |
| December | 391.09 | 391.53 | -0.45 | 391.42 | 391.6 | -0.18 | 391.23 | 392.57 | -1.34 | 389.98 | 390.52 | -0.54 |

higher (-0.36 ppm) over North Africa than over Equatorial Africa (-0.17 ppm ) and Southern Africa (0.01 ppm). This is likely due to the presence of strong local source from emissions and long range transport from the Northern Hemisphere as reported in other studies (Williams et al., 2007; Carré et al., 2010).

Figs. 6d - 8d display annual growth rate of $XCO_2$ which ranges from 1.5 to 2.7 $ppm\ yr^{-1}$. Moreover, the two datasets are consistent in determining the annual growth rate. The results are found in good agreement with the observed variability in the globally annual growth rate from surface measurements (http://www.esrl.noaa.gov/ gmd/ccgg/trends/global.html) which is 1.67, 2.39, 1.70, 2.40, 2.51 ppm $yr^{-1}$ global during 2009 - 2013 respectively, and 1.89, 2.42, 1.86,2.63, 2.06 ppm $yr^{-1}$ for Mauna Loa during 2009 - 2013 respectively, with error bars of 0.05 - 0.09 ppm $yr^{-1}$ for global and 0.11 ppm $yr^{-1}$ for Mauna Loa data sets(Kulawik et al., 2015). The growth rate may not be conclusive due to short length of the data sets used. However, it reflects how the CT and GOSAT observations perform with respect to each other.

**3.4 Comparison of seasonal climatology**

Seasonal cycle has important implications for flux estimates (Keppel-Aleks et al., 2012). It is important to analyse whether there are seasonally dependent biases that are affecting the seasonal cycle, and whether the data sets are capturing the same seasonal cycle. The four seasons considered here are: winter (December, January and February or in short DJF), spring (March, April and May or in short MAM ), summer (June, July and August or in short JJA), and autumn (September, October and November or in short SON). Fig. 9 shows the seasonal distributions of CT2016 (left panels) and GOSAT (middle panels) $XCO_2$ and their difference (CT2016 - GOSAT, right panels). The distribution clearly shows that $XCO_2$ concentration is maximum during spring (MAM) and minimum during autumn (SON) over the North Africa. On the other hand maxima is found during autumn

(SON) and minima during winter (DJF) over the Southern Africa. These features are in good agreement with the rainfall climatology of northern and southern hemispheres. Moreover, Table 5 shows seasonally varying biases. Seasonal biases affect the seasonal cycle and amplitudes, which are important for biospheric flux attribution (Lindqvist et al., 2015).

[Figure]

**Figure 9.** Seasonal climatology of $XCO_2$ for NOAA CT2016 (left panels) and GOSAT (midel panels) and their difference (right panels).

The right panels in Fig. 9 show that the seasonal mean difference (CT2016 - GOSAT) ranges from -4 to 6 ppm. A maximum difference of 6 ppm over the gulf of Guinea and Congo during JJA. However, such maximum difference was observed over Southern Africa during DJF. A minimum of -4 ppm over annual mean ITCZ region was observed during DJF and MAM. Moreover, the difference is above 1 ppm over Southern Africa regions during DJF and MAM (wet season of the region). This implies high spatial variability in the seasonal mean difference (see also Table 5). It also suggests that the discrepancy between the CT2016 and GOSAT becomes significant when vegetation cover is weak during DJF and MAM (dry seasons) over North Africa.

During SON the seasonal difference in most Africa's land region ranges from -2 to 1 ppm. The result implies CT2016 simulates lower values of $XCO_2$ than that of GOSAT observation indicating that there is a better spatial consistency during this season. Furthermore, during these seasons both the Northern and Southern Africa have a moderate vegetation cover following their respective summer seasons. The two datasets show lower regional variation (i.e., only from -2 to 2 ppm) over most of Africa land mass. However, the Equatorial Africa exhibits the mean difference lower than -2 ppm during DJF and MAM. This indicates the model tends to simulate lower than GOSAT retrievals $XCO_2$ over the region. In addition, this strong negative bias is partially due to positive bias in GOSAT $XCO_2$ retrieval due to cirrus clouds. For example, O'Dell et al. (2012) noted that GOSAT $XCO_2$ retrievals are positively biased due to thin cirrus clouds. Fig. 9 (right panels) reveals $XCO_2$ from CT2016

[Figure]

**Figure 10.** Histogram of difference for the seasonal $XCO_2$ climatology for DJF (a), MAM(b), JJA (c) and SON (d) seasons.

is lower than GOSAT $XCO_2$ over Northern Africa. The underestimation of observed $XCO_2$ by NOAA CT2016 model is likely related with the skill of driving ERA-Interim data as noted from previous studies. For example, Mengistu Tsidu (2012) has shown that the ERA-Interim data has a wet bias over Ethiopian highlands. Mengistu Tsidu et al. (2015) have also shown that ERA-Interim precipitable water is higher than measurements from radio-sonde, FTIR and GPS observations. Therefore,

5   such wet bias in the driving ERA-Interim GCM might have forced NOAA CT2016 to generate dense vegetation which serve as $CO_2$ sink. In other study, Nagarajan and Aiyyer (2004) found ECMWF has a cold bias in the lower atmosphere between 1000 to 750 hPa against independent upper-air sounding data which may affects $CO_2$.

   Fig. 10 shows mean difference between CT2016 and GOSAT $XCO_2$ seasonal means which ranges from -0.37 to 0.04 ppm with a standard deviation within a range of 1.00 to 1.91 ppm over the continent. The highest mean difference of $XCO_2$ (-0.37

10  ppm) occurs during SON and the lowest (0.04 ppm) occurs during MAM. Table 5 presents the summary of statistical values for spatial mean of each season mean. The comparison between the two data sets also shows there is a strong correlation (>0.5) during each season over the continent. However, there is moderate correlations (0.3 to 0.5) during DJF and MAM over North Africa and during DJF over Southern Africa. The low correlation over Northern Africa may be linked to a weak absorption by vegetation and a strong emission from human activities during winter as reported elsewhere (Liu et al., 2009; Kong et al., 2010).

15  Moreover, Table 5 shows that the seasonal biases are negative over North Africa while they are mostly positive over Equatorial and Southern Africa. Negative biases are observed during DJF and SON over Equatorial and Southern Africa respectively implying that $XCO2$ from CT2016 are lower than GOSAT during dry seasons.

**Table 5.** Summary of statistical relation between CT2016 and GOSAT $XCO_2$: Bias, correlation (R), Root mean square deviation (RMSD), standard deviation of $XCO_2$ from CT2016 simulation (CT2016 std), standard deviation of $XCO_2$ from GOSAT observation (GOSAT std), aggregate number of coincident observations (number of data) and number of grids over the region (grid). Negative bias means CT2016 is lower than GOSAT. The statistics are on the basis of spatial average of seasonal averages of bias, correlation, RMSD and standard deviations.

| Region | Statistics | Bias (ppm) | R | RMSD (ppm) | CT2016 std (ppm) | std in GOSAT (ppm) | number of data | grid |
|---|---|---|---|---|---|---|---|---|
| Africa | DJF | 0.06 | 0.73 | 1.91 | 1.15 | 2.57 | 135865 | 409 |
| | MAM | 0.04 | 0.92 | 1.62 | 1.98 | 3.25 | 95942 | 410 |
| | JJA | 0.22 | 0.65 | 1.59 | 1.12 | 2.08 | 116360 | 400 |
| | SON | -0.37 | 0.76 | 1 | 0.94 | 1.52 | 124233 | 408 |
| North Africa | DJF | -0.25 | 0.36 | 1.08 | 0.67 | 1.12 | 103913 | 204 |
| | MAM | -0.72 | 0.44 | 1.11 | 0.62 | 1.24 | 65115 | 204 |
| | JJA | -0.42 | 0.73 | 1.17 | 0.9 | 1.66 | 60854 | 204 |
| | SON | -0.35 | 0.66 | 0.53 | 0.52 | 0.71 | 91778 | 204 |
| Equatorial Africa | DJF | -0.52 | 0.68 | 2.47 | 1.06 | 3.07 | 22639 | 121 |
| | MAM | 0.18 | 0.9 | 1.88 | 1.94 | 3.46 | 8300 | 115 |
| | JJA | 1.51 | 0.59 | 2.02 | 1.46 | 2.52 | 12714 | 104 |
| | SON | 0.25 | 0.7 | 1.3 | 1.16 | 1.83 | 10213 | 113 |
| Southern Africa | DJF | 1.61 | 0.42 | 1.72 | 0.88 | 1.9 | 9313 | 84 |
| | MAM | 1.56 | 0.67 | 0.97 | 0.82 | 1.31 | 22527 | 91 |
| | JJA | 0.18 | 0.81 | 0.78 | 0.93 | 1.31 | 42792 | 92 |
| | SON | -1.16 | 0.77 | 0.81 | 0.84 | 1.26 | 22242 | 91 |

**3.5 Comparison of mean $XCO_2$ from NOAA CT16NRT17 and OCO-2**

The strong El Niño event occurred during 2015-2016 provides an opportunity to compare the performance of CT16NRT17 during strong El Niño events. Because of the decline in terrestrial productivity and enhancement of soil respiration, the concentration of $CO_2$ increases during El Niño events (Jones et al., 2001). In this section we compare mean $XCO_2$ of NOAA CT16NRT17 and NASA's OCO-2 covering the period from January 2015 to December 2016. OCO-2 is the most recent full-time dedicated $CO_2$ measuring satellite with greater spatio-temporal resolution.

The comparison was done based on the selection criteria discussed in Section 2.4. Fig. 11 shows mean distribution of $XCO_2$ from CT16NRT17 (Fig. 11a) and OCO-2 (Fig. 11b) over Africa's land mass. CT16NRT17 shows high ( $> 400$ ppm) $XCO_2$ values over North Africa while these high $XCO_2$ values are observed over Equatorial Africa in the case of OCO-2

[Figure]

**Figure 11.** Distribution of two years average $XCO_2$ of CT16NRT17 **(a)** and OCO-2 **(b)** $XCO_2$ and their difference **(c)** gridded in $3^0 \times 2^0$ bins; and **(d)** the total number of datasets at each grid

observation. The two datasets show a discrepancy over Equatorial Africa, where CT16NRT17 simulates low $XCO_2$ values ($< 401$ ppm) while OCO-2 observes high values of $XCO_2$ ($> 401$ ppm). Both datasets show moderate $XCO_2$ values which ranges from 397 to 400 ppm over Southern Africa. The $XCO_2$ distribution from OCO-2 is consistent with the maximum $CO_2$ concentration reported in past study (Williams et al., 2007) implying that the CT16NRT17 likely underestimates $XCO_2$ values over Equatorial Africa. It is also possible that the discrepancy is a compounded effect of OCO-2 $XCO_2$ positive bias over the region (O'Dell et al., 2012; Chevallier, 2015). Fig. 11c shows the mean difference between two years mean of $XCO_2$ from CT16NRT17 and OCO-2, which is in the range from -2 to 2 ppm. However, high ($<$-2 ppm) negative mean difference between the two data sets over rain forest regions (Gulf of Guinea and Congo basin) and ITCZ zone over Eastern Africa (South Sudan and southeastern Sudan) is observed implying that CT16NRT17 simulates lower $XCO_2$ values than that of OCO-2 observation over regions where vegetation uptake is strong. Conversely, high ($>$1) positive mean difference over the Sahara desert, Somalia and Tanzania implies CT16NRT17 simulates higher $XCO_2$ values than OCO-2 observation where the vegetation uptake is weak. Moreover, a positive ($>$2) mean difference over Egypt, Libya, Sudan, Chad, Niger, Mali and Mauritania is likely due to overestimates of $XCO_2$ emission from local sources by CT16NRT17. Overall, the two datasets show a fairly reasonable agreement with a correlation of 0.60 and offset of 0.36 ppm, a regional precision of 2.51 ppm and a regional accuracy of 1.21 ppm.

**Table 6.** Summary of statistical relation between CT16NRT17 and OCO-2 observation. The statistical tools shown are the mean correlation coefficient (R), the average of bias (Bias), the average root mean square deviation (RMSD), the standard deviation in bias (std of Bias), mean posteriori estimate of $XCO_2$ error from OCO-2 (OCO-2 err), the standard deviation in CT16NRT17 $XCO_2$ (CT16NRT17 std) and the standard deviation in OCO-2 $XCO_2$ (OCO-2 std). Positive Bias indicates that CT16NRT17 is higher than OCO-2. The number of data used in the statistics is 1,659,411 over 426 pixels covering the study period. Distribution at each grid point is shown in Fig 11d.

| Statistical tool | R | Bias (ppm) | RMSD (ppm) | std of Bias (ppm) | OCO-2 err (ppm) | CT16NRT17 std (ppm) | OCO-2 std (ppm) |
|---|---|---|---|---|---|---|---|
| Values | 0.6 | 0.34 | 2.57 | 1.21 | 0.55 | 0.55 | 1.28 |

Fig. 12a shows the histogram of two years mean difference, which is characterized by a positive mean of 0.34 ppm and a standard deviation of 1.21 ppm. This suggests that CT16NRT17 simulates high $XCO_2$ as compared to observations from OCO-2 over Africa's land mass.

[Figure]

**Figure 12.** Histogram of the difference of CT16NRT17 relative to OCO-2 (left panel) and color code scatter diagram of $XCO_2$ concentration as derived from CT16NRT17 and OCO-2 (right panel). Color indicates the relative distance as shown in colorbar between datasets.

Because of presence of spatial and temporal mismatch of some level between CT16NRT17 and OCO-2 datasets, it is important to assess the effect of relative distance between the datasets. Fig. 12b shows a color coded distribution of the two datasets. In the figure color codes indicate the relative distance. The random scatter of blue dots implies that the statistical discrepancies do not arise from the relative distance between the two datasets. More specifically, a statistical comparison of datasets lower

and higher than the $50^{th}$ percentile ($1.2^0$) shows bias of 0.58 and 0.57 ppm, correlation of 0.57 and 0.57 and RMSD of 2.65 and 2.67 ppm respectively.

[Figure]

**Figure 13.** The bias (a), correlation (b), RMSD (c) of model and OCO-2 $XCO_2$ and mean posteriori estimate of $XCO_2$ error from OCO-2 (d).

Fig. 13 shows the comparison of mean $XCO_2$ from CT16NRT17 and OCO-2 covering the period from January 2015 to December 2016. The number of data used are displayed in Fig. 11d. Fig. 13a depicts the bias which ranges from -2 to 2 ppm
5 with a mean bias of 0.34 ppm. However higher biases (<-2 ppm) are observed over Equatorial Africa along the annual average location of ITCZ. Fig. 13b shows the correlation map with values from 0.2 to 0.8 over Africa's land mass. A good correlation exceeding 0.6 are seen over many regions of the continent while weak correlation of less than 0.2 and higher root mean square error ($>$ 3 ppm ) are observed over small pockets of Equatorial and Eastern Africa regions (see Fig. 13c). These regions also show high ($>$ 0.65 ppm) error in satellite retrieval (see Fig. 13d). In addition, Fig. 11d shows the number of observations
10 are small ($<$ 1000 ) over the regions. This may contribute to the observed discrepancy over these regions. However, weak correlations are also observed over a wider area in North Africa such as Mauritania, Mali, Algeria and some regions of Niger where satellite errors are low and sufficient data are obtained. This indicates the necessity of incorporating more measurement in Carbon Tracker assimilation over North Africa in order to tune the assimilation model such that it captures the sub-regional carbon cycles. The poor correlation and high RMSD values observed over Ethiopia highland is likely due to the inefficiency of
15 retrieving $XCO_2$ from satellites over high-latitude lands (Chevallier, 2015).

**3.6 Categorical comparison of $XCO_2$ from NOAA CT16NRT17 and OCO-2**

Fig. 14 depicts the QBias, QPOD, QCSI, QMISS and QFAR between CT16NRT17 and OCO-2 at different thresholds. The maps clearly show that QFAR and QMISS increase with increasing threshold. On the other hand, QBias, QPOD and QCSI decrease with increasing thresholds (see also Fig. 15).

5     In our analysis, threshold is determined based on OCO-2 observation. The value of $XCO_2$ at $90^{th}$ percentile is 404.25 ppm and most CT16NRT17 $XCO_2$ value are below 404 ppm over North Africa. Therefore, categorical analysis shows white space over these regions for thresholds above $90^{th}$ percentile. Fig. 14 also shows that CT16NRT17 simulates the observation at lower quantiles. However, discrepancy between CT16NRT17 and OCO-2 is significant at higher quantiles. At $75^{th}$ quantile the QPOD and QCSI values are less than 0.6 over Africa's land mass which implies more than 40% of the OCO-2 observation 10  were not captured in the CT16NRT17 simulation.

**3.7 Comparison of monthly average time series of NOAA CT16NRT17 and OCO-2 $XCO_2$**

Figs. 16 - 18 show a two year monthly average time series comparison of $XCO_2$ from CT16NRT17 and OCO-2 over North Africa, Equatorial Africa and Southern Africa respectively. Fig. 16a shows the existence of good agreement between the two datasets in describing pattern over North Africa. Moreover, both datasets show a decreasing trend of $XCO_2$ from May 15  to September while increasing trend from October to April. On the other hand, consistent with the climate condition and associated $CO_2$ exchange, the monthly mean $XCO_2$ shows a maximum value of 403.37 ppm for CT16NRT17 and 402.06 ppm for OCO-2 during May. Conversely, a minimum concentration of 398.77 ppm from CT16NRT17 simulation and 398.27 ppm from OCO-2 observation are found in September. In addition, both CT16NRT17 and OCO-2 show maximum $XCO_2$ values (402.15 ppm for CT16NRT17 and 402.03 ppm for OCO-2) in December. These pick values in December are not 20  surprising, because the 2015-2016 El Niño started on March 2015 and reached pick in December 2015 which added extra $CO_2$ into the atmosphere (Chatterjee et al., 2017). Fig. 16a also shows that $XCO_2$ from CT16NRT17 simulation are higher than OCO-2 observation over North Africa.

    Fig. 16b shows the monthly mean difference between CT16NRT17 and OCO-2 which ranges from -0.5 to 2 ppm. As a consequence of strong El Niño, the region misses the short rain season during spring (MAM) when vegetation are still under 25  the influence of the dormancy of winter (DJF). However, following the summer vegetation awakens from the long dormancy and becomes a sink of $CO_2$. Starting from August 2015, the difference between the two datasets is minimum; this implies that CT simulates low values of $XCO_2$ when the vegetation uptake is strong. On the other hand, a maximum difference of exceeding 1 ppm was observed during MAM, implying higher $XCO_2$ values from CT16NRT17 simulation than that of OCO-2 when vegetation uptake is weak following the strong El Niño over North Africa. Moreover, Fig. 16c displays a monthly mean 30  regional mean bias of 0.87 ppm, correlation of 0.95 and a root mean square deviation of 0.72 ppm between CT16NRT17 and OCO-2 $XCO_2$. This implies that CT16NRT17 is in a good agreement with OCO-2. However, a small discrepancies arose due to likely a strong anthropogenic emission from Nigeria, Egypt and Algeria together with the establishment of plantation over

[Figure]

**Figure 14.** Distribution of categorical metrics over the study area for quantiles exceeding 5% (first row), 75% (second row), 90% (third row) and 95% (fourth row).

[Figure]

**Figure 15.** Summary of categorical metrics (QBias, QPOD, QCSI, Qmis and QFAR) averaged over Africa's land region for 5, 10, 25, 50, 75, 90 and 95 percentiles.

[Figure]

**Figure 16.** The monthly mean time series of CT16NRT17 and OCO-2 from January 2015 to December 2016 averaged over North Africa (a), bias associated to the monthly means (b), the histogram of difference (c) and the annual growth rate obtained by subtracting the mean from the mean of the next year (d). The error bars in (a) shows the OCO-2 a posteriori $XCO_2$ uncertainty.

North Africa, which recently exceeded deforestation, and resulted in net flux of carbon sink (Canadell et al., 2009). This might have contributed to the observed discrepancy over North Africa.

Figs. 17a - 18a show monthly mean time series of $XCO_2$ from the model and OCO-2 instrument over Equatorial Africa and Southern Africa which are also in good agreement in terms of pattern. However, the figures show that CT16NRT17 simulations

5   are lower than those of OCO-2 during October, November and December whereas it is opposite during April, May and June over Equatorial Africa and Southern Africa. Figs. 17b and 18b depict a seasonal bias in the monthly time series over Equatorial Africa and Southern Africa respectively. Positive biases are observed during dry seasons while negative biases are during wet seasons. Moreover, the datasets have monthly averaged regional mean biases of 0.13 and 0.11 ppm, correlation of 0.90 and 0.94, RMSD of 0.84 and 0.73 ppm over Equatorial Africa and Southern Africa respectively. This shows that existence of

10   better agreement between CT16NRT17 and OCO-2 over these regions in terms of monthly average regional mean values. Figs. 16d-18d show both CT16NRT17 and OCO-2 are in good agreement in estimating the annual growth rate. Patra et al. (2017) found a global mean of more than 3 ppm of $CO_2$ added to the atmosphere due to the strong El Niño event that occurred during 2015-2016. In agreement with this, both CT16NRT17 and OCO-2 shows an annual growth rate that ranges from 3.10 to 3.42 ppm year$^{-1}$ of $XCO_2$ over Africa's land mass (see also Table 7). However, over all regions of Africa's land mass CT16NRT17

15   shows lower $XCO_2$ annual growth rate than those of OCO-2.

[Figure]

**Figure 17.** The same as in Fig. 16 but over Equatorial Africa.

**3.8   Comparison of seasonal means of NOAA CT16NRT17 and OCO-2 $XCO_2$**

Fig. 19 depicts seasonal means of $XCO_2$ over Africa's land mass from CT16NRT17 (left panels), OCO-2 (middle panels) and their difference (right panels) covering period of January 2015 to December 2016. The white space seen over some regions

[Figure]

**Figure 18.** The same as in Fig. 16 but over Southern Africa.

**Table 7.** Annual growth rate (AGR) of $XCO_2$ over Africa land mass from CT16NRT17 and OCO-2. The results are obtained as the mean annual difference of 2015 and 2016 values

| Region | AGR of CT (ppm year$^{-1}$) | AGR Of OCO-2 (ppm year$^{-1}$) |
|---|---|---|
| North Africa | 3.10 | 3.33 |
| Equatorial Africa | 3.14 | 3.42 |
| Southern Africa | 3.20 | 3.16 |

(e.g., Mali during JJA) is due to insufficient coincident satellite data according to the selection criteria during these seasons. $XCO_2$ increases from winter to spring and then decreases from spring peak to summer minimum over the whole continent. The decrease from spring maximum to summer continued into autumn over northern half of Africa in contrast to southern half of Africa which exhibits an increase in $XCO_2$. The decrease from spring to autumn (northward of equator) and until

5   summer (southward of equator) is likely to be a consequence of the land vegetation awakening from dormancy of winter and partly spring. Conversely, the decomposition of died and decayed vegetation which began in autumn and continued throughout winter adds extra $CO_2$ leading to a maximum concentration during spring (Idso et al., 1999). In agreement with this, both CT16NRT17 and OCO-2 show maximum $XCO_2$ during MAM over North Africa and during SON over Southern Africa. Conversely, minimum $XCO_2$ are observed during SON over North Africa and during DJF over South Africa.

10   Fig. 19 (right panels) shows the seasonal mean difference of CT16NRT17 and OCO-2. A higher mean difference of greater than 1 ppm are observed over North Africa during DJF and MAM when the vegetation cover over the region decreases. This indicates that $XCO_2$ values from CT16NRT17 are higher than that of OCO-2 when vegetation uptake is weak. On the other

[Figure]

**Figure 19.** Seasonal mean of $CO_2$ for NOAA CT16NRT17 (left panels) and OCO-2 (middle panels) and their difference (right panels).

hand, higher negative mean difference of less than -2 ppm are observed over Equatorial Africa during DJF and during SON over Southern Africa. This difference between the CT and OCO-2 arises likely during forest fire that naturally occurs following their respective dry season. Consistent with report by Liang et al. (2017), low seasonal variability is observed between CT16NRT17 and OCO-2 in the range from -4 to 4 ppm with greater amplitude over North and Equatorial Africa than over Southern Africa
5  (see Figs. 19 (right panels)).

Fig. 20 shows the histogram of seasonal mean difference of CT16NRT17 and OCO-2. The smaller standard deviation of 1.49 and 1.07 are observed during JJA and SON. On the other hand, higher standard deviation of 1.69 and 1.75 ppm are observed during DJF and MAM respectively. The results indicate that CT16NRT17 and OCO-2 show a better consistency during wet seasons and this consistency decreases as the vegetation cover decreases over most regions of Africa land mass during dry
10  seasons.

**4   Conclusions**

In this study, the NOAA CT2016 $XCO_2$ values are compared with two full-time $CO_2$ dedicated satellites, GOSAT and OCO-2 over Africa land mass. Comparison between CT2016 and GOSAT were done using a five years datasets covering the period from May 2009 to April 2014. This comparison is important to test the performance of CT2016 in capturing climatology of the
15  $XCO_2$. Comparison of CT16NRT17 with OCO-2 was done using two years data during the strong El Niño event from January 2015 to December 2016. This provide opportunity to assess the performance of CT16NRT17 simulation during strong El Niño

[Figure]

**Figure 20.** Histogram of difference for the seasonal $CO_2$ climatology for DJF (a), MAM(b), JJA (c) and SON (d) seasons.

events. Comparison of Carbon Tracker with the two satellites reveals biases of -0.28 and 0.34 ppm, correlations of 0.83 and 0.60 and root mean square deviations of 2.30 and 2.57 ppm with respect to GOSAT and OCO-2 respectively. The performance of the model in capturing the whole distribution is also assessed. It is found that Carbon Tracker can capture more than 93% of the observation higher than the lowest ($5^{th}$ percentile) thresholds. However, the quantile probability of detection and the
5    quantile critical success index decrease with increasing threshold implying the model and satellite show some discrepancies over the extreme values exceeding $90^{th}$ percentile.

The monthly average time series of CT2016 over North Africa, Equatorial Africa and Southern Africa are separately compared with $XCO_2$ from the two satellites. CT2016 agrees well with measurements from the two instruments in terms of pattern and amplitude. However, this agreement deteriorates over Equatorial and Southern Africa in terms of amplitude. It is
10   also found that there is a seasonal dependent bias between them which is negative during dry seasons while it is positive during wet seasons. This indicates results of CT2016 are mostly lower than the GOSAT observation. High spatial mean of seasonal mean RMSD of 1.91 during DJF and 1.75 ppm during MAM and low RMSD of 1.00 and 1.07 ppm during SON in the model $XCO_2$ with respect to GOSAT and OCO-2 are observed respectively thereby indicating better agreement between CT and the satellites during autumn. CT2016 has the ability to capture monthly time series and seasonal cycles. However, $XCO_2$ from
15   CT2016 is lower than GOSAT observations over North Africa during all seasons whereas $XCO_2$ from CT2016 is higher than that of GOSAT over Equatorial and Southern Africa with the exceptions of DJF over Equatorial Africa and SON over Southern Africa. In addition, CT2016 simulates lower $XCO_2$ than the observations over some regions (e.g., Congo, South Sudan and southwestern Ethiopia) and during summer season over the whole continent following large vegetation uptake. In contrast, $XCO_2$ from CT16NRT17 is higher than that of OCO-2 over North Africa whereas it is lower than that of OCO-2 during DJF
20   and SON over Equatorial and Southern Africa respectively.

In general, $XCO_2$ from NOAA CT shows a very small bias with respect to GOSAT and OCO-2 observation over Africa's land mass. Moreover, there is a good agreement between CT simulation and observations in terms spatial distribution, monthly average time series and seasonal climatology. However, there are some discrepancies between the model and the two $XCO_2$ datasets from GOSAT and OCO-2 implying that the accuracy of the model data needs further improvements for the rain forest regions (e.g., Congo) through assimilation of in-situ observations and tuning of the model through process studies. Further work may also be needed to improve the $XCO_2$ from satellites and models in the extreme parts of $XCO_2$ distribution.

*Acknowledgements.* The authors acknowledge NOAA Earth System Research Laboratory and NASA GOSAT for the data products. The first author also acknowledges Addis Ababa University, Addis Ababa Science and Technology University, Botswana International University of Science and Technology for their support through fellowship and access to the research facilities..